# The PMIP4 contribution to CMIP6 – Part 1: Overview and over-arching analyses plan*

Masa Kageyama [1], Pascale Braconnot[1], Sandy P. Harrison[2], Alan M. Haywood[3], Johann H. Jungclaus[4], Bette L. Otto-Bliesner[5], Jean-Yves Peterschmitt[1], Ayako Abe-Ouchi[6,7], Samuel Albani[8], Patrick J. Bartlein[9], Chris Brierley[10], Michel Crucifix[11], Aisling Dolan[3], Laura Fernandez-Donado[12], Hubertus Fischer[13], Peter O. Hopcroft[14], Ruza F. Ivanovic[3], Fabrice Lambert[15], Daniel J. Lunt[14], Natalie M. Mahowald[16], W. Richard Peltier[17], Steven J. Phipps[18], Didier M. Roche[1,19], Gavin A. Schmidt[20], Lev Tarasov[21], Paul J. Valdes[14], Qiong Zhang[22], Tianjun Zhou[23]

[1]Laboratoire des Sciences du Climat et de l'Environnement, LSCE/IPSL, CEA-CNRS-UVSQ, Université Paris-Saclay, F-91191 Gif-sur-Yvette, France
[2]Centre for Past Climate Change and School of Archaeology, Geography and Environmental Science (SAGES) University of Reading, Whiteknights, RG6 6AH, Reading, United Kingdom
[3]School of Earth and Environment, University of Leeds, Woodhouse Lane, Leeds, LS2 9JT, United Kingdom
[4]Max Planck Institute for Meteorology, Bundesstrasse 53, 20146 Hamburg, Germany
[5]National Center for Atmospheric Research, 1850 Table Mesa Drive, Boulder, Colorado 80305, United States of America
[6]Atmosphere Ocean Research Institute, University of Tokyo, 5-1-5, Kashiwanoha, Kashiwa-shi, Chiba 277-8564, Japan
[7]Japan Agency for Marine-Earth Science and Technology, 3173-25 Showamachi, Kanazawa, Yokohama, Kanagawa, 236-0001, Japan
[8]Institute for Geophysics and Meteorology, University of Cologne, Cologne, Germany
[9]Department of Geography, University of Oregon, Eugene, OR 97403-1251, United States of America
[10]University College London, Department of Geography, WC1E 6BT, United Kingdom
[11]Université catholique de Louvain, Earth and Life Institute, Louvain-la-Neuve, Belgium
[12]Dpto. Física de la Tierra, Astronomía y Astrofísica II, Instituto de Geociencias (CSIC-UCM), Universidad Complutense de Madrid, Spain
[13]Climate and Environmental Physics, Physics Institute & Oeschger Centre for Climate Change Research, University of Bern, Sidlerstrasse 5, CH-3012 Bern, Switzerland
[14]School of Geographical Sciences, University of Bristol, Bristol, United Kingdom
[15]Catholic University of Chile, Department of Physical Geography, Santiago, Chile
[16]Department of Earth and Atmospheric Sciences, Bradfield 1112, Cornell University, Ithaca, NY 14850, United States of America
[17]Department of Physics, University of Toronto, 60 St. George Street, Toronto, Ontario M5S 1A7, Canada
[18]Institute for Marine and Antarctic Studies, University of Tasmania, Private Bag 129, Hobart, TAS 7001, Australia
[19]Earth and Climate Cluster, Faculty of Earth and Life Sciences, Vrije Universiteit Amsterdam, Amsterdam, the Netherlands
[20]NASA Goddard Institute for Space Studies and Center for Climate Systems Research, Columbia University 2880 Broadway, New York, NY 10025, United States of America
[21]Department of Physics and Physical Oceanography, Memorial University of Newfoundland and Labrador, St. John's, NL, A1B 3X7, Canada
[22]Department of Physical Geography, Stockholm University, Stockholm, Sweden
[23]LASG, Institute of Atmospheric Physics, Chinese Academy of Sciences, P.O. Box 9804, Beijing 100029, China

*Correspondence to*: Masa Kageyama (Masa.Kageyama@lsce.ipsl.fr)

* This paper is the first of a series of 4 GMD papers on the PMIP4-CMIP6 experiments. Part 2 (Otto-Bliesner et al. 2016) gives the details about the two PMIP4-CMIP6 interglacial experiments, Part 3 (Jungclaus et al., 2016) about the last millennium experiment, and Part 4 (Kageyama et al., 2017) about the Last Glacial Maximum experiment. The mid Pliocene Warm Period experiment is part of the Pliocene Model Intercomparison Project (PlioMIP) - Phase 2, detailed in Haywood et al. (2016).

**Abstract.**

The goal of the Paleoclimate Modelling Intercomparison Project (PMIP) is to understand the response of the climate system to different climate forcings and feedbacks. Through comparison with observations of the environmental impact of these climate changes, or with climate reconstructions based on physical, chemical or biological records, PMIP also addresses the issue of how well state-of-the-art numerical models simulate climate change. Paleoclimate states can be radically different from those of the recent past documented by the instrumental record, and thus provide an out-of-sample test of the models used for future climate projections and a way to assess whether they have the correct sensitivity to forcings and feedbacks. Five different periods have been designed to contribute to the objectives of the sixth phase of the Coupled Model Intercomparison Project (CMIP6): the millennium prior to the industrial epoch (past1000), the mid-Holocene, 6,000 years ago (midHolocene); the Last Glacial Maximum, 21,000 years ago (lgm); the Last Interglacial, 127,000 years ago (lig127k) and mPWP, the mid-Pliocene Warm Period, 3.2 million years ago (midPliocene-eoi400). These climatic periods are well documented by paleoclimatic and paleoenvironmental records, with climate and environmental changes relevant for the study and projections of future climate changes. This manuscript describes the motivation for the choice of these periods and the design of the numerical experiments, with a focus on their novel features compared to the experiments performed in previous phases of PMIP and CMIP. Analyses of the individual periods, across all the periods and comparisons with other CMIP6 simulations, will allow examination of relationships between forcings of different nature and amplitude and climate responses, and comparison of the processes involved in these responses. The evolution of interannual variability in the past is also expected to provide some clues on the linkages between mean climate and climate variability. This manuscript also describes the information needed to document each experiment, the experimental protocols, and the model outputs required for analysis and benchmarking.

# 1 Introduction

## 1.1 Why model paleoclimates?

Instrumental meteorological and oceanographic data, available for the period extending from the middle of the 19[th] century, describe the manner in which Earth's surface climate has evolved since the beginning of the industrial revolution. These data show a global warming of ~0.85°C has occurred since this time, a warming that is more intense over land than over the oceans, and more intense at high latitudes compared to the tropics (Hartmann et al., 2013; Sutton et al., 2007). This recent climate change has been largely induced by the increase of atmospheric greenhouse gases due to human activities, amplified by the action of feedbacks such as those associated with atmospheric water vapor and clouds (e.g. Dufresne and Bony, 2008) and the albedo of snow and ice. Changes in the land cover or in ocean properties and circulation (Cubasch et al., 2013) are other feedbacks on climate. Aerosol forcing related to human activities has also had an impact on climate, although at regional level. This process-based understanding of the climate system is embedded in the climate models used to project changes in future climates. The skill of these climate models is most commonly evaluated in comparison to the present climate and climate change since the pre-industrial age (1850 CE). However, concentrations of atmospheric greenhouse gases are projected to increase significantly during the 21[st] century, reaching levels well outside the range of recent millennia. Thus, in making future projections, models are operating well outside the conditions for which they have been calibrated. Current climate conditions do not provide a full understanding of how climate responds to various external factors. The credibility of climate projections needs to be assessed using information about longer-term paleoclimate changes, particularly for intervals when the climate change compared to present was as large as the anticipated future change.

We have to look back several million years to find a period of Earth's history when atmospheric $CO_2$ concentrations were similar to the present day (the mid-Pliocene warm period, ca. 3.3 to 3 million years ago) and several tens of million years (e.g. the early Eocene, ~55 to 50 million years ago) to find concentrations similar to those possible by the end of this century under current emissions trajectories. During these ancient periods, land surface topography, ocean bathymetry, land-ocean distributions and/or the geometry of the ice sheets were different from today, and the mechanisms which led to high atmospheric $CO_2$ concentrations likely acted on timescales much slower than anthropogenic fossil fuel emissions. Although these periods are not perfectly analogous to the future, they can offer key insights into climate processes that operate in a higher $CO_2$, warmer world (e.g. Lunt et al., 2010, 2012; Caballero and Huber, 2010). During the Quaternary (2.58 million years ago to present), the Earth's geography was very similar to today and the main external factors driving climatic changes are the astronomical parameters, which determine the seasonal and latitudinal distribution of incoming solar energy. Changes in greenhouse gas concentrations and in ice sheets effectively acted as additional forcing factors on the dynamics of the atmosphere and the ocean. Rapid climate transitions, on human-relevant timescales (decades to centuries), have been documented for this most recent period (e.g. Marcott et al., 2014; Steffensen et al., 2008). By combining several past periods, we can provide a broad picture of the climate response to external forcings, and benefit from the rich resource of paleoclimates and paleoenvironments. There are numerous paleoclimate records documenting the evolution of Earth's climate before instrumental records (e.g., Harrison and Bartlein, 2012; Masson-Delmotte et al., 2013). These show large variations in the Earth's climate prior to the industrial era, commensurate with the magnitude of projected changes in the future.

Replicating the totality of those climate changes with state-of-the-art climate models, driven by appropriate forcings (e.g. insolation, atmospheric composition) and boundary conditions (e.g. ice sheets), is a challenge (Braconnot et al., 2012; Harrison et al., 2015). It is challenging, for example, to represent the correct amplitude of past climate changes such as glacial-interglacial temperature differences (e.g. the temperatures at the Last Glacial Maximum, ~21,000 years ago, vs. the pre-industrial temperatures, cf. Harrison et al., 2014) or the correct spatial patterns such as the northward extension of the African monsoon during the mid-Holocene, ~ 6,000 years ago (Perez-Sanz et al., 2014). Interpreting paleoenvironmental data can also be challenging, in particular if one wants to disentangle the relationships between changes in large-scale atmospheric or oceanic circulation, broad-scale regional climates and local environmental responses to these changes. This challenge is paralleled by concerns about future local or regional climate changes and their impact on the environment. Modelling paleoclimates is therefore a means to understand past climate and environmental changes better, using physically based tools, as well as a means to evaluate model skill in forecasting the responses to major drivers.

**1.2 The Paleoclimate Modelling Intercomparison Project (PMIP)**

The Paleoclimate Modelling Intercomparison Project (PMIP) was established in the 1990s (Joussaume and Taylor, 1995) in order to understand the mechanisms of past climate changes, in particular the role of the different climate feedbacks, and to evaluate how well the climate models used for climate projections simulate well-documented climate states outside the range of present and recent climate variability. To achieve these goals, PMIP has actively fostered paleoclimatic data syntheses, model-data comparisons and multi-model analyses. PMIP also provides a forum for discussion of experimental design and appropriate techniques for comparing model results with paleoclimatic reconstructions.

Since its beginning, PMIP has closely followed developments in climate modelling, in parallel to the Atmospheric Model Intercomparison Project (AMIP) and the Coupled Model Intercomparison project (CMIP). Each new phase of PMIP has therefore included the study of additional processes and/or feedbacks of the climate system and new possibilities for model-data comparisons (Braconnot et al., 2007; Braconnot et al., 2012). Two climatic periods have been a major focus throughout PMIP's history: the mid-Holocene (MH, ~6,000 years ago) and the Last Glacial Maximum (LGM, ~21,000 years ago). These two periods are considered as reference points for assessing the sensitivity of the climate system to changes in atmospheric $CO_2$ concentration and orbitally-induced changes in tropical circulation and the monsoons (Braconnot et al., 2012; Harrison et al., 2015). Evaluations of the MH and LGM simulations made in successive phases of PMIP provide a unique overview of the evolution of the ability of climate models to reproduce large changes compared to today (Harrison et al. 2013; Flato et al., 2013).

Paleoclimate experiments were included for the first time in the ensemble of simulations made during the fifth phase of CMIP (CMIP5, Taylor et al., 2012), equivalent to the third phase of PMIP (PMIP3). In addition to the MH and LGM simulations described above, transient simulations of the millennium prior to the industrial epoch (LM, 850-1849 CE, Schmidt et al., 2011, 2012) were also included in CMIP5 to study the mechanisms of decadal to centennial climate variability (natural variability vs. impact of solar, volcanic and anthropogenic

forcings). Thanks to this formal inclusion of paleoclimate simulations in the CMIP5 exercise, it was possible to compare the mechanisms causing past and future climate changes in a rigorous way and evaluate the models used for projections under climate states very different from the present one (e.g. Harrison et al., 2014, Harrison et al., 2015), providing out-of-sample validation.

A number of other time periods were included in PMIP3, in particular the mid-Pliocene Warm Period (mPWP, ca. 3.3 to 3 million years ago) via the PlioMIP project (Haywood et al., 2010, 2011) and the last interglacial period (130,000 to 115,000 years before present, Lunt et al., 2013) to examine whether climate models could produce a rate of ice-sheet melting in agreement with a global sea level at least 5m higher than now (Masson-

Delmotte et al., 2013; Dutton et al., 2015). Discussions on transient simulations of climate behaviour, focusing on the last interglacial period and the last deglaciation (Ivanovic et al., 2016) were also initiated, as were simulations of deeper time, in particular the early Eocene, ~50 million years ago (Lunt et al, 2012; Lunt et al, 2016).

A measure of the success of PMIP3 is provided by the number of participating modelling groups (more than 20) and the prominent role of PMIP results in the fifth IPCC assessment report (Masson-Delmotte et al., 2013; Flato et al., 2013). Moreover, PMIP3 also identified significant knowledge gaps and areas where progress is needed. PMIP4 has been designed to address these issues.

**1.3 PMIP4 experiments in CMIP6**

The design of the PMIP4 simulations included in CMIP6 was built on the recognition that PMIP simulations naturally address the key CMIP6 question "How does the Earth System respond to forcing?" (Eyring et al, 2016), for multiple forcings and in climates states very different from the current or historical climates. Comparisons with environmental observations and climate reconstructions enable us to determine whether the modelled responses are realistic. PMIP also addresses key question 2 "What are the origins and consequences of

systematic model biases?" PMIP simulations and data-model comparisons will show whether the biases in the present-day simulations are found in other climate states. More importantly, analyses of PMIP simulations will show whether present-day biases have an impact on the magnitude of simulated climate changes. Finally, PMIP is also relevant to question 3 "How can we assess future climate changes given climate variability, predictability and uncertainties in scenarios?" through examination of these questions for documented past climate states and

via the use of the last millennium simulations as reference state for natural variability.

The choice of time periods for paleoclimate experiments in CMIP6 is based on previous experience in PMIP. For each target period, there is a quantified understanding of the relevant climate drivers and an extensive network and/or synthesis of environmental observations (cf. Sections 2 and 4). The five periods proposed for PMIP4-

CMIP6 represent climate states with different greenhouse gas concentrations, astronomical parameters, ice sheet extents, and volcanic and solar activities (Figure 1), consistent with the need to provide a large sample of the climate response to different forcings. While the five periods represent very different climate states, all of them cover aspects of the climate system that are relevant to future climate change (Table 1). The periods are,

(abbreviated name is provided before the full name, name of corresponding PMIP4-CMIP6 experiment is given in italics within parentheses at the end of each line):

- LM, the millennium before the start of the industrial revolution, from 850 to 1849 CE (*past1000*)
- MH, the mid-Holocene, 6,000 years ago (*midHolocene*)
- 5    LGM, the Last Glacial Maximum, 21,000 years ago (*lgm*)
- LIG, the Last Interglacial, 127,000 years ago (*lig127k*)
- mPWP, the mid-Pliocene Warm Period, 3.2 million years ago (*midPliocene-eoi400*)

All experiments except *past1000* are equilibrium experiments, in which the imposed forcings are constant. All the experiments have been run by several modelling groups, and all except *lig127k* have been run as formal

intercomparisons with a standardized protocol. We have kept the PMIP3-CMIP5 (Taylor et al. 2012) names for simulations which were already part of this project, i.e. *past1000*, *midHolocene* and *lgm*, and the name of the mPWP experiment, *midPliocene-eoi400*, is consistent with the PlioMIP phase 2 naming convention for the mid-Pliocene Warm Period (Haywood et al, 2016). The mPWP experiment focuses on a specific interglacial, dated at ~3.2 Ma before present, during the wider mid-Pliocene Warm Period (3.3 to 3 Ma before present). All the

experiments can be run independently and have value for comparison to the CMIP6 DECK (Diagnostic, Evaluation and Characterization of Klima) and historical experiments (Eyring et al., 2016). They are therefore all considered as Tier 1 within CMIP6 (Table 1). It is not mandatory for groups wishing to take part in PMIP4-CMIP6 to run all five PMIP4-CMIP6 experiments. It is however mandatory to run at least one of the experiments that were run in previous phases of PMIP, i.e. the *midHolocene* or the *lgm*. These are considered as

"entry cards" for participation in PMIP4-CMIP6.

Figure 1: Context of the PMIP4 experiments (from left to right: mPWP, mid-Pliocene Warm Period; LIG, last interglacial; LGM, last glacial maximum; MH, mid-Holocene; LM, last millennium; H, CMIP6 historical simulation): (a)-(d) insolation anomalies (differences from 1950 CE), for July at 65°N, calculated using the programs of Laskar et al. (2004, panel (a)) and Berger (1978, panels (b)-(d)); (e) $\delta^{18}O$ (magenta, Lisiecki and Raymo, 2005, scale at left), and

sea level (blue line, Rohling et al., 2014; blue shading, a density plot of eleven mid-Pliocene sea level estimates (Dowsett and Cronin 1990; Wardlaw and Quinn, 1991; Krantz, 1991; Raymo et al., 2009; Dwyer and Chandler, 2009; Naish and Wilson, 2009; Masson-Delmotte et al., 2013; Rohling et al., 2014; Dowsett et al., 2016) scale at right); (f) and (g) $\delta^{18}O$ (magenta, Lisiecki and Raymo, 2005, $\delta^{18}O$ scale at left), and sea level (blue dots, with light-blue 2.5, 25, 75 and 97.5 percentile bootstrap confidence intervals, Spratt and Lisiecki, 2015; blue rectangle, LIG high-stand

range, Dutton et al., 2015; dark blue lines, Lambeck et al., 2014, sea-level scale at right on panel (g)), (h) sea level (Kopp, et al., 2016, scale at right); (i) $CO_2$ for the interval 3.0-3.3 Ma shown as a density plot of eight mid-Pliocene estimates (Raymo et al., 1996; Stap et al., 2016; Pagani et al., 2010; Seki et al., 2010; Tripati et al., 2009; Bartoli et al., 2011; Seki et al., 2010; Kurschner et al., 1996); (j) and (k) $CO_2$ measurements (Bereiter et al., 2015, scale at left); (l) $CO_2$ measurements (Schmidt et al, 2011, scale at right); (m) and (n) $CH_4$ measurements (Loulergue et al., 2008, scale

at left); (o) $CH_4$ measurements (Schmidt et al, 2011, scale at right); (p) volcanic radiative forcing (Schmidt et al., 2012, scale at right); (q) total solar irradiance (Schmidt et al., 2012, scale at right).

Table 1: Characteristics, purpose and CMIP6 priority of the five PMIP4-CMIP6 experiments

Intercomparisons of the simulated responses to specific drivers across different models are valuable as sensitivity experiments are a key aspect of the contribution of PMIP to answer the first of the CMIP6 key questions. However, the true power of PMIP is the connection to the environmental observations and climate reconstructions, which allows an assessment of model skill. As model-data comparisons are as essential to PMIP as comparisons between models, it is important to assess all the issues that might make those comparisons

difficult. Uncertainties in the paleoenvironmental observations, or perhaps more broadly in the climate

inferences made from those observations, are a key part of PMIP analyses, as is the structural uncertainty across the model responses. Both of these factors have been part of the PMIP approach from the beginning. Improved reconstructions, increased complexity and realism of climate simulations require putting more emphasis on understanding of impacts of the uncertainties in the driver themselves. This encompasses time-uncertainty in the reconstructions (e.g. are all data synchronous? what date should be used to compute the astronomical parameters to compare with available data?) as well as structural uncertainty in the boundary conditions applied (e.g. in the continental reconstructions, ice sheet height and extent, vegetation cover), and in the transient forcings (for instance in the last millennium simulations for solar, volcanic aerosol or land use/land cover change). Differences between plausible reconstructions of boundary conditions and forcings can impact the assessment of model skill. In these cases, we have included alternative forcings and boundary conditions for the PMIP4-CMIP6 experiment or to be used in PMIP4 sensitivity experiments (Jungclaus et al., 2016; Otto-Bliesner et al., 2016; Kageyama et al., 2017; Haywood et al., 2016).

In section 2, we give more background on the periods chosen for the CMIP6 experiments and the associated forcings and boundary conditions. The experimental set-up of the experiments is described in section 3. The analysis plan is outlined in Section 4. A short conclusion is given in section 5.

## 2. The PMIP4-CMIP6 simulations

### 2.1 PMIP4-CMIP6 entry cards: the mid-Holocene (*midHolocene*) and last glacial maximum (*lgm*)

The MH and LGM periods provide examples of strongly contrasting climate states (Figure 1, Table 1). There are extensive syntheses of marine and terrestrial data for both intervals, documenting environmental responses to changing climate (cf. Section 4). The MH provides an opportunity to examine the response to orbitally-induced changes in the seasonal and latitudinal distribution of insolation. It is a period during which the northern hemisphere was characterised by enhanced northern hemisphere summer monsoons, extra-tropical continental aridity and much warmer summers. The LGM provides an opportunity to examine the impact of changes in ice sheets and continental extent (which increases due to the drop in sea level) and of the decrease in atmospheric greenhouse gases on climate. The LGM is particularly relevant because the forcing and temperature response from the LGM to the Holocene was as large as to that projected from present to the end of the 21st century (Braconnot et al., 2012). Because these periods have been studied in earlier phases of PMIP, they provide the opportunity to evaluate whether increased model resolution and complexity, as well as the increased realism of the experimental set up, leads to improvement in model performance.

Evaluation of the PMIP3-CMIP5 MH and LGM experiments has demonstrated that climate models simulate changes in large-scale features of climate that are governed by the energy and water balance reasonably well, including changes in land-sea contrast (Figure 2a) and high-latitude amplification of temperature changes (Izumi et al., 2013; Izumi et al., 2015). They also simulate the scaling of precipitation changes with respect to temperature changes at the hemispheric scale realistically (Li et al., 2013). The evaluation of the PMIP3-CMIP5 MH and LGM simulations confirms that the simulated relationships between large-scale patterns of temperature and precipitation change in future projections are credible (Harrison et al., 2015). However, the PMIP3-CMIP5

simulations of MH and LGM climates show only moderate skill in predicting reconstructed patterns of climate change overall (Hargreaves et al., 2013; Hargreaves and Annan, 2014; Harrison et al., 2014; Harrison et al., 2015). This arises because of persistent problems in simulating regional climates (e.g. Mauri et al., 2014; Perez-Sanz et al., 2014; Harrison et al., 2015). State-of-the-art models cannot reproduce the northward penetration of the African monsoon in response to MH orbital forcing (Figure 2b, Perez-Sanz et al., 2014; Pausata et al., 2016), for example. This discrepancy was already noted in PMIP1 (Joussaume et al, 1999) and Figure 2b shows that there has been no improvement from PMIP1 to PMIP3; nor are models that include additional feedbacks (such as vegetation or carbon cycle) any better. While this could reflect inadequate representation of feedbacks, model biases could also contribute to this mismatch (e.g. Zheng and Braconnot, 2013).

Although the benchmarking of the PMIP3-CMIP5 MH and LGM experiments shows that some models consistently perform better than others (Harrison et al., 2014), better performance in paleoclimate simulations is not consistently related to better performance under modern conditions (Harrison et al., 2015). Hence the ability to simulate modern climate regimes and processes does not guarantee that a model will be good at simulating climate changes, emphasizing the importance of testing models against the paleoclimate record to increase confidence in projections of future climate (Braconnot et al., 2012; Hargreaves and Annan, 2014; Schmidt et al., 2014).

Figure 2: Data-model comparisons in PMIP2 and PMIP3-CMIP5: (a) Land-ocean contrast in past, present and projected future climates. The black dots are the simulated long-term mean differences (experiment – piControl) in the relative warming/cooling over global land and global ocean. The red crosses show simulated changes where the model output has been sampled only at the locations for which there are temperature reconstructions for the lgm, midHolocene and historical (post-1850 CE) CMIP5 simulations. Area averages of paleoclimate data are shown by bold blue crosses, with reconstruction uncertainties indicated by the finer lines. The regression line (magenta) shows that land-ocean contrasts are maintained across different climate states and are also consistent with paleoclimatic data. (b) Boxplots of reconstructions based on fossil-pollen data (gray, Bartlein et al. 2011) and simulations (at the locations of the data) for the difference in mean annual precipitation (MAP) for the mid-Holocene (relative to present) in northern Africa (20°W-30°E; 5-30°N). The comparison shows that although all models simulated wetter-than-present conditions in northern Africa for the mid-Holocene, they systematically underestimated the magnitude of the precipitation difference.

For PMIP4-CMIP6, we have modified the experimental design of the *midHolocene* and *lgm* experiments with the aim of obtaining more realistic representations of these climates. One of these modifications is the inclusion of changes in atmospheric dust loading, which can have a large effect on regional climate changes. Dust has now been implemented in many CMIP6 models, either by using models with an interactive representation of dust or by prescribing atmospheric dust content. In PMIP3, the *midHolocene* $CO_2$ concentration was prescribed to be the same as in the pre-industrial control simulation, because the focus was on testing the impact of the insolation forcing on meridional climate gradients and seasonality. Realistic values of the $CO_2$ concentration and other trace gases will be used in PMIP4-CMIP6 (Table 2). This will allow the *midHolocene* experiment to be used as the initial state for transient simulations of the late Holocene planned as part of PMIP4, and ensure consistency of forcing between the *midHolocene* PMIP4-CMIP6 snapshot experiment and the transient simulations (Otto-Bliesner et al., 2016). The PMIP3 LGM experiments considered a single ice sheet reconstruction (Abe-Ouchi et al., 2015). However, there is uncertainty about the geometry of the ice sheets at the Last Glacial Maximum. Thus the protocol for the PMIP4-CMIP6 *lgm* simulations includes a choice between the old PMIP3 ice sheet (Abe-

Ouchi et al., 2015) or one of two new 21ky BP reconstructions based on somewhat different approaches: ICE-6G_C (Argus et al., 2014; Peltier et al., 2015) and GLAC-1D (Tarasov et al., 2012; Briggs et al., 2014, Ivanovic et al., 2016). Groups wishing to use the *lgm* equilibrium experiment to initialise PMIP4 transient simulations of the last deglaciation (Ivanovic et al., 2016) must use either ICE-6G_C or GLAC-1D because these are consistent with the ice sheet and meltwater forcings provided for the transient experiments. The impact of these different ice-sheet forcings will be a focus for sensitivity experiments in PMIP4 (Kageyama et al., 2017). There are uncertainties regarding other boundary conditions for the *midHolocene* and *lgm* experiments, including dust and vegetation (section 3.5), and these will also be investigated as part of the analysis of the entry-card simulations.

**2.2 The last millennium (*past1000*)**

The millennium prior to the industrial era, 850-1849 CE, provides a well-documented (e.g. PAGES2k-PMIP3 group, 2015) period of multi-decadal to multi-centennial changes in climate, with contrasting periods such as the Medieval Climate Anomaly and the Little Ice Age. This interval was characterised by variations in solar, volcanic and orbital forcings (Figure 1), which acted under climatic background conditions not too different from today. This interval provides a context for earlier anthropogenic impacts (e.g. land-use changes) and the current warming due to increased atmospheric greenhouse gas concentrations. It also helps constrain the uncertainty in the future climate response to a sustained anthropogenic forcing.

The PMIP3-CMIP5 *past1000* simulation provided an assessment of climate variability on decadal and longer scales and information on predictability under forced and unforced conditions. The importance of forced variability on multi-decadal to centennial time scales was highlighted by comparing spectra from *past1000* simulations with those from control experiments (Fernández-Donado et al., 2013). Other studies focused on the temperature difference between the warmest and coldest centennial or multi-centennial periods and the relation to changes in external forcing, in particular variations in solar irradiance (Fernández-Donado et al., 2013; Hind and Moberg, 2013). Single-model ensembles have provided improved understanding of the importance of internal versus forced variability and the individual forcings when compared to reconstructions at both global and regional scales (Man et al., 2012; Phipps et al., 2013; Schurer et al., 2014; Man et al., 2014; Man and Zhou, 2014; Otto-Bliesner et al., 2016). The *past1000* simulations show relatively good agreement with regional climate reconstructions for the northern hemisphere, but less agreement with southern hemisphere records. The simulations exhibit more regional coherence than shown by southern hemisphere records, though it is not clear whether this is due to deficiencies in the southern hemisphere records, poor representation of internal variability and/or an overestimation of the forced response in the simulations.

The PMIP4-CMIP6 *past1000* simulations build on the DECK experiments, in particular the pre-industrial control (*piControl*) simulation as an unforced reference, and the *historical* simulations (Eyring et al., 2015). Moreover, the *past1000* simulations provide initial conditions for *historical* simulations starting in the 19th century that are considered superior to the *piControl* state, as they include integrated information from the forcing history (e.g. large volcanic eruptions in the early 19th century). It is therefore mandatory to continue the *past1000* simulation into the historical period (Jungclaus et al, 2016). The PMIP4-CMIP6 *past1000* simulation will use a new, more comprehensive reconstruction of volcanic forcing (Sigl et al., 2015) and an experimental

protocol that ensures a more continuous transition from the pre-industrial past to the future. The final choices result from strong interactions with the groups producing the different forcing fields for the historical simulations (Jungclaus et al, 2016). Higher-resolution simulations will allow the analysis of a greater range of regional processes, such as the role of storm-tracks and blocking on regional precipitation.

## 2.3 The last interglacial (*lig127k*)

The Last Interglacial (ca 130-115 kyr BP) was characterised by a northern hemisphere insolation seasonal cycle even larger than for the mid-Holocene (Figure 1, Table 1). This resulted in a strong amplification of high-latitude temperatures and reduced Arctic sea ice. Global sea level was at least 5 m higher than now for at least several
thousand years (Masson-Delmotte et al., 2013; Dutton et al., 2015). Both the Greenland and Antarctic ice sheets contributed to this sea level rise, making it an important period for testing our knowledge of climate-ice sheet interactions in warm climates. The availability of quantitative climate reconstructions for the Last Interglacial (e.g. Capron et al., 2014) makes it feasible to evaluate these simulations and assess regional climate changes.

Climate model simulations of the Last Interglacial, reviewed and assessed in the AR5 (Masson-Delmotte et al., 2013), varied in their forcings and were not necessarily made with the same model or at the same resolution as the CMIP5 future projections. There are large differences between simulated and reconstructed mean annual surface temperature anomalies compared to present, particularly for Greenland and the Southern Ocean, and in the temperature trends in transient experiments run for the whole interglacial (Bakker et al., 2013; Lunt et al.,
2013). Part of this discrepancy stems from the fact that the climate reconstructions were of the local maximum interglacial warming, and this was not globally synchronous, an issue which is addressed in the PMIP4-CMIP6 protocol.

The PMIP4-CMIP6 *lig127k* experiment will help to determine the interplay of warmer atmospheric and oceanic
temperatures, changed precipitation, and changed surface mass and energy balance on ice sheet thermodynamics and dynamics (Table 1). The major changes in the experimental protocol for *lig127k*, compared to the pre-industrial DECK experiment, are changes in the astronomical parameters and greenhouse gas concentrations (Table 2; Otto-Bliesner et al., 2016). Meaningful analyses of these simulations are now possible because of the concerted effort to synchronise the chronologies of individual records and thus provide a spatial-temporal picture
of last interglacial temperature change (Capron et al., 2014, 2016), and also to document the timing of the contributions of Greenland and Antarctica to the global sea level (Winsor et al., 2012; Steig et al., 2015). Regional responses of tropical hydroclimate and of polar sea ice can be assessed and compared to the *mid-Holocene*. Outputs from the *lig127k* experiment will be used by ISMIP6 to force stand-alone ice sheet experiments (*lastIntergacialforcedism*) in order to quantify the potential sea level change associated with this
climate. The *lig127k* experiment will also be the starting point for a transient experiment covering the interglacial to be run within PMIP4.

## 2.4 The mid-Pliocene Warm Period (*midPliocene-eoi400*)

The mid-Pliocene Warm Period (mPWP, ca. 3.3 to 3 million years ago) was the last time in Earth history when atmospheric $CO_2$ concentrations approached current values (~400 ppmv) with a continental configuration similar

to today (Figure 1, Table 1). Vegetation reconstructions (Salzmann et al., 2008) indicate that the area of deserts was smaller than today and boreal forests were present in high northern latitude regions which are covered by tundra today. Climate model simulations produce global mean surface air temperature anomalies ranging from +1.9 °C to +3.6 °C (relative to each model's pre-industrial control) and an enhanced hydrological cycle (Haywood et al., 2013) with strengthened monsoons (Zhang et al., 2013). These simulations also show that meridional temperature gradients were reduced (due to high latitude warming), which has significant implications for the stability of polar ice sheets and sea level in the future (e.g. Miller et al., 2012). Model–data comparisons provide high confidence that mean surface temperature was warmer than pre-industrial (Dowsett et al., 2012; Haywood et al., 2013; Masson-Delmotte et al., 2013). However, as is the case for the Last Interglacial, the mid-Pliocene simulations were not always derived from the same model at the same resolution as the CMIP5 future projections.

The PMIP4-CMIP6 *midPliocene-eoi400* experiment is designed to understand the long-term response of the climate system to a near modern concentration of atmospheric $CO_2$ (longer term climate sensitivity or Earth System Sensitivity). It will also be used to address the response of ocean circulation, Arctic sea-ice and modes of climate variability (e.g. El Niño Southern Oscillation), as well as the global response in the hydrological cycle and regional changes in monsoon systems (Table 1). The simulation has the potential to to be informative about required emission reduction scenarios designed to prevent an increase in global annual mean temperatures by more than 2 °C after 2100 CE. Boundary conditions include modifications to global ice distributions, topography/bathymetry, vegetation and $CO_2$ (Table 2, Section 3) and are provided by the US Geological Survey Pliocene Research and Synoptic Mapping Project (PRISM4: Dowsett et al., 2016).

## 3. Experimental set up and model configuration

The forcings and boundary conditions for each PMIP4-CMIP6 paleoclimate simulation are summarised in Table 2. The complete justification of the experimental protocols and analysis plans are given in a series of companion papers: Otto-Bliesner et al. (2016) for the *midHolocene* and *lig127ka* experiments, Kageyama et al. (2017) for the *lgm*, Jungclaus et al. (2016) for the *past1000* and Haywood et al. (2016) for the *midPliocene-eoi400* experiment. These papers also explain how the boundary conditions for each period have been designed and constitute key references for the experimental protocol for each of the PMIP4-CMIP6 simulations. Here we provide guidelines that are common to all of the experiments, focusing on the implementation of the boundary conditions where there is a need to ensure consistency between CMIP6 and PMIP4 experiments.

### 3.1 Model version and set-up

The climate models taking part in CMIP6 are very diverse: some represent solely the physics of the climate system, some include the carbon cycle and other biogeochemical cycles, and some include interactive natural vegetation and/or interactive dust cycle/aerosols. It is mandatory that the model version used for the PMIP4-CMIP6 experiments is exactly the same as for the other CMIP6 experiments, in particular the DECK and *historical* simulations. Except for the *past1000* simulation, all the other PMIP4-CMIP6 simulations are equilibrium experiments, in which the boundary conditions and forcings are constant from one year to another. The experimental set-up for each simulation is based on the DECK pre-industrial experiment (Eyring et al.,

2015); the forcings and boundary conditions for the DECK pre-industrial experiments are modified to obtain the forcings and boundary conditions necessary for each PMIP4-CMIP6 paleoclimate experiment (Table 2). No additional interactive component (such as vegetation or dust) should be included in the model unless it is already included in the DECK version. Such changes would affect the global energetics (Braconnot and Kageyama, 2015) and therefore prevent rigorous analyses integrating across multiple time periods or between MIPs (sections 4.2 and 4.3).

Table 2: summary of changes in boundary conditions w.r.t. *piControl* for each PMIP4-CMIP6 experiment

For each experiment, the greenhouse gases and astronomical parameters should be modified from the DECK *piControl* experiment according to Table 2. In the following sections, we give more detail on the implementation of the boundary conditions that require specific attention to ensure consistency within CMIP6 and PMIP4.

### 3.2 Implementation of ice sheets

The *midPliocene-eoi400* and *lgm* experiments require changes in ice sheets. This implies changes in topography, land surface type (adding or removing land ice), sea level and hence land-sea mask and ocean bathymetry (Figure 3). These changes in boundary conditions should be implemented as follows:

1. The land-sea distribution should be implemented in the ocean and atmosphere/land surface models. This step is optional for the *midPliocene-eoi400* experiment, but mandatory for the *lgm*. It is important to check the newly glaciated areas in the *lgm* experiment, to ensure that grid cells under the grounded ice sheets (e.g. in the Hudson Bay area and over present-day Barents-Kara Seas) are not specified as ocean cells.

2. The ice sheet extent should be implemented in the atmosphere/land surface model.

3. Changes in topography should be implemented by adding the topographic anomaly provided by PMIP4 web site (http://pmip4.lsce.ipsl.fr) for the LGM or the PlioMIP web site (http://geology.er.usgs.gov/egpsc/prism/7_pliomip2.html) for the mPWP to the topography used for the *piControl* simulation. This may require re-computing parameters based on topography, such as those used in gravity wave drag parameterisations, because of the difference in surface roughness between ice sheets and non-glaciated terrain.

Steps 2 and 3 are compulsory for both the *lgm* and *midPliocene-eoi400* experiments.

4. If feasible, changes in ocean bathymetry should be implemented, by using the more detailed bathymetry provided with the ice-sheet reconstructions. For the *midPliocene-eoi400* experiment the alternative is to leave bathymetry unchanged (i.e. the same as in the *piControl*). The alternative for the *lgm* experiment is to lower mean sea level by the amount consistent with the ice-sheet reconstruction used. If the ocean model includes a parameterization of the impact of tides on ocean circulation, re-computation of the parameters as a function of the new bathymetry and ocean boundaries is recommended.

**Figure 3: Changes in boundary conditions related to changes in ice sheets for the *midPliocene-eoi400* (top) and *lgm* (middle: ICE-6G_C and bottom: GLAC-1D) experiments. Coastlines for paleo-period shown as brown contours. Ice**

Some ice-sheet related changes must also be implemented in the initial conditions:

- The atmospheric mass must be the same as today. For some models, this means that the initial surface pressure field has to be adjusted to the change in surface elevation.
- At the beginning of the *lgm* simulation, the mean ocean salinity has to be increased by +1 PSU everywhere, to account for the lowering of sea level. Alkalinity also needs to be adjusted accordingly if an ocean biogeochemistry model is used.

## 3.3 River run-off

When the land-sea distribution is modified, river pathways and basins must also be adjusted so that fresh water is conserved at the Earth's surface and rivers reach the ocean. This is particularly important for the *lgm* run, given the large lowering of sea level (cf. Alkama et al., 2008). River routing files will be provided for the *lgm* on the PMIP web site (http://pmip4.lsce.ipsl.fr), which indicate how to change the course of rivers in regions covered by ice sheets. For the *midPliocene-eoi400* experiment, rivers pathways remain unchanged from modern except on new land grid cells, when rivers should be routed to the nearest ocean grid box or most appropriate river outflow point. For all other periods, river pathways should be kept unchanged. This is a conservative choice due to the lack of global paleo data set describing these changes.

## 3.4 Vegetation and land use

Paleoenvironmental records show that natural vegetation patterns during each of the PMIP4-CMIP6 period were different from today. However, in order to ensure comparability between past, present and future climate simulations, the PMIP4-CMIP6 paleoclimate simulations should follow the same protocol as the DECK and historical simulations. If the DECK and historical simulations use dynamic vegetation, then the PMIP4-CMIP6 paleoclimate simulations should also. If the DECK and historical simulations use prescribed vegetation, then the same vegetation should be prescribed in the PMIP4-CMIP6 paleoclimate simulations. One exception to this is the *midPliocene-eoi400* experiment, where models which prescribe vegetation in the DECK and historical simulations should prescribe the mid-Pliocene vegetation (Haywood et al., 2016). The other exception is for models including interactive dust cycle for the LGM, which should impose vegetation which allows dust emissions over LGM dust emission regions. Simulations to examine the impact of vegetation changes during other periods are of interest and could be evaluated using paleoclimate data. These could be made using prescribed vegetation changes, by running a model such as BIOME4 (https://pmip2.lsce.ipsl.fr/) off line to compute vegetation patterns compatible with a past climate state, or by running additional simulations with a non-standard version of the model with dynamic vegetation. Sensitivity experiments such as these will be encouraged within PMIP4 but are not part of the PMIP4-CMIP6 experiments.

For the *past1000* simulation, land-use changes have to be implemented in the same manner as for the *historical* simulation, using the land-use forcing provided by the Land Use Model Intercomparison Project (Lawrence et al., 2016) and the CMIP6 Land Use Harmonization dataset (https://cmip.ucar.edu/lumip; Hurtt et al., in prep.;

Jungclaus et al., 2016). This data set is derived from the HYDE3.2 (Klein Goldewijk, 2016) estimates of the area of cropland, managed pasture, rangeland, urban, and irrigated land. Different crop types are treated separately and estimates of wood harvest are also provided.

### 3.5 Natural aerosols

### 3.5.1 Mineral Dust

Natural aerosols show large variations on glacial-interglacial time scales, with glacial climates having higher dust loadings than interglacial climates (Kohfeld and Harrison, 2001; Maher et al., 2010). Dust emissions from northern Africa were significantly reduced during the MH (McGee et al., 2013). As is the case with vegetation, the treatment of dust in the *midHolocene, lig127k* and *lgm* simulations should parallel the treatment in the *piControl*. However, some of the models in CMIP6 include representations of interactive dust. For those models, maps of soil erodibility that account for changes in the extension of possible dust sources, will be provided for the *midHolocene, lig127k* and *lgm* experiments. The maps are the same for the interglacial experiments. Dust anomalies/ratios compared to the pre-industrial background should be used, for consistency with the DECK *piControl* simulation. As there have been instances of runaway climate-vegetation-dust feedback, leading to unrealistically cold LGM climates (Hopcroft and Valdes, 2015a), it is advisable to test model behaviour with an atmosphere-only model before running the entire *lgm* simulation.

To allow experiments with prescribed dust changes, three-dimensional monthly climatologies of dust atmospheric mass concentrations will be provided for the *piControl*, *midHolocene*, and *lgm*. These are based on two different models (Albani et al., 2014, 2015, 2016; Hopcroft et al., 2015, Figure 4) and modelling groups are free to choose between these data sets. Additional dust-related fields (dust emission flux, dust load, dust aerosol optical thickness, short- and long-wave, surface and top of the atmosphere dust radiative forcing) are also available from these simulations. Implementation should follow the same procedure as for the *historical* run. The implementation for *lig127k* experiment should use the same data set as for the *midHolocene* one. Since dust plays an important role in ocean biogeochemistry (e.g. Kohfeld et al, 2005), three dust maps are provided for the *lgm* run. Two of these are consistent with the climatologies of dust atmospheric mass concentrations; the other is primarily derived from paleoenvironmental observations (Lambert et al., 2015, Figure 4). The modelling groups should use consistent data sets for the atmosphere and the ocean biogeochemistry. The Lambert et al. (2015) data set can therefore be used for models which cannot include the changes in atmospheric dust according to the other two data sets.

**Figure 4: Maps of dust deposition (g m-2 a-1) simulated with the Community Earth System Model for a. PI (Albani et al., 2016), b. MH (Albani et al., 2015), and c. LGM (Albani et al., 2014). Maps of dust deposition (g m-2 a-1) for the LGM d. simulated with the Hadley Centre Global Environment Model 2-Atmosphere (Hopcroft et al, 2015), and reconstructed from a global interpolation of paleodust data (Lambert et al., 2015).**

### 3.5.2 Volcanoes and stratospheric aerosols

The *past1000* experiment includes changes in volcanic aerosols. Changes in volcanic aerosols are not included in the other PMIP4-CMIP6 experiments, where the pre-industrial forcings (if any) should be used. Estimates of sulphur injections, derived from a recent compilation of synchronized Antarctic and Arctic ice core records, provide an improved history of the timing and magnitude of eruptions over the last 2500 years (Sigl et al., 2013).

Ice core sulphate fluxes are translated into a time series of stratospheric sulphur injection via linear scaling (following Gao et al., 2008) and by matching the ice-core signals to historically confirmed eruptions. Unidentified eruptions are assigned as tropical when there are matching northern and southern hemisphere signals. Eruptions with signals only registered in the northern or southern hemisphere are considered to be extratropical in origin. Modelling groups using interactive aerosol modules and sulphur injections in their historical simulations should follow the same method for the *past1000* experiment and use sulphur injection estimates directly. For the other models, estimates of aerosol radiative properties as a function of latitude, height, and wavelength should be calculated using the Easy Volcanic Aerosol (EVA) module (Toohey et al., 2016). This parameterized three-box model of stratospheric transport uses simple scaling relationships to derive mid-visible aerosol optical depth (AOD) and aerosol effective radius ($r_{eff}$) from stratospheric sulphate mass. EVA uses model-specific information (grid, wave-length distribution) to produce annual volcanic aerosol forcing files for wavelength dependent aerosol extinction (EXT), single scattering albedo (SSA) and scattering asymmetry factor (ASY) as function of time, latitude, height and wave length. There are uncertainties associated with this approach. Additional sensitivity studies allowing the assessment of the impacts of these uncertainties on the *past1000* simulations will be made as part of the PMIP4 past1000 Tier 2 experiments (see Jungclaus et al., 2016). The sulphur injection time series and the EVA software package are provided via the PMIP4 web page (https://pmip4.lsce.ipsl.fr/doku.php/exp_design:lm).

**3.6 Solar irradiance**

For the *past1000* experiment, new reconstructions of TSI and SSI are provided that are based on recent estimates of cosmogenic isotopes and improved irradiance models (see Jungclaus et al., 2016 for details). The forcing prescribed for the Tier 1 past1000 experiment (https://pmip4.lsce.ipsl.fr/doku.php/exp_design:lm) is constructed using a [14]C based reconstruction (Usoskin et al., 2016) of yearly sunspot numbers and an updated version of the Viera et al. (2011) irradiance model. To achieve a smooth transition to the industrial period for historical experiments (1850 − 2015 CE) that start from the end of the *past1000* simulations, the forcing is scaled to match the CMIP6 historical forcing (Matthes et al., 2016). Alternative forcing reconstructions, reflecting uncertainty in the cosmogenic isotopes and the methods used in solar irradiance models, are provided as a basis for additional Tier 2 experiments (Jungclaus et al., 2016).

**3.6 Spin-up and duration of experiments**

The data stored in the CMIP6 database should be representative of the equilibrium climates of the MH, LGM, LIG and mPWP periods, and of the transient evolution of climate between 850-1849 CE for the *past1000* simulations. Spin-up procedures will differ according to the model and type of simulation, but the spin up should be long enough to avoid significant drift in the analysed data. We recommend that the spin-up should be run until the trend in global mean sea-surface temperature is <0.05 K per century and the Atlantic Meridional Overturning Circulation (AMOC) is stable. A parallel requirement for carbon-cycle models and/or models with dynamic vegetation is that the 100-year average global carbon uptake or release by the biosphere is <0.01 Pg C $a^{-1}$. Initial conditions for the spin-up can be taken from an existing simulation. A minimum of 100 years output is required for the equilibrium simulations but, given the increasing interest in analysing multi-decadal

variability (e.g. Wittenberg, 2009), modelling groups are encouraged to provide outputs for 500 years or more if possible.

**3.7 Documentation**

Detailed documentation of the PMIP4-CMIP6 simulations is required. This should include:

- a description of the model and its components;
- information about the boundary conditions used, particularly when alternatives are allowed (Table 2);
- information on the implementation of boundary conditions and forcings. Figures showing the land-sea mask, land-ice mask, and topography as implemented in a given model are useful for the *lgm* and *midPliocene-eoi400* experiments, while figures showing insolation are particularly important for the

 *midHolocene* and *lig127k* experiments. Check lists for the implementation of simulations are provided in the PMIP4 papers which give detailed information for each experiment (*midHolocene*: Otto-Bliesner et al., 2016; *lgm*: Kageyama et al., 2017; *past1000*: Jungclaus et al., 2016; *lig127k*: Otto-Bliesner et al., 2016; *midPliocene-eoi400*: Haywood et al., 2016);
- information about the initial conditions and spin-up technique used. A measure of the changes in key

 variables (e.g. globally averaged 2 m surface air temperatures, sea-surface temperatures, bottom ocean temperatures, top-of-the-atmosphere radiative fluxes, soil carbon storage) should be provided in order to assess remaining drift.

Documentation should be provided via the ESDOC website and tools provided by CMIP6 (http://es-doc.org/) to facilitate communication with other CMIP6 MIPs. This documentation should also be provided on the PMIP4

website to facilitate linkages with non-CMIP6 simulations to be carried out in PMIP4. A PMIP4 special issue, shared between *Geoscientific Model Development* and *Climate of the Past*, will provide a further opportunity for modelling groups to document specific aspects of their simulations.

**4. Plan of Analyses**

The community using PMIP simulations is very broad, from climate modellers and palaeoclimatologists to

biologists studying recent changes in biodiversity and archaeologists studying potential impacts of past climate changes on human populations. Because of this, we do not aim to give a comprehensive plan of PMIP analyses, but focus instead on topics closely related to the CMIP6 key questions. We first present examples of paleoclimate reconstructions available for comparison to the PMIP4-CMIP6 simulations (Section 4.1). We then outline topics of analysis for specific periods and the full PMIP4-CMIP6 ensemble of simulations (Section 4.2).

Links established with other CMIP6 MIPs (Section 4.3) will make it possible to capitalise on their analyses to improve understanding of specific aspects of past climates and vice versa. Finally, in Section 4.4, we discuss the implications of this analysis plan in terms of requested output.

**4.1 Paleoclimatic and paleoenvironmental reconstructions, model-data comparisons.**

Past environmental and climatic changes are typically documented at specific sites, whether on land, in ocean sediments or in corals, or from ice cores. The evaluation of climate simulations requires these paleoclimatic and paleoenvironmental data to be synthesised for specific time periods. A major challenge in building such

syntheses is to synchronise the chronologies of the different records. There are many syntheses of information on past climates and environments and, although Table 3 lists some of the sources of quantitative reconstructions for the PMIP4-CMIP6 time periods, it is not our goal here to provide an extensive review of these resources. Much of this information stems from the impact of climatic changes on the environment, such as on fires, dust, marine microfauna and vegetation (which can be documented by pollen records, or for the recent period, by tree rings). Past climatic information is also contained in isotopic ratios of oxygen and carbon, which can be found in ice sheets, speleothems, or in the shells of marine organisms. Furthermore, the ocean circulation can also be documented by geochemical tracers in marine sediments from the sea floor (e.g. $\Delta^{14}$C, $\delta^{13}$C, $^{231}$Pa/$^{230}$Th, $\varepsilon_{Nd}$). The fact that these physical, chemical or biological indicators are indirect records of the state of the climate system and can also be sensitive to other factors (such as atmospheric $CO_2$ concentrations for vegetation) has to be taken into account in model-data comparisons. Comparisons with climate model output can therefore be performed from different points of view: either the climate model output can be directly compared to reconstructions of past climate variables, or the response of the climatic indicator itself can be simulated from climate model output and compared to the climate indicator. Such "forward" models include dynamical vegetation models, tree ring models, or models computing the growth of foraminifera (cf. Section 4.4). Some paleoclimatic indicators such as meteoric water isotopes have to be computed as the climate model is running, but are also examples of this forward modelling approach. Modelling the impacts of past climate changes on the environment is key to understand how climatic signals are transmitted to past climate records. It also provides an opportunity to test the types of models that are used in the assessment of the impacts of future climate changes on the environment.

| Table 3: examples of data syntheses for the PMIP4-CMIP6 periods |
|---|

Reconstructing paleoclimates and paleoenvironments, as well as building new syntheses of these reconstructions, are very active areas of research. We expect new data sets to become available, which will increase the number of possible model-data comparisons for the PMIP4-CMIP6 periods.

## 4.2 Overview of analysis plan

Each PMIP4-CMIP6 period has been selected for specific reasons (Table 1). The initial analyses for each period will focus on these purposes. Sensitivity experiments have also been designed, as part of the PMIP4 project, to analyse the PMIP4-CMIP6 simulations in more detail (see Jungclaus et al., 2016; Otto-Bliesner et al., 2016; Kageyama et al., 2017; Haywood et al., 2016 for more details about these). Here, we list several topics of analyses which are important for single periods as well as for the full PMIP4-CMIP6 ensemble.

### 4.2.1 Role of forcings and feedbacks

Quantifying the role of forcings and feedbacks in creating climates different from today has been a focus of PMIP for many years. Compared to the PMIP3-CMIP5 models, many CMIP6 models will include new processes, such as dust, or improved representations of major radiative feedback processes, such as clouds. Improvements to the design of the *past1000*, *midHolocene* and *lgm* experiments, such as realistic atmospheric greenhouse gas concentration for the *midHolocene* experiment, improved volcanic and solar forcings for *past1000*, and the inclusion of dust forcing or feedback for the *lgm* should have a noticeable impact on regional

climates. We will evaluate the impact of these changes on the PMIP4-CMIP6 climates at global, large-scale (e.g. polar amplification, land-sea contrast) as well as regional scales, together with the mechanisms explaining these impacts.

All the PMIP4-CMIP6 experiments will be made with the same model version, facilitating analyses across the five time periods to examine potential relationships between forcings of different nature and amplitude and the climate responses, and compare the processes involved in these responses (e.g. Izumi et al., 2013). Multi-period analyses are useful to understand the relationship between background climate state and the nature and strength of specific feedbacks. For example, there are temperature thresholds that determine whether snow and ice can be present, and temperature thresholds also play a part in determining the distribution of specific vegetation types. Thus, a given change in climate could have different effects on snow/ice or vegetation feedback depending on the base climate state. Density thresholds also play a part in controlling the oceanic overturning circulation, again leading to the possibility that ocean changes may be modulated by background state. Multi-period analyses provide a way of determining whether systematic model biases affect the overall response and the strength of feedbacks independent of climate state. They will allow us to determine, for example, whether the persistent failure to reproduce the observed magnitude of change in monsoon precipitation and the relatively small impact of vegetation feedback during the MH is related to biases or base climate. Similarly, they will help to quantify whether simulated changes in ocean circulation at the LGM are affected by systematic model biases or threshold behavior. Model-data comparisons (cf. Section 4.2.2) will be used to assess the realism of the simulated climate change and to detect key mechanisms affecting model behaviour independently of the base climate state. One challenge will be to develop new approaches to analyse the PMIP4-CMIP6 ensemble so as to separate the impacts of model resolution, content, or complexity on the simulated climate.

Ice sheets represent strong changes in radiative forcing, as well as a direct forcing on atmosphere circulation. The PMIP4-CMIP6 ensemble will allow new analyses of the impact of smaller (mPWP) or larger (LGM) ice sheets. The ocean and sea-ice feedbacks will also be analysed. The representation of sea ice and Southern Ocean proved to be problematic in previous simulations of colder (LGM, Roche et al., 2012) and warmer climates (LIG, Bakker et al., 2013, Lunt et al., 2013). For the LGM, there is evidence of a shallower, and yet active overturning circulation in the North Atlantic (e.g. Lynch-Stieglitz et al., 2007, Böhm et al., 2015). Understanding this oceanic circulation as well as its links to surface climate is a topic of high importance since the Atlantic Meridional Overturning Circulation could modulate future climate changes at least in regions around the North Atlantic (IPCC 2013).

The PMIP4-CMIP6 protocol seeks to address uncertainties in the forcings by providing alternative data sets. This approach was already used in the PMIP3 *past1000* experiments (Schmidt et al., 2010, 2011) but will now be applied to the *lgm* experiment, for which we provide significantly different ice sheet reconstructions and dust forcings (for models which do not include interactive dust). Both the ice sheets and dust could result in climate differences at the regional scale at least, and at larger scales if, for instance, they have an impact on the ocean circulation. Overall, allowing for different forcings for PMIP4-CMIP6 experiments is a new topic in PMIP and should lead to an improved sampling of modelled climates through better sampling of the possible forcings.

**4.2.2 Benchmarking the PMIP4-CMIP6 simulations**

The compatibility of past, historical and future climate simulations, through the use of seamless forcings and identical model versions, will allow benchmarking based on syntheses of paleoenvironmental data and paleoclimate reconstructions (Section 4.1) to be applied to models used for future projections. We will make full use of the fact that modelling groups must also run the *piControl* and *historical* experiments. Indeed, the *piControl* and *historical* simulations provide two alternative reference states for paleoclimate simulations. Existing paleoclimate reconstructions have used different modern reference states, and this has been shown to have an impact on the magnitude of reconstructed changes (e.g. Hessler et al., 2014). Comparisons of the simulated *piControl* and the *historical* climates will provide a way of quantifying this source of reconstruction uncertainty, as will comparisons with present-day observations and reanalysis data sets (Obs4MIPS, Ferraro et al, 2015).

Systematic benchmarking of each of the PMIP4-CMIP6 simulations will be a major aspect of the planned multi-period approach. This will benefit from the existing paleoclimatic data sets (Section 4.1, Table 3) and from the development of new data syntheses, assessments of the regional-scale consistency of different sources of information, as well as the use of forward modelling to quantify uncertainties in the climate reconstructions. Large-scale features, such as polar amplification, land-sea contrast, and precipitation changes scaling to temperature changes, as well as more regional features such as the monsoons and mid-continental climates, and climate responses over specific data-rich areas (such as Europe or North America) are a prime target for this benchmarking. The ensemble of metrics developed for the PMIP3-CMIP5 *midHolocene* and *lgm* simulations (e.g. Harrison et al. 2014) will be expanded to include more process-oriented metrics. Benchmarking results from the PMIP4- *midHolocene* and *lgm* "entry card" simulations will be compared to benchmark metrics from previous generations of PMIP to provide a rigorous assessment of model improvements since the last phase of CMIP.

There are many aspects of the climate system which are difficult to measure directly, and which are therefore difficult to evaluate using traditional methods. The "emergent constraint" (e.g. Sherwood et al., 2014) concept, which is based on identifying a relationship to a more easily measurable variable, has been successfully used by the carbon-cycle and modern climate communities and holds great potential for the analysis of paleoclimate simulations. This could be particularly valuable to examine the realism of e.g. cloud feedbacks in the simulations or the contribution of seasonal climate changes to hydrological budgets. Using multiple time periods to examine emergent constraints ensures that they are robust across climate states.

**4.2.3 Relating past and future climate changes**

Attempts to constrain climate sensitivity using information about the LGM period have been hampered by the fact that there were too few *lgm* experiments to draw statistically-robust conclusions (Hargreaves et al., 2012; Harrison et al., 2014; Hopcroft and Valdes, 2015b). These attempts also ignored uncertainties in forcings and boundary conditions. PMIP4-CMIP6 is expected to result in a much larger ensemble of *lgm* experiments, including simulations examining the impact of forcing and boundary condition uncertainties, and thus to allow advances to be made towards constraining climate sensitivity.

The issue of climate sensitivity (*sensu stricto*) and earth-system sensitivity (PALEOSENS Project Members, 2012) will also be examined through joint analysis of multiple paleoclimate simulations and climate reconstructions from different archives. Our analyses will capitalise on the DECK *piControl* and *abrupt4xCO2* experiments. The relationship between radiative forcing and global temperature is not straightforward (Crucifix,

2006; Yoshimori et al., 2011), partly because the nature of the forcing that drives the Earth into different climate states preferentially triggers short wave or long wave radiative responses, that have different impacts on the energy or water exchanges, on the feedbacks between different climate system components, or have different large- or regional- scale patterns. Nevertheless, estimates of climate sensitivity based on past climate states provide a starting point in establishing the bounds of the sensitivity of the climate system to a doubling of the

$CO_2$ concentration (Hargreaves, 2012). Furthermore, analyses of land-sea contrast or polar amplification have highlighted many similarities between past climate and future climate projections (Izumi et al., 2013; Masson et al., 2013; Izumi et al., 2015). Also, similar feedbacks occur in different climates for particular seasons, for which it is possible to isolate specific model behaviours (Braconnot and Kageyama, 2015) and thereby assess model credibility. The multi-period approach will thus bring new constraints to this analysis by providing further

insight on possible analogies between climate feedbacks operating under different external forcings, better understanding on the relationship between patterns and time scales of external forcings and patterns and timing of the climate responses, as well as improved model-data comparisons to link regional climate reconstructions to the Earth's global energetic and climate sensitivity. Additional constraints can be obtained by using perturbed-physics experiments, in which different versions of the same model are run using different values of key

parameters (Annan et al., 2005: Yoshimori et al., 2011). The 'perturbed forcing' approach (Bounceur et al., 2015; Araya-Melo, 2015), using sensitivity experiments carried out in PMIP4, could provide a way to chart the sensitivity of the climate system in a multi-dimensional space of forcing conditions.

### 4.2.4 Changes in mean climate vs. changes in climate variability

Multi-period analyses will also be useful for understanding the relationship between mean climate state and modes of natural variability (e.g. Liu et al., 2014; Saint-Lu et al., 2015). Future changes in modes of climate variability, such as El Niño Southern Oscillation (ENSO), are poorly constrained (Christiansen et al., 2013) because model projections are insufficiently long to provide robust statistics for low frequency (multi-decadal and centennial) variations. Robust statistics of ENSO changes have been derived through analysis of high-

resolution paleo-records (Emile-Geay et al., 2016). The equilibrium paleoclimate experiments in PMIP4-CMIP6 will provide an opportunity to sample simulations for long enough to obtain robust estimates of ENSO changes (Stevenson et al., 2010) and analyses of multiple long simulations with different forcings should provide a better understanding of changes in ENSO behaviour (Zheng et al., 2008; An et al., 2014) and help determine whether state-of-the-art climate models underestimate low frequency noise (Laepple and Huybers, 2014). The PMIP

Paleovariability Working Group will develop diagnostics for climate variability (Philips et al., 2014) to be applied to all the PMIP4-CMIP6 simulations. Analyses will focus on how models reproduce the relationship between changes in seasonality and interannual variability (Emile-Geay et al., 2016), the diversity of El-Niño events (Capotondi et al., 2015; Karamperidou et al., 2015; Luan et al., 2015), and the stability of teleconnections within the climate system (e.g. Gallant et al., 2013; Batehup et al., 2015).

For shorter time scales, the *past1000* simulations and corresponding high temporal resolution data are one of the only means to examine the mechanisms and realism of the relationships between events at the daily scale (e.g. weather extremes) and longer-term climatic changes.

**4.3 Interactions with other CMIP6 MIPs and the WCRP Grand Challenges**

Interactions between PMIP and other CMIP6 MIPs have mutual benefits: PMIP provides simulations of large climate changes that have occurred in the past and evaluation tools which capitalise on extensive data syntheses, while other MIPs will employ diagnostics and analyses which will be useful for analyzing the PMIP4 experiments. This is the case for AerChemMIP (Collins et al., 2016) for the aerosol forcings, SIMIP (Notz et al., 2016) and OMIP (Griffies et al., 2016) for the sea-ice and ocean components, LS3MIP (van den Hurk et al.,

2016) for the land surface, C4MIP (Jones et al., 2016) for the carbon cycle, ISMIP (Nowicki et al., 2016) for ice sheets, and CFMIP (Webb et al., 2016) for the cloud forcing and feedback analyses. The analytical tools developed in RFMIP (Pincus et al., 2016) will be useful for assessing the LGM GHG radiative forcing and those developed in VolMIP (Zanchettin et al., 2016) and LUMIP (Lawrence et al., 2016) will be relevant for the analyses of the impacts of volcanic and land use forcings in the *past1000* simulation. The *past1000* experiment

also offers a long time series perturbed by natural forcings and reconstructed land use changes for detection and attribution exercises and it is therefore relevant for DAMIP (Gillett et al., 2016). We have ensured that all the outputs necessary for the application of common diagnostics across PMIP and other CMIP6 MIPs will be available (see section 4.5).

PMIP has already developed strong links with several other CMIP6 MIPs (Table 4). CFMIP includes an idealized experiment mimicking the *lgm* simulation: *AMIPminus4K* is an atmosphere-only experiment in which the sea-surface temperatures are uniformly lowered by 4K (a mirror of the *AMIP4K* experiment in which sea-surface temperatures are increased by 4K). These experiments allow investigations of cloud feedbacks and associated circulation changes in a colder versus a warmer world and this will assist in disentangling the

processes at work in the *lgm* climate. Some MIPs have designed experiments based on PMIP data, including VolMIP for the study of the impact of large past volcanic eruptions and ISMIP6 for the impact of the last interglacial climate on the Greenland ice sheet. Links with CFMIP and ISMIP6 mean that PMIP will also contribute to the World Climate Research Programme (WCRP) Grand Challenges "Clouds, Circulation and Climate Sensitivity" and "Cryosphere and Sea Level" respectively. Furthermore, PMIP will provide input to the

WCRP Grand Challenge on "Regional Climate Information", through a focus on evaluating the mechanisms of regional climate change in the past, for example in the Arctic.

| Table 4: interactions of PMIP with other CMIP6 MIPs |
|---|

**4.4 Implications: required model output for the PMIP4-CMIP6 database**

The list of variables required to analyse the PMIP4-CMIP6 paleoclimate experiments (https://wiki.lsce.ipsl.fr/pmip3/doku.php/pmip3:wg:db:cmip6request) reflects plans for multi-time period analyses and for interactions with other CMIP6 MIPs. We have included pertinent variables from the data requests of other MIPs, including the CFMIP-specific diagnostics on cloud forcing, as well as land surface, snow, ocean, sea ice, aerosol, carbon cycle and ice sheet variables from LS3MIP, OMIP, SIMIP, AerChemMIP,

C4MIP, and ISMIP6 respectively. Some of these variables are also required to diagnose how climate signals are recorded by paleoclimatic sensors via models of e.g. tree growth (Li et al., 2014), vegetation dynamics (Prentice et al., 2011) or marine planktonic foraminifera (e.g. Lombard et al., 2011; Kageyama et al., 2013). The only set of variables defined specifically for PMIP are those describing oxygen isotopes in the climate system. Isotopes are widely used for paleoclimatic reconstruction and are explicitly simulated in several models. We have asked that average annual cycles of key variables are included in the PMIP4-CMIP6 data request for equilibrium simulations, as these proved exceptionally useful for analyses in PMIP3-CMIP5.

Variations in the shape of the Earth's orbit govern the latitudinal and seasonal distribution of insolation, and also produce variations in the lengths of individual "months" (where months are defined alternatively as either (a) the duration in days for the Earth to complete one-twelfth of its orbit (the "celestial" or "angular" calendar), or (b) a specific number of days, e.g. 31 days in January, 30 days in June (the "conventional" or "modern" calendar). When eccentricity is high, then the months around the time of year of perihelion are shorter, and those near aphelion are longer (Joussaume and Braconnot, 1997). For example, at 6 ka, perihelion occurs in August, aphelion in February and those months were approximately 1.5 days shorter and longer than at present, respectively (Fig. 5a). Variations in the lengths of months (or seasons) must therefore be taken into consideration when examining experiment minus control long-term mean differences, because the effect of the changing calendar on the calculation of long-term means can be as large as the potential differences among the means themselves (Joussaume and Braconnot, 1997; Pollard and Reusch, 2002; Timm et al., 2008; Chen et al., 2011). The size of the potential calendar effect or bias is illustrated in Fig. 5 (b-d), which shows the difference between present-day long-term means for October temperature and precipitation, and those calculated using the appropriate month lengths for 6 and 127 ka. Modifications to month length have not usually been taken into account in the model output post-treatment procedures (but see Harrison et al., 2014). The most straightforward way for dealing with the calendar effect is to save and use daily data for the calculation of monthly or seasonal means, and so we include those in the PMIP4-CMIP6 data request for some key variables. A second approach, less desirable, but probably adequate for our purposes, is to use a bias-correction approach, in particular, like that of Pollard and Reusch (2002), with the mean-preserving daily interpolation approach of Epstein (1991).

Daily values are also useful for running regional models. It is important to test the use of regional models for climate model projections at the regional scale. Regional models are also used to produce fine-scale palaeoclimate scenarios for use by the impact community, for example to study past climate impacts on biodiversity via ecological niche modelling. Paleoclimate indicators often respond to climate features not adequately captured with monthly data alone (such as growing season length). Daily weather variables are therefore required for some forward models, as well as to compute bioclimatic variables which are reconstructed e.g. based on pollen data (e.g. Bartlein et al., 2011).

## 5. Conclusions

PMIP4-CMIP6 simulations provide a framework to compare current and future anthropogenic climate change with past natural variations of the Earth's climate. PMIP4-CMIP6 is a unique opportunity to simulate past climates with exactly the same models as used for simulations of the future. This approach is only valid if the

model versions and implementation of boundary conditions are consistent for all periods, and if these boundary conditions are seamless for overlapping periods.

PMIP4-CMIP6 simulations are important in terms of model evaluation for climate states significantly different from the present and historical climates. We have chosen climatic periods well documented by paleoclimatic and paleoenvironmental records, with climate and environmental changes relevant for the study and projections of future climate changes: the millennium prior to the industrial epoch (*past1000*), 6,000 years ago (*midHolocene*), the last glacial maximum (*lgm*), the last interglacial (*lig127k*) and the mid-Pliocene (*midPliocene-eoi400*). .

The PMIP4-CMIP6 experiments will also constitute reference simulations for projects developed in the broader PMIP4. The corresponding sensitivity experiments, or additional experiments, are embedded in the PMIP4 project and are described in the companion papers to this overview (Haywood et al., 2016, Otto-Bliesner et al., 2016, Jungclaus et al., 2016, Kageyama et al., 2017). They are essential for a deeper understanding of the drivers of past climate changes for the PMIP4-CMIP6 climates or as initial conditions for transient simulations (e.g.
Ivanovic et al., 2016, for the last deglaciation, Otto-Bliesner et al., 2016 for the last interglacial and the Holocene), or for examining time periods from deeper with high atmospheric $CO_2$ concentrations (Lunt et al, 2016). Figure 6 summarises the position of the PMIP4-CMIP6 experiments with respect to the other PMIP4 experiments and projects on the right-hand-side. The left-hand-side shows how the PMIP4-CMIP6 experiments relate to the CMIP6 DECK and some other CMIP6 MIPs. PMIP4-CMIP6 experiments have been designed to be
analyzed by both communities.

**Figure 6: the PMIP4-CMIP6 experiments in the framework of CMIP6, with associated MIPs, and in the framework of PMIP4, with its working groups.**

The PMIP community anticipates major benefits from analysis techniques developed by the other CMIP6 MIPs, in particular in terms of learning about the processes of past climate changes in response to forcings (e.g. greenhouse gases, astronomical parameters, ice sheet and sea level changes) as well as the role of feedbacks (e.g. clouds, ocean, sea-ice). Collaborations have already been developed with e.g. CFMIP, ISMIP6 and VolMIP, and
the hope is to build additional collaborations with other CMIP6 MIPs. PMIP4-CMIP6 has the potential to be mutually beneficial for the paleoclimate and present/future climate scientists to learn about natural large climate changes and the mechanisms at work in the climate system for climates states as different from today as future climate is projected to be.

**Data availability**

All data mentioned in the present manuscript can be found on the following web sites:
- http://pmip4.lsce.ipsl.fr,
- http://geology.er.usgs.gov/egpsc/prism/7_pliomip2.html,
along with the web sites cited in Table 3, from which climatic reconstructions are available.

They will also be provided via the ESGF system, along with forcing files for other CMIP6 experiments, when the boundary conditions are approved.

**Acknowledgements**

MK and QZ acknowledge funding from the French-Swedish project GIWA. PB, JJ and SPH acknowledge funding from the JPI-Belmont project "PAleao-Constraints on Monsoon Evolution and Dynamics (PACMEDY)" through their respective national funding agencies. SPH also acknowledges funding from the Australian Research Council (DP1201100343) and from the European Research Council for "GC2.0: Unlocking the past for a clearer future". AMH and AMD acknowledge funding from the European Research Council under the European Union's Seventh Framework Programme (FP7/2007–2013)/ERC grant agreement no. 278636 and the EPSRC-supported Past Earth Network. RFI is funded by a NERC Independent Research Fellowship [#NE/K008536/1]. SJP's contribution is supported under the Australian Research Council's Special Research Initiative for the Antarctic Gateway Partnership (Project ID SR140300001). FL acknowledges support from CONICYT projects 15110009, 1151427, ACT1410, and NC120066.

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

TABLES

| Period | Purpose | CMIP6 Priority |
|---|---|---|
| **Last millennium** (*past1000*) 850-1849 CE | a) Evaluate the ability of models to capture reconstructed variability on multi-decadal and longer time-scales. b) Determine what fraction of the variability is attributable to "external" forcing and what fraction reflects purely internal variability. c) Provide a longer-term perspective for detection and attribution studies. | Tier 1* |
| **Mid-Holocene** (*midHolocene*) 6 kyr ago | a) Compare the model response to known orbital forcing changes and changes in greenhouse gas concentrations to paleodata, describing major temperature and hydrological changes. b) Relationships between changes in mean state and variability | Tier 1* PMIP4-CMIP6 entry card |
| **Last Glacial Maximum** (*lgm*) 21 kyr ago | a) Compare the model response to ice-age boundary conditions with paleodata. b) Attempt to provide empirical constraints on global climate sensitivity. | Tier 1* PMIP4-CMIP6 entry card |
| **Last Interglacial** (*lig127k*) 127 kyr ago | a) Evaluate climate model for warm period in northern hemisphere and high sea-level stand. b) Impacts of this climate on sea ice and ice sheets. | Tier 1* |
| **Mid-Pliocene Warm Period** (midPlioceneEoi400) 3.2 Ma ago | a) Earth System response to a long term to $CO_2$ forcing analogous to that of the modern. b) Significance of $CO_2$-induced polar amplification for the stability of the ice sheets, sea-ice and sea-level. | Tier 1* |

**Table 1: Characteristics, purpose and CMIP6 priority of the five PMIP4-CMIP6 experiments. * All experiments can be run independently. It is not mandatory to perform all Tier 1 experiments to take part in PMIP4-CMIP6, but it is mandatory to run at least one of the PMIP4-CMIP6 entry cards.**

| Period | GHG | Astronomical parameters | Ice-sheets | Tropospheric aerosols * | Land surface** | Volcanoes | Solar activity | Reference to be cited |
|---|---|---|---|---|---|---|---|---|
| **PMIP4-CMIP6 entry cards** | | | | | | | | |
| **Mid-Holocene** (*midHolocene*) 6 ky ago | $CO_2$: 264.4 ppm $CH_4$: 597 ppb $N_2O$: 262 ppb CFC : 0 $O_3$: pre-industrial | 6 kyr BP | as in PI | modified (if possible) | Interactive vegetation OR Interactive carbon cycle OR fixed to present day (depending on model complexity) | as in PI | as in PI | Otto-Bliesner et al, 2016 |
| **Last Glacial Maximum** (*lgm*) 21 ky ago | $CO_2$: 190 ppm $CH_4$: 375 ppm $N_2O$: 200 ppb CFC 0 $O_3$: pre-industrial | 21 kyr BP | modified (larger) | modified (if possible) | Interactive vegetation OR Interactive carbon cycle OR fixed to present day (depending on model complexity) | as in PI | as in PI | Kageyama et al, in prep, 2016 |
| **Tier 1 PMIP4-CMIP6 experiments** | | | | | | | | |
| **Last millennium** (*past1000*) 850-1849 CE | Time varying (Meinshausen et al., this issue) | time varying (Berger 1978, Schmidt et al., 2011) | as in PI | as in PI | time varying (land use) | time varying radiative forcing due to stratospheric aerosols | time varying | Jungclaus et al, to 2016 |
| **Last Interglacial** (*lig127k*) 127 ky ago | $CO_2$: 275 ppm $CH_4$: 685 ppb $N_2O$: 255 ppb CFC 0 $O_3$: pre-industrial | 127 ky BP | as in PI | modified (if possible) | Interactive vegetation OR Interactive carbon cycle OR fixed to present day (depending on model complexity) | as in PI | as in PI | Otto-Bliesner et al, 2016 |
| **Mid-Pliocene Warm Period** (midPlioceneEoi400) 3.2 My ago | $CO_2$: 400 ppm | as in PI | modified (smaller) | as in PI | Interactive vegetation OR Modified to mid-Pliocene OR fixed to present day (depending on model complexity) | as in PI | as in PI | Haywood et al, 2016 |

5 **Table 2: summary of change in boundary conditions with respect to** *piControl* **(abbreviated as "PI") for each PMIP4-CMIP6 experiment \* Only for models without fully interactive dust (see section 3.3). \*\* interactive carbon cycle, with computation of some characteristics of the vegetation such as the leaf area index (LAI), but without full vegetation dynamics.**

| Reference | Variables | Time period | Comments | Data available from |
|---|---|---|---|---|
| Mann et al. (2009) | MAT | 500-2006 CE | Gridded data set (5°) | http://science.sciencemag.org/content/suppl/2009/11/25/326.5957.1256.DC1 |
| PAGES 2k Consortium (2013) | MAT | past 2000 years | Individual sites; Arctic data updated 2014 | https://www.ncdc.noaa.gov/cdo/f?p=519:1:::::P1_STUDY_ID:12621 |
| Bartlein et al. (2011) | MAT, MAP, α, MTCO, MTWA | 6000±500 yr BP; 21000±1000 yr BP | Gridded data set (2°) | https://www.ncdc.noaa.gov/paleo/study/9897 |
| MARGO Project Members (2009) | Mean annual, winter, summer SST | 21000±2000 yr BP | Gridded data set (5°) | http://www.ncdc.noaa.gov/paleo/study/12034 http://doi.pangaea.de/10.1594/PANGAEA.733406 |
| Turney and Jones (2010) | MAT, SST | Maximum warmth during LIG | Individual sites (100 terrestrial; 162 marine) | http://onlinelibrary.wiley.com/store/10.1002/jqs.1423/asset/supinfo/JQS_1423_sm_suppInfo.pdf?v=1&s=1726938c44b8762e15aaf17514fc076c855b8ed1 |
| Capron et al. (2014); Capron et al. (subm.) | MAT, summer SST | 114-116ka, 119-121ka, 124-126ka, 126-128 ka, 129-131ka | 47 high latitude sites | doi.pangaea.de/10.1594/PANGAEA.841672 |
| Dowsett et al, (2012) | SST | 3.264-3.025 Ma | Further information available in Dowsett et al. (2016) | http://www.nature.com/nclimate/journal/v2/n5/full/nclimate1455.html#supplementary-information |
| Salzmann et al., 2013 | MAT | 3.3-3.0 Ma | | http://www.nature.com/nclimate/journal/v3/n11/extref/nclimate2008-s1.pdf |

5  Table 3: Examples of data syntheses for the PMIP4-CMIP6 periods. MAT: Mean Annual Temperature, MAP: Mean Annual Precipitation, α: ratio of the actual evaporation over potential evaporation, MTCO: Mean Temperature of the Coldest Month, MTWA: Mean Temperature of the Warmest Month, SST: Sea Surface Temperature.

| MIP abbreviaton | MIP full name | Themes of interactions |
|---|---|---|
| CF-MIP | Cloud Feedback Model Intercomparison Project | dedicated common idealized sensitivity experiment to be run in aquaplanet set up, *AMIPminus4K*, to be co-analysed in CF-MIP and PMIP. |
| ISMIP6 | Ice Sheet Model Intercomparison Project for CMIP6 | Assessment of the climate and cryosphere interactions and the sea level changes associated with large ice sheets. In particular, the *lig127k* simulation will be used to force ice sheet models in ISMIP6. Additional experiments co-designed by the PMIP and ISMIP groups are foreseen outside the CMIP6 exercise: transient interglacial experiments, with climate model output forcing an ice sheet model, and coupled climate-ice sheet experiments. |
| OMIP | Ocean Model Intercomparison Project | Mutual assessment of the role of the ocean in low-frequency variability, e.g. multi-decadal changes in ocean heat content or heat transport. Provide initial conditions for the ocean including long-term forcing history. |
| SIMIP | Sea Ice Model Intercomparison Project | Assessment of role of sea-ice in climate changes |
| AerChemMIP | Aerosols and Chemistry Model Intercomparison Project | Assessment of role of aerosols in climate changes, very helpful since this is a new aspect in PMIP experiments for the midHolocene, last interglacial and LGM |
| LS3MIP | Land Surface, Snow and Soil Moisture Model Intercomparison Project | Assessment of role of land surface processes in climate changes. |
| C4MIP | Coupled Climate Carbon Cycle Model Intercomparison Project | Assessment of carbon-cycle evolution and feedbacks between sub-components of the Earth System. Evaluation of paleo reconstructions of carbon storage. |
| LUMIP | Land-Use Model Intercomparison Project | Analysis of climate changes associated with Land Use changes (*past1000* experiment) |
| VolMIP | Volcanic Forcings Model Intercomparison Project | Analysis of specific volcanic events very useful for critical analysis of *past1000* simulations. VolMIP would systematically assess uncertainties in the climate response to volcanic forcing, whereas *past1000* simulations describe the climate response to volcanic forcing in long transient simulations where related uncertainties are due to chosen input data for volcanic forcing: mutual assessment of forced response. |
| DAMIP | Detection and Attribution Model Intercomparison Project | *past1000* simulations provide long-term reference background including natural climate variability for detection and attribution. |
| RFMIP | Radiative Forcing Model Intercomparison Project | Compare radiative forcing from LGM GHG as computed by climate models and by off-line fine-scale radiative transfer codes. |

5      **Table 4:interactions of PMIP with other CMIP6 MIPs**

# FIGURE CAPTIONS

Figure 1: Context of the PMIP4 experiments (from left to right: mPWP, Mid-Pliocene Warm Period; LIG, last interglacial; LGM, last glacial maximum; MH, mid-Holocene; LM, last millennium; H, CMIP6 historical simulation): (a)-(d) insolation anomalies (differences from 1950 CE), for July at 65°N, calculated using the programs of Laskar et al. (2004, panel (a)) and Berger (1978, panels (b)-(d)); (e) $\delta^{18}O$ (magenta, Lisiecki and Raymo, 2005, scale at left), and sea level (blue line, Rohling et al., 2014; blue shading, a density plot of eleven Mid-Pliocene sea level estimates (Dowsett and Cronin 1990; Wardlaw and Quinn, 1991; Krantz, 1991; Raymo et al., 2009; Dwyer and Chandler, 2009; Naish and Wilson, 2009; Masson-Delmotte et al., 2013; Rohling et al., 2014; Dowsett et al., 2016) scale at right); (f) and (g) $\delta^{18}O$ (magenta, Lisiecki and Raymo, 2005, $\delta^{18}O$ scale at left), and sea level (blue dots, with light-blue 2.5, 25, 75 and 97.5 percentile bootstrap confidence intervals, Spratt and Lisiecki, 2015; blue rectangle, LIG high-stand range, Dutton et al., 2015; dark blue lines, Lambeck et al., 2014, sea-level scale at right on panel (g)), (h) sea level (Kopp, et al., 2016, scale at right); (i) $CO_2$ for the interval 3.0-3.3 Ma shown as a density plot of eight Mid-Pliocene estimates (Raymo et al., 1996; Stap et al., 2016; Pagani et al., 2010; Seki et al., 2010; Tripati et al., 2009; Bartoli et al., 2011; Seki et al., 2010; Kurschner et al., 1996); (j) and (k) $CO_2$ measurements (Bereiter et al., 2015, scale at left); (l) $CO_2$ measurements (Schmidt et al, 2011, scale at right); (m) and (n) $CH_4$ measurements (Loulergue et al., 2008, scale at left); (o) $CH_4$ measurements (Schmidt et al, 2011, scale at right); (p) volcanic radiative forcing (Schmidt et al., 2012, scale at right); (q) total solar irradiance (Schmidt et al., 2012, scale at right).

Figure 2. Data-model comparisons in PMIP2 and CMIP5/PMIP3: (a) Land-ocean contrast in past, present and projected future climates. The black dots are the simulated long-term mean differences (*experiment – piControl*) in the relative warming/cooling over global land and global ocean. The red crosses show simulated changes where the model output has been sampled only at the locations for which there are temperature reconstructions for the *lgm*, *midHolocene* and *historical* (post-1850 CE) CMIP5 simulations. The red crosses overlap the black dots for the *midHolocene* and *historical* experiments. Area averages of palaeoclimate data are shown by bold blue crosses, with reconstruction uncertainties indicated by the finer lines. The regression line (magenta) shows that land-ocean contrasts are maintained across different climate states and are also consistent with palaeoclimatic data. (b) Boxplots of reconstructions based on fossil-pollen data (gray, Bartlein et al. 2011) and simulations (at the locations of the data) for the difference in mean annual precipitation (MAP, in mm/year) for the mid-Holocene (relative to present) in northern Africa (20°W-30°E; 5-30°N). OA: ocean-atmosphere coupled models; OAV: ocean-atmosphere-vegetation coupled models; OAC: ocean-atmopshere-carbon-cycle models. The comparison shows that although all models simulated wetter-that-present conditions in northern Africa for the mid-Holocene, they systematically underestimated the magnitude of the precipitation difference.

Figure 3: Changes in boundary conditions related to changes in ice sheets for the midPliocene-eoi400 (top) and lgm (middle: ICE-6G_C and bottom: GLAC-1D) experiments. Coastlines for palaeo-period shown as brown contours. Ice sheet boundaries for each period shown as red contour. Bright shading: changes in altitude over regions covered by ice sheets during the considered palaeo-period. Faded shading: changes in altitude over ice-free regions.

Figure 4: Maps of dust deposition (g m-2 a-1) simulated with the Community Earth System Model for the a. PI (pre-industrial) (Albani et al., 2016), b. MH (Mid-Holocene) (Albani et al., 2015), and c. LGM (Albani et al., 2014). Maps of dust deposition (g m-2 a-1) for the LGM d. simulated with the Hadley Centre Global Environment Model 2-Atmosphere (Hopcroft et al, 2015), and reconstructed from a global interpolation of paleodust data (Lambert et al., 2015).

Figure. 5: The calendar effect: (a) month-length anomalies, 140 ka to present, with the PMIP4 experiment times indicated by vertical lines.  The month-length anomalies were calculated using the formulation in Kutzbach and Gallimore (1988).  (b and c) The calendar effect on October temperature at 6 and 127 ka, calculated using Climate Forecast System Reanalysis near-surface air temperature ([https://www.earthsystemcog.org/ projects/obs4mips/](https://www.earthsystemcog.org/projects/obs4mips/)), 1981-2010 long-term means, and assuming the long-term mean differences in temperature

are zero everywhere.  (e and f) The calendar effect on October precipitation at 6 and 127 ka, calculated using the CPC Merged Analysis of Precipitation (CMAP) enhanced precipitation (http://www.esrl.noaa.gov/ psd/data/gridded/ data.cmap.html), 1981-2010 long-term means, and again assuming that the long-term mean differences in temperature are zero everywhere.  Calendar effects were calculated by interpolating present-day monthly temperature or precipitation to a daily time step as in Pollard and Reusch, 2002 (but using a mean-

preserving algorithm for pseudo-daily interpolation for monthly values; Epstein, 1991), and then recalculating the monthly means using the appropriate paleo calendar (Bartlein and Shafer, 2016).  Note that the 6 and 127 ka map patterns for both variables, while broadly similar, are not simply rescaled versions of one another.

     Figure 6: the PMIP4-CMIP6 experiments in the framework of CMIP6, with associated MIPs, and in the

framework of PMIP4, with its working groups.

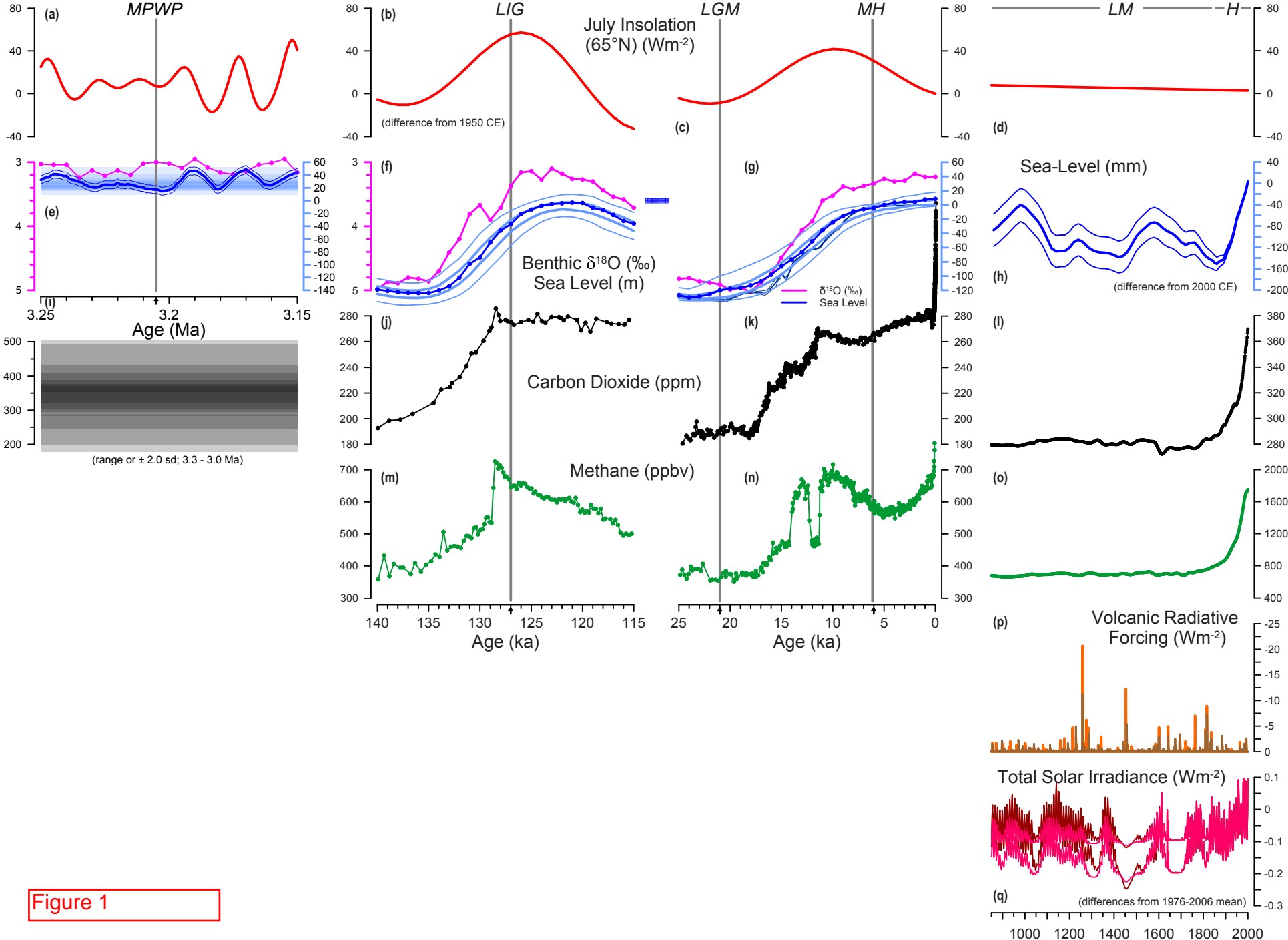

**Figure 1**

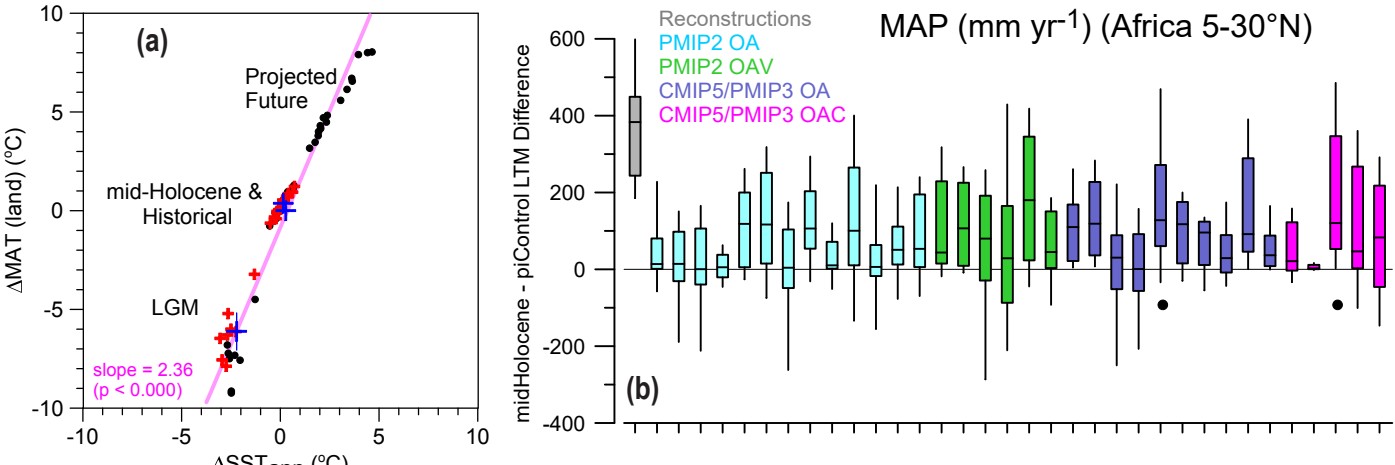

Ice mask ——  Ice Sheet  -200  0  1000  2000  3000  (m)

Land/sea mask ——  Sea/Land  -40  0  40  80  120 (m)

Figure 4

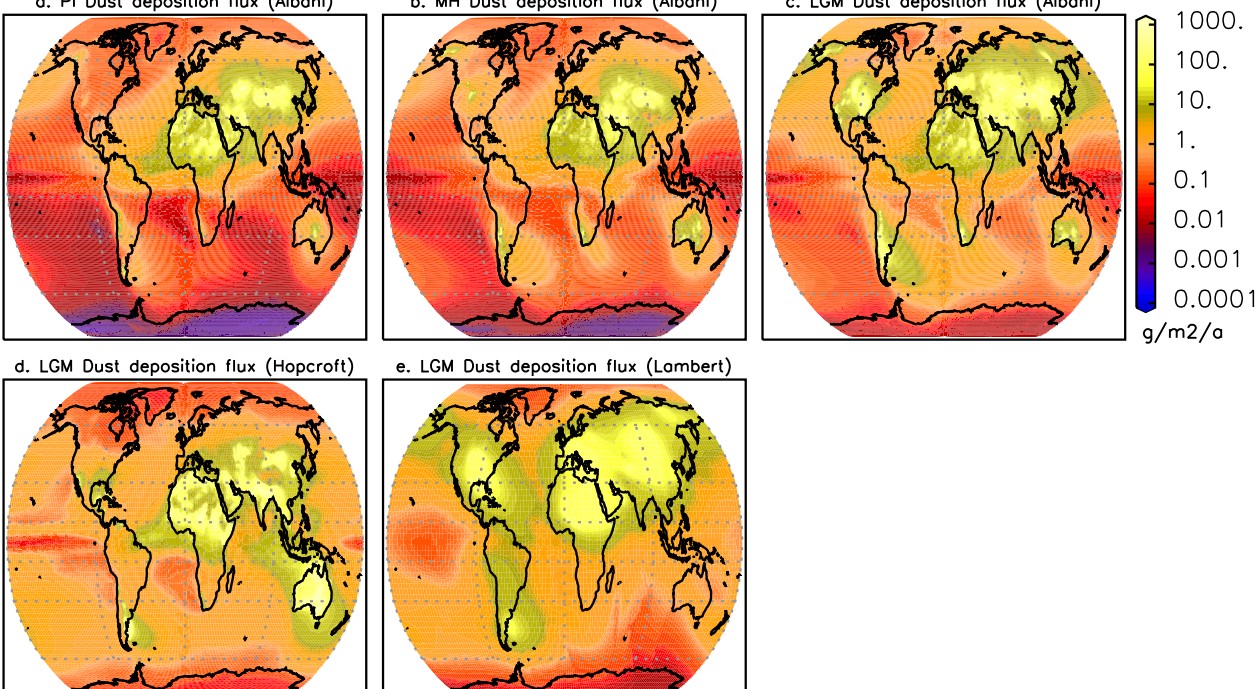

a. PI Dust deposition flux (Albani)   b. MH Dust deposition flux (Albani)   c. LGM Dust deposition flux (Albani)

d. LGM Dust deposition flux (Hopcroft)   e. LGM Dust deposition flux (Lambert)

1000.
100.
10.
1.
0.1
0.01
0.001
0.0001
g/m2/a

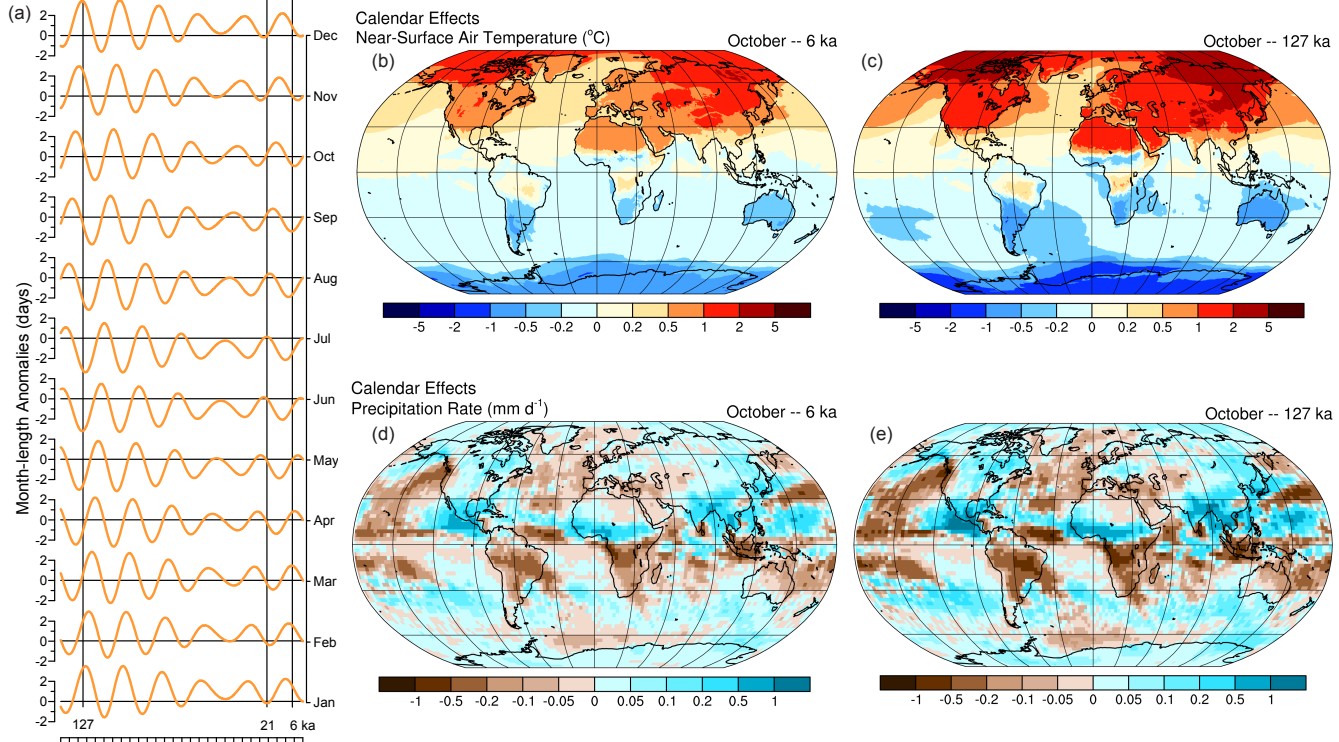

Figure 5

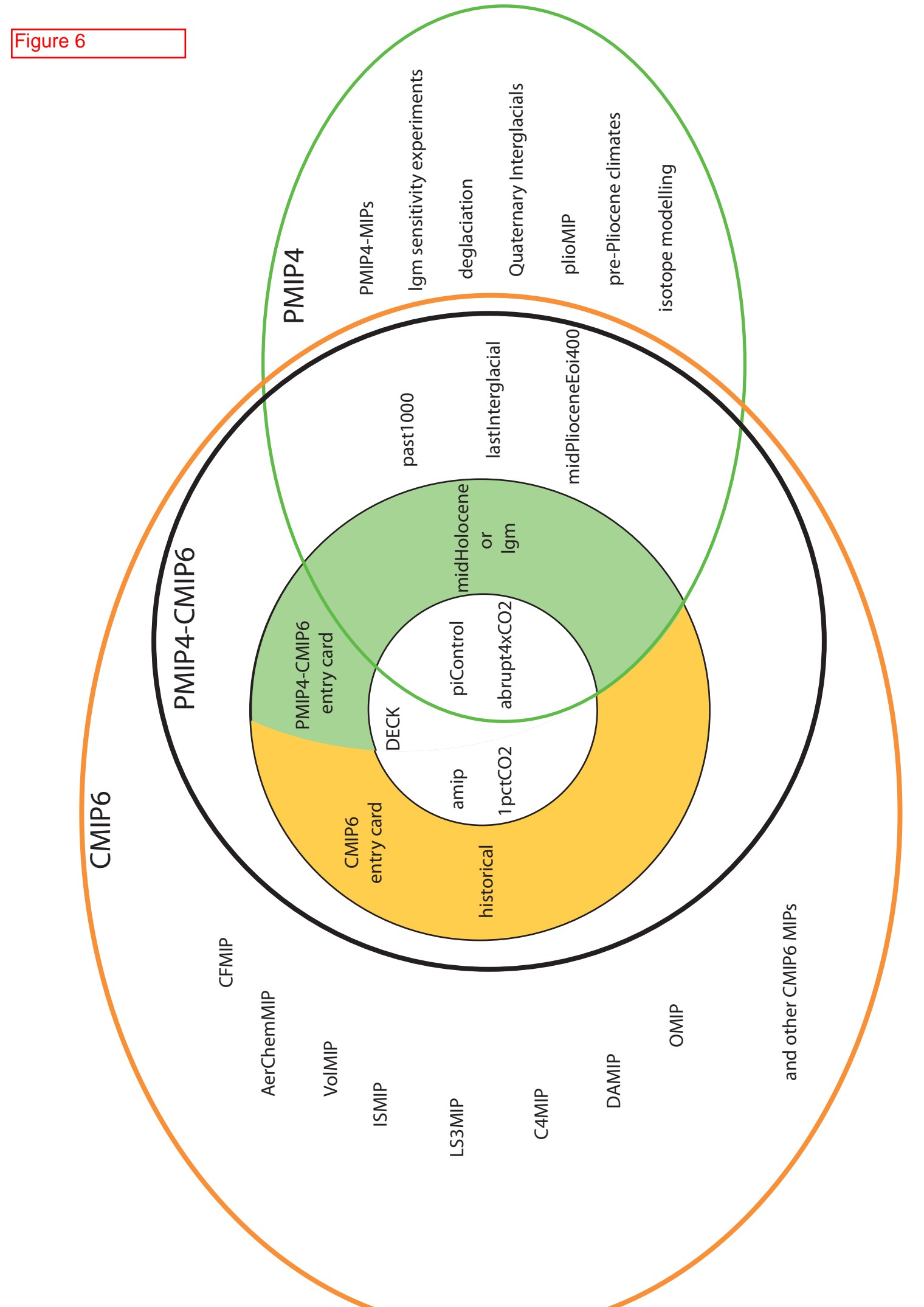