# Peer review of "The PMIP4 contribution to CMIP6 - Part 1: Overview and over-arching analyses plan\"

_Geoscientific Model Development, 2016_

## Author Comment (AC1) · 26 May 2016

**PMIP4-CMIP6: the contribution of the Paleoclimate Modelling Intercomparison Project to CMIP6**

Masa Kageyama [1], Pascale Braconnot[1], Sandy P. Harrison[2], Alan M. Haywood[3], Johann Jungclaus[4], Bette L. Otto-Bliesner[5], Jean-Yves Peterschmitt[1], Ayako Abe-Ouchi[6,7], Samuel Albani[8], Patrick J. Bartlein[9], Chris Brierley[10], Michel Crucifix[11], Aisling Dolan[3], Laura Fernandez-Donado[12], Hubertus Fischer[13], Peter O. Hopcroft[14], Ruza F. Ivanovic[3], Fabrice Lambert[15], Dan J. Lunt[14], Natalie M. Mahowald[16], W. Richard Peltier[17], Steven J. Phipps[18], Didier M. Roche[1,19], Gavin A. Schmidt[20], Lev Tarasov[21], Paul J. Valdes[14], Qiong Zhang[22], Tianjun Zhou[23]

[1]Laboratoire des Sciences du Climat et de l'Environnement, LSCE/IPSL, CEA-CNRS-UVSQ, Université Paris-Saclay, F-91191 Gif-sur-Yvette, France
[2]Centre for Past Climate Change and School of Archaeology, Geography and Environmental Science (SAGES) University of Reading, Whiteknights, RG6 6AH, Reading, United Kingdom
[3]School of Earth and Environment, University of Leeds, Woodhouse Lane, Leeds, LS2 9JT, United Kingdom
[4]Max Planck Institute for Meteorology, Bundesstrasse 53, 20146 Hamburg, Germany
[5]National Center for Atmospheric Research, 1850 Table Mesa Drive, Boulder, Colorado 80305, United States of America
[6]Atmosphere Ocean Research Institute, University of Tokyo, 5-1-5, Kashiwanoha, Kashiwa-shi, Chiba 277-8564, Japan
[7]Japan Agency for Marine-Earth Science and Technology, 3173-25 Showamachi, Kanazawa, Yokohama, Kanagawa, 236-0001, Japan
[8]Institute for Geophysics and Meteorology, University of Cologne, Cologne, Germany
[9]Department of Geography, University of Oregon, Eugene, OR 97403-1251, United States of America
[10]University College London, Department of Geography, WC1E 6BT, United Kingdom
[11]Université catholique de Louvain, Earth and Life Institute, Louvain-la-Neuve, Belgium
[12]Dpto. Física de la Tierra, Astronomía y Astrofísica II, Instituto de Geociencias (CSIC-UCM), Universidad Complutense de Madrid, Spain
[13]Climate and Environmental Physics, Physics Institute & Oeschger Centre for Climate Change Research, University of Bern, Sidlerstrasse 5, CH-3012 Bern, Switzerland
[14]School of Geographical Sciences, University of Bristol, Bristol, United Kingdom
[15]Catholic University of Chile, Department of Physical Geography, Santiago, Chile
[16]Department of Earth and Atmospheric Sciences, Bradfield 1112, Cornell University, Ithaca, NY 14850, United States of America
[17]Department of Physics, University of Toronto, 60 St. George Street, Toronto, Ontario M5S 1A7, Canada
[18]Institute for Marine and Antarctic Studies, University of Tasmania, Private Bag 129, Hobart, TAS 7001, Australia
[19]Earth and Climate Cluster, Faculty of Earth and Life Sciences, Vrije Universiteit Amsterdam, Amsterdam, the Netherlands
[20]NASA Goddard Institute for Space Studies and Center for Climate Systems Research, Columbia University 2880 Broadway, New York, NY 10025, United States of America
[21]Department of Physics and Physical Oceanography, Memorial University of Newfoundland and Labrador, St. John's, NL, A1B 3X7, Canada
[22]Department of Physical Geography, Stockholm University and Bolin Centre for Climate Research, Stockholm, Sweden
[23]LASG, Institute of Atmospheric Physics, Chinese Academy of Sciences, P.O. Box 9804, Beijing 100029, China

*Correspondence to*: Masa Kageyama (Masa.Kageyama@lsce.ipsl.fr)

[Figure]

[Figure]

**Abstract.**

The goal of the Palaeoclimate Modelling Intercomparison Project (PMIP) is to understand the response of the climate system to changes in different climate forcings and to feedbacks. Through comparison with observations of the environmental impacts of these climate changes, or with climate reconstructions based on physical, chemical or biological records, PMIP also addresses the issue of how well state-of-the-art models simulate climate changes. Palaeoclimate states are radically different from those of the recent past documented by the instrumental record and thus provide an out-of-sample test of the models used for future climate projections and a way to assess whether they have the correct sensitivity to forcings and feedbacks. Five distinctly different periods have been selected as focus for the core palaeoclimate experiments that are designed to contribute to the objectives of the sixth phase of the Coupled Model Intercomparison Project (CMIP6). This manuscript describes the motivation for the choice of these periods and the design of the numerical experiments, with a focus upon their novel features compared to the experiments performed in previous phases of PMIP and CMIP as well as the benefits of common analyses of the models across multiple climate states. It also describes the information needed to document each experiment and the model outputs required for analysis and benchmarking.

[Figure]

[Figure]

**1 Introduction**

**1.1 Why model paleoclimates?**

Instrumental meteorological and oceanographic data, available for the period extending from the middle of the 19th century, describe the manner in which Earth's surface climate has evolved since the beginning of the industrial revolution. These data show a global warming of ~0.85°C to have occurred since this time, a warming that is more intense over land than over the oceans, and more intense at high latitudes compared to the tropics (Hartmann et al, 2013, Sutton et al, 2007). This recent climate change has been substantially controlled by the increase of atmospheric greenhouse gases due to human activities, amplified by the action of feedbacks associated with atmospheric water vapor and clouds (e.g. Dufresne and Bony, 2008), the albedos of snow and ice, with changes in the land cover or in ocean properties and circulation (Cubasch et al, 2013). This process-based understanding of the climate system is embedded within the climate models used to project changes in future climates. The skill of these climate models is most commonly evaluated in comparison to the present climate and climate change since the pre-industrial age (1850 CE). However concentrations of atmospheric greenhouse gases are projected to increase significantly during the 21st century, reaching levels well outside the range of recent millennia. Thus, in making future projections, models are operating well outside the conditions for which they have been validated. The credibility of climate projections needs to be assessed using information on longer-term palaeoclimate changes, particularly for intervals when the climate change compared to present was as large as the anticipated future change.

We have to look back several million years to find a period of Earth's history when atmospheric $CO_2$ concentrations were similar to the present day (the mid-Pliocene warm period, 3.2 million years ago) and several tens of million years (e.g. the early Eocene, ~55 to 50 million years ago) for much higher levels. During these ancient periods, topography, bathymetry, land-ocean distributions and/or ice sheets were different from today, and the mechanisms for increasing atmospheric $CO_2$ were likely much slower than anthropogenic fossil fuel emissions. However, although these periods are not perfectly analogous to the future, they offer key insight into climate processes that operate in a higher $CO_2$, warmer world (e.g. Lunt et al, 2010, 2012, Caballero and Huber, 2010). On the other hand, the main drivers of climatic changes in Earth's most recent period, the Quaternary (2.5 million years ago to present), are the astronomical parameters driving the seasonal and latitudinal distribution of incoming solar energy, as well as greenhouse gas fluctuations, with levels much lower than present. During this period, the Earth's geography was more similar to today and some of the more rapid climate transitions that took place occurred on human-relevant timescales (decades to centuries; e.g. Marcott et al, 2014, Steffensen et al, 2008). By combining several past periods, we can provide a broad picture of the climate response to external forcings, and to benefit from the rich resource of paleoclimates and paleoenvironments.

There are numerous palaeoclimate records documenting the evolution of Earth's climate before instrumental records (Masson-Delmotte et al, 2013). Some of these records are based on physical and chemical properties of the atmosphere, vegetation and ocean; such as oxygen and carbon isotopes, which have been preserved in various geological archives such as ice, speleothems or microscopic plankton shells (e.g. Caley et al, 2014, for a model-isotopic data comparison). Other records, such as changes in marine and terrestrial floral and faunal assemblages and distributions (MARGO Project Members, 2009; Prentice et al., 2000) for changes in and

surface hydrology and water storage (Kohfeld and Harrison 2000), reflect the impact of climate changes on the ambient environment but can be used to reconstruct climate parameters either qualitatively or statistically (e.g. MARGO Project Members, 2009; Bartlein et al., 2011). Overall, there is a wealth of palaeoclimatic and palaeoenvironmental data showing large variations in the Earth's climate prior to the industrial era, commensurate with the magnitude of projected changes in the future.

Replicating the totality of those climate changes with state-of-the-art climate models is a challenge (Braconnot et al, 2012, Harrison et al, 2015). It is challenging, for example, to represent the correct amplitude of past climate changes such as glacial-interglacial temperature differences (e.g. the temperatures at the Last Glacial Maximum, ~21,000 years ago, vs. the pre-industrial temperatures, cf. Harrison et al., 2014) or the correct spatial patterns such as the northward extension of the African monsoon during the mid-Holocene, ~ 6,000 years ago (Perez-Sanz et al., 2014). Interpreting palaeoenvironmental data can also be challenging, and in particular disentangling the relationships between changes in large-scale atmospheric or oceanic circulation, broad-scale regional climates and local environmental responses to these changes. This challenge is paralleled by concerns about future local or regional climate changes and their impact on the environment. Modelling palaeoclimates is therefore a means to understand past climate and environmental changes better, using physically based tools, as well as a means to evaluate model skill in forecasting the responses to major drivers.

**1.2 The Palaeoclimate Modelling Intercomparison Project (PMIP)**

The Palaeoclimate Modelling Intercomparison Project (PMIP) was established in the 1990's in order to understand the mechanisms of past climate changes, in particular the role of the different climate feedbacks, and to evaluate how well climate models used for climate projections simulate well-documented climates outside the range of present and recent climate variability. To achieve these goals, PMIP has actively fostered paleo-data syntheses, model-data comparisons and multi-model analyses. PMIP provides a forum for discussion of experimental design and appropriate techniques for comparing model results with palaeoclimatic reconstructions.

Since its initial phase the evolution of PMIP has closely followed model developments for the Atmospheric Model Intercomparison Project (AMIP) and then the Coupled Model Intercomparison project (CMIP). The initial focus was on the results from Atmospheric General Circulation Models (PMIP1, Joussaume and Taylor 1995) and was extended to coupled Atmosphere-Ocean General Circulation Models (AOGCMs) and AOGCMs including representations of the carbon cycle feedbacks in PMIP2 (Braconnot et al, 2007) and PMIP3 (Braconnot et al, 2012). Two climatic periods have been a major focus in PMIP since its initial phase: the mid-Holocene (MH, ~6,000 years ago) and the Last Glacial Maximum (LGM, ~21,000 years ago). The rationale for studying the Last Glacial Maximum was to evaluate model performance in a well-documented cold climatic extreme and to examine the role of forcings and feedbacks in creating this climate state. The rationale for the mid-Holocene was to evaluate and analyse the models during a period when the northern hemisphere was characterized by enhanced monsoons, extra-tropical continental aridity and much warmer summers. These two periods are considered as reference points for assessing the sensitivity of the climate system to changes in atmospheric $CO_2$ concentration and orbitally-induced changes in tropical circulation and the monsoons,

respectively (Braconnot et al 2012, Harrison et al. 2015). Evaluations of the simulations of these two periods made in successive phases of PMIP provide a unique overview of the evolution of the ability of climate model to reproduce large changes compared to today (Harrison et al. 2013, Flato et al, 2013).

5  Palaeoclimate experiments were included for the first time in the ensemble of simulations made in during the fifth phase of CMIP (Taylor et al, 2012). In addition to the MH and LGM simulations described above, transient simulations of the millennium prior to the industrial epoch (LM, 850-1850 CE) were also included in CMIP5 (Schmidt et al, 2011, 2012), to study the mechanisms of decadal to centennial climate variability (natural variability vs. impact of solar, volcanic and anthropogenic forcings). Simulations of the LM have used models of

10  varying complexity, evolving from energy balance models (e.g. Crowley, 2000), via Earth system models of intermediate complexity (Goosse et al., 2005), to complex coupled atmosphere –ocean general circulation models (AOGCM, e.g. Gonzalez-Rouco et al., 2006) and Earth System Models that include components like the carbon cycle (Jungclaus et al., 2010). The focus in CMIP5 has been on coupled model evaluation based on a common protocol describing a variety of suitable forcing boundary conditions (Schmidt et al., 2011; 2012). and

15  process understanding (e.g. Lehner et al., 2013; Sicre et al., 2013; Jungclaus et al., 2014), including the assessment of variability modes (e.g. Raible et al., 2014) and comparisons with reconstructions (e.g. Bothe et al., 2013; Fernandez-Donado et al., 2013). Single-model ensembles of simulations have provided an understanding of the importance of internal versus forced variability and the individual forcings when comparing to reconstructions (Phipps et al., 2013; Schurer et al., 2014; Otto-Bliesner et al., 2016). Thanks to this formal

20  inclusion of the LM, MH and LGM simulations in the CMIP5 exercise, it was possible to compare the mechanisms causing past and future climate changes in a rigorous way and evaluate of the models used for projections under very different climate states from the present one (e.g. Harrison et al, 2013, Harrison et al, 2015).

25  In its third phase, PMIP became an umbrella for analyses of other time periods and provided a framework for analyses across multiple time periods. PlioMIP (Haywood et al., 2010, 2011) coordinates climate model experiments for the mid-Pliocene Warm Period (mPWP, ca. 3.3 to 3 million years ago). The mPWP had $CO_2$ levels similar to today, but vegetation reconstructions (Salzmann et al., 2008) indicate that the area of deserts decreased and boreal forests replaced tundra.  Climate model simulations produce global mean surface air

30  temperature ranging from +1.9°C and +3.6°C (relative to each model's pre-industrial control) and an enhanced hydrological cycle (Haywood et al., 2013), with strengthened monsoons (Zhang et al. 2013). These simulations also show that meridional temperature gradients were reduced (due to high latitude warming), which has significant implications for the stability of polar ice sheets and sea level in the future (e.g. Miller et al. 2012). PMIP3 also saw the initiation of comparison of available simulations and reconstruction for the last interglacial

35  period (Lunt et al. 2013) and discussions about the ability of climate models to produce a rate of ice-sheet melting in agreement with a global sea level at least 5m higher than now (Masson-Delmotte et al., 2013; Dutton et al., 2015). First discussions on transient simulations of climate behaviour, focusing on the last interglacial period and the last deglaciation (Ivanovic et al, 2015) were also initiated.

[Figure]

[Figure]

A measure of the success of PMIP3 is provided by the number of participating groups (more than 20) and the fact that PMIP results were used for ten figures in the last IPCC report (Masson-Delmotte et al. 2013, Flato et al. 2013). However, the project also identified significant knowledge gaps and areas where progress is needed; PMIP4 has been designed to address these.

5   **1.3 PMIP4 experiments in CMIP6**

The design of PMIP4 simulations to be included as part of CMIP6 was built on the recognition that PMIP simulations naturally address the key CMIP6 question "How does the Earth System respond to forcing" for multiple forcings and in climates states very different from the current or historical climates. Comparisons with observations enable us to determine whether the modelled responses are realistic. PMIP also addresses key

10   question 2 "What are the origins and consequences of systematic model biases?" PMIP simulations and data-model comparisons will show whether the biases in the present-day simulations are also found in other climate states. More importantly, analyses of PMIP simulations will show whether present-day biases have an impact on the magnitude of simulated climate changes. Finally, PMIP is also relevant for question 3 "How can we assess future climate changes given climate variability, predictability and uncertainties in scenarios?" through

15   examination of these questions for documented past climate states and via the use of the last millennium simulations as reference state for natural variability.

The choice of time periods for palaeoclimate experiments in CMIP6 is based on previous experience in the PMIP project. For each target period, there is a quantified understanding of the relevant climate drivers and an

20   extensive network and/or synthesis of environmental observations. The five periods proposed for PMIP4-CMIP6 represent climate states with different greenhouse gas concentrations, astronomical parameters, smaller or larger ice sheets and modified hydrological cycles (Figure 1), consistent with the need to provide a large sample of the climate response to different forcings. While the five periods represent very different climate states, all of them cover aspects of the climate system that are relevant to future climate change (Figure 1). The periods are:

25   -   the millennium before the start of the industrial revolution, from 850 to 1850 CE (*past1000*)

-   the mid-Holocene, 6,000 years ago (*midHolocene*)

-   the Last Glacial Maximum, 21,000 years ago (*lgm*)

-   the Last Interglacial, 127,000 years ago (*lig127k*)

-   the mid-Pliocene Warm Period, 3.2 million years ago (*midPliocene-eoi400*)

30   All the experiments have been run by several modelling groups, most as formal intercomparisons with a standardized protocol (e.g. LM, MH, LGM, mPWP). The names of the experiments in PMIP4 simulations included in CMIP6 are consistent with the PMIP3-CMIP5 names for the last millennium, mid-Holocene and Last Glacial Maximum and consistent with the PlioMIP naming convention for the mid-Pliocene Warm Period (Haywood et al, 2016). All the experiments can be run independently and have value for comparison to the

35   CMIP6 DECK and historical experiments. We have therefore given them equal priority, Tier 1, within CMIP6 (Table 1). It is not mandatory for groups wishing to take part in PMIP4-CMIP6 to run all five PMIP4-CMIP6 Tier 1 experiments. It is however mandatory to run at least one of the experiments that were run in previous phases of PMIP, i.e. the *midHolocene* or the *lgm*. These are considered as "entry cards" for participation in PMIP4-CMIP6.

[revised manuscript text omitted]

As discussed above, the MH and the LGM provide examples of strongly contrasted climate states (Figure 1, Table 1). There are extensive syntheses of marine and terrestrial data for both intervals, documenting environmental responses to changing climate. The MH provides an opportunity to examine the response to

orbitally-induced changes in the seasonal and latitudinal distribution of insolation. The LGM provides an opportunity to examine the impact of changes in ice sheets, land-sea distribution and greenhouse gases on climate. The LGM is particularly relevant because the forcing and temperature response was as large as (although of opposite sign) to that projected for the end of the 21st century. Both periods constitute test cases for

5    our understanding of mechanisms of climate change, such as the interplay between circulation changes and radiation/cloud changes, the respective strengths of feedbacks from different components of the climate system, and for our understanding of the connections between global and regional climate changes. Because these periods have been studied in earlier phases of PMIP, they provide the opportunity to evaluate whether increased model resolution and complexity has led to improvement in the representation of circulation patterns and in the

10   fidelity of regional climate changes.

Evaluation of the PMIP3-CMIP5 MH and LGM experiments has demonstrated that climate models simulate changes in large-scale features of climate that are governed by the energy and water balance reasonably well, including changes in land-sea contrast (Figure 2a) and high-latitude amplification of temperature changes

15   (Izumi et al., 2013; Izumi et al., 2015). They also simulate the scaling of precipitation changes with respect to temperature changes at a hemispheric scale realistically (Li et al., 2013). Thus, evaluation of the PMIP3-CMIP5 MH and LGM simulations confirms that the relationships between large-scale patterns of temperature and precipitation change in future projections are believable (Harrison et al., 2015). However, the PMIP3-CMIP5 simulations of MH and LGM climates show only moderate skill in predicting observed patterns of climate

20   change overall (Hargreaves et al., 2013; Hargreaves and Annan, 2014; Harrison et al., 2014; Harrison et al., 2015) and this arises because of persistent problems in simulating regional climates (e.g. Mauri et al., 2014; Perez-Sanz et al., 2014; Harrison et al., 2015). State-of-the-art models still cannot reproduce the northward penetration of the African monsoon in response to MH orbital forcing (Figure 2b, Perez-Sanz et al., 2014, Pausata et al, 2016), for example. Both inadequate representation of feedbacks and model biases could contribute

25   to this mismatch (see e.g. Zheng and Braconnot, 2013) but are unlikely to be sufficient to reconcile the PMIP3-CMIP5 simulations with observations.

Systematic biases in the simulation of regional climates means that state-of-the-art models are generally better at simulating mean values of any climate variable than at simulating the spatial variability or the geographical

30   patterning in that variable (Harrison et al., 2014). Although the benchmarking of the PMIP3-CMIP5 MH and LGM experiments shows that some models consistently perform better than others (Harrison et al., 2014), better performance in palaeo-simulations is not consistently related to better performance under modern conditions (Harrison et al., 2015). The ability to simulate modern climate regimes and processes does not guarantee that a model will be good at simulating climate changes, emphasising the importance of testing models against the

[revised manuscript text omitted]

5   deficiencies in the southern hemisphere records, or poor representation of internal variability and/or an overestimation of the forced response in the simulations.

**Figure 3 : Color lines: temperatures anomalies (w.r.t. the 1500-1850 CE average) simulated by PMIP3-CMIP5 models for the last millennium and historical periods, averaged for the northern (l.h.s) and southern (r.h.s) hemisphere. Grey**
10   **shading: uncertainty envelope of available reconstructions. All series are filtered using a 31-point moving average filter. Adapted from Fernandez-Donado, 2015.**

The PMIP4-CMIP6 *past1000* simulations will be based on experience gained in PMIP3-CMIP5, in which more than a dozen modelling groups participated and a total of 15 *past1000* experiments where stored in the ESGF
15   database. The PMIP4-CMIP6 *past1000* simulations build on the DECK experiments, in particular the pre-industrial control (*piControl*) simulation as unforced reference, and the *historical* simulations (Eyring et al., 2015). Moreover, *past1000* simulations provide initial conditions for *historical* simulations starting in the 19th century that are considered superior to the *piControl* state as it includes integrated information from the forcing history (e.g. large volcanic eruptions in the early 19th century). The PMIP4-CMIP6 *past 1000* simulation will
20   benefit from a new, more comprehensive reconstruction of volcanic forcing (Sigl et al., 2015) and an experimental protocol that ensures a more continuous transition from the pre-industrial past to the future. Higher-resolution simulations will allow a greater range of regional processes, such as the role of storm-tracks and blocking on regional precipitation, to be analyzed.

25   **2.3 The last interglacial (*lig127k*)**
The Last Interglacial (ca 130-115 ka) was characterized by a northern hemisphere insolation seasonal cycle even larger than for the mid-Holocene (Figure 1, Table 1), resulting in a strong polar amplification of temperatures and reduced Arctic sea ice, and global sea level was at least 5 m higher than now for at least several thousand years (Masson-Delmotte et al., 2013; Dutton et al., 2015). Both the Greenland and Antarctic ice sheets
30   contributed to this sea level rise, making it an important period for testing our knowledge of climate-ice sheet interactions in warm climates. There are more quantitative climate reconstructions available for the Last Interglacial than earlier interglacials, despite challenges in establishing the reliable chronologies, making it feasible to assess regional climate changes.

35   Climate model simulations of the Last Interglacial, reviewed and assessed in the AR5, varied in their forcings and were not necessarily made with the same model/same resolution as the CMIP5 future projections. Quantitative reconstructions of annual surface temperature change were available for comparison to these simulations (Figure 4) though with the caveat that the warmest phases were not necessarily globally synchronous (Masson-Delmotte et al., 2013). Nevertheless, comparison exercises showed large-scale discrepancies between
40   simulations and reconstructions, particularly in regard to temperature trends over Greenland and the Southern Ocean (Bakker et al., 2013, Lunt et al, 2013).

[Figure]

[Figure]
* * *
**Figure 4: Figure 5.6 from Chapter 5 of the IPCC AR5 WGI report (Masson-Delmotte et al., 2013, page 408). Changes in surface temperature for the Last Interglacial (LIG) as reconstructed from data and simulated by an ensemble of climate model experiments in response to orbital and well-mixed greenhouse gas (WMGHG) forcings. (a) Proxy data syntheses of annual surface temperature anomalies as published by Turney and Jones (2010) and McKay et al. (2011). McKay et al., (2011) calculated an annual anomaly for each record as the average sea surface temperature (SST) of the 5-kyr period centred on the warmest temperature between 135 ka and 118 ka and then subtracting the average SST of the late Holocene (last 5 kyr). Turney and Jones (2010) calculated the annual temperature anomalies relative to 1961–1990 by averaging the LIG temperature estimates across the isotopic plateau in the marine and ice records and the period of maximum warmth in the terrestrial records (assuming globally synchronous terrestrial warmth). (b) Multi-model average of annual surface air temperature anomalies simulated for the LIG computed with respect to preindustrial. The results for the LIG are obtained from 16 simulations for 128 to 125 ka conducted by 13 modelling groups (Lunt et al., 2013). (c) Seasonal SST anomalies. Multi-model zonal averages are shown as solid line with shaded bands indicating 2 standard deviations. Plotted values are the respective seasonal multi-mean global average. Symbols are individual proxy records of seasonal SST anomalies from McKay et al. (2011). (d) Seasonal terrestrial surface temperature anomalies (SAT). As in (c) but with symbols representing terrestrial proxy records as compiled from published literature (Table 5.A.5). Observed seasonal terrestrial anomalies larger than 10°C or less than –6°C are not shown. In (c) and (d) JJA denotes June – July – August and DJF December – January – February, respectively.**
* * *
The PMIP4-CMIP6 *lig127k* experiment will help to determine the interplay of warmer atmospheric and oceanic temperatures, changed precipitation, and changed surface energy balance on ice sheet thermodynamics and dynamics (Table 1). The major changes in the experimental protocol for lig127k, compared to the pre-industrial DECK experiment, are changes in astronomical parameters and greenhouse gases (Table 2; Otto-Bliesner et al, 2016). Analyses of these simulations will benefit from the concerted effort by the paleodata community to provide a spatial-temporal picture of last interglacial temperature change (Capron et al., 2014) as well as phasing of the timing of the contributions of Greenland and Antarctica to the global sea level (Winsor et al., 2012; Steig et al., 2015). Regional responses of tropical hydroclimate and of polar sea ice can be assessed and compared to the *mid-Holocene*. Outputs from the *lig127k* experiment will be used by ISMIP6 to force standalone ice sheet experiments (*lastIntergacialforcedism*). The *lig127k* experiment will also be the starting point of a transient experiment covering the interglacial to be run within PMIP4.

**2.4 The mid-Pliocene Warm Period (*midPliocene-eoi400*)**

The Pliocene epoch was the last time in Earth history when atmospheric $CO_2$ concentrations approached modern values (~400 ppmv) whilst at the same time retaining a near modern continental configuration (Figure 1, Table 1, **Erreur ! Source du renvoi introuvable.**). The IPCC 5[th] Assessment report chapter 5 (Masson-Delmotte et al., 2013) states that model–data comparisons for the Pliocene provide high confidence that mean surface temperature was warmer than pre-industrial (Dowsett et al., 2012; Haywood et al., 2013). However, as was the case for the Last Interglacial, the mid-Pliocene simulations were not always derived from the same model at the same resolution as the CMIP5 future projections.
* * *
**Figure 5: Figure 1 in Box 5.1 from Chapter 5 of the IPCC AR5 WGI report (Masson-Delmotte et al., 2013, page 397). Comparison of data and multi-model mean (MMM) simulations, for four periods of time, showing (a) sea surface temperature (SST) anomalies, (b) zonally averaged SST anomalies, (c) zonally averaged global (green) and land (grey) surface air temperature (SAT) anomalies and (d) land SAT**

[Figure]

[Figure]

**anomalies. The time periods are 2081–2100 for the Representative Concentration Pathway (RCP) 8.5 (top row), Last Glacial Maximum (LGM, second row), mid-Pliocene Warm Period (MPWP, third row) and Early Eocene Climatic Optimum (EECO, bottom row). Model temperature anomalies are calculated relative to the pre-industrial value of each model in the ensemble prior to calculating the MMM anomaly (a, d; colour shading). Zonal MMM gradients (b, c) are plotted with a shaded band indicating 2 standard deviations. Site specific temperature anomalies estimated from proxy data are calculated relative to present site temperatures and are plotted (a, d) using the same colour scale as the model data, and a circle-size scaled to estimates of confidence. Proxy data compilations for the LGM are from Multiproxy Approach for the Reconstruction of the Glacial Ocean surface (MARGO) Project Members (2009) and Bartlein et al. (2011), for the MPWP are from Dowsett et al. (2012), Salzmann et al. (2008) and Haywood et al. (2013) and for the EECO are from Hollis et al. (2012) and Lunt et al. (2012). Model ensemble simulations for 2081–2100 are from the CMIP5 ensemble using RCP 8.5, for the LGM are seven Paleoclimate Modelling Intercomparison Project Phase III (PMIP3) and Coupled Model Intercomparison Project Phase 5 (CMIP5) models, for the Pliocene are from Haywood et al., (2013), and for the EECO are after Lunt et al. (2012). [ Note: permission has been sought to use the third line of this figure only (i.e. the MPWP results). Until this permission is received, we follow the IPCC rules for using the figures from the fifth assessment report. ]**

The PMIP4-CMIP6 *midPliocene-eoi400* experiment is designed to understand the long term response of the climate system to a near modern concentration of atmospheric $CO_2$ (longer term climate sensitivity or Earth System Sensitivity), and to understand the response of ocean circulation, Arctic sea-ice, modes of climate variability (e.g. El Niño Southern Oscillation), as well as the global response in the hydrological cycle and regional changes in monsoon systems (Table 1). Boundary conditions are provided by the US Geological Survey Pliocene Research and Synoptic Mapping Project (PRISM4: Dowsett et al. 2016). These include required modifications to global ice distributions, topography/bathymetry, vegetation and $CO_2$ (Table 2, Section 3). The simulation has societal relevance because of its potential to inform policy makers on required emission reduction scenarios designed to prevent an increase in global annual mean temperatures by more than 2 to 3 °C beyond 2100 AD.

**3. Experimental set up and model configuration**

The modified forcings and boundary conditions for each PMIP4-CMIP6 palaeoclimate simulation are summarised in Table 2. The complete details of the experimental protocols are given in a series of companion papers: Otto-Bliesner et al for the *midHolocene* and *lig127ka* experiments, Kageyama et al for the *lgm*, Jungclaus et al for the *past1000* and Haywood et al (2016) for the *midPliocene-eoi400* experiment. These papers also explain how the boundary conditions for each period have been built and constitute key references for the experimental protocol for each of the PMIP4-CMIP6 simulations. Here we provide guidelines that are common to all of the experiments, focusing particularly on the implementation of the boundary conditions where there is a need to ensure consistency between CMIP6 and PMIP4 experiments.

**3.1 Model version and set-up**

The climate models taking part in CMIP6 are very diverse: some representing the solely physics of the climate system; some including the carbon cycle and other biogeochemical cycles; some even including interactive natural vegetation and/or interactive dust cycle/aerosols. It is mandatory that the model versions used for the PMIP4-CMIP6 experiments are the exactly the same as for the other CMIP6 experiments, in particular the

[Figure]

[Figure]

DECK and *historical* simulations. Except for the *past1000* simulation, all the other PMIP4-CMIP6 simulations are equilibrium experiments, in which the boundary conditions and forcings are constant from one year to another. The experimental set-up for each simulation is based on the DECK pre-industrial experiment (Eyring et al, 2015); the forcings and boundary conditions for the DECK pre-industrial experiments are modified to obtain

5    the forcings and boundary conditions necessary for each PMIP4-CMIP6 palaeoclimate experiment (Table 2). No additional interactive component (such as vegetation or dust) should be included in the model unless it is already included in the DECK version because such changes would affect the global energetics (Braconnot and Kageyama, 2015) and therefore prevent rigorous analyses integrating across multiple time periods or MIPs (sections 4.2 and 4.3).

> **Table 2: summary of changes in boundary conditions w.r.t. *piControl* for each PMIP4-CMIP6 experiment**

For each experiment, the greenhouse gases and astronomical parameters should be modified from the DECK piControl experiment according to Table 2. In the following sections, we give more detail on the implementation

15    of the boundary conditions which require specific attention to ensure consistency withing CMIP6 and PMIP4.

**3.2 Implementation of ice sheets**

The mid-Pliocene and Last Glacial Maximum experiments require changes in ice sheets. This implies changes in ice sheet height, land surface type, seas level and hence land-sea mask, and ocean bathymetry (Figure 6). These changes in boundary conditions should be implemented as follows:

20    1. The land-sea mask should be implemented in the ocean and atmosphere/land surface models. This step is optional for the *midPlioceneEoi400* experiment, but mandatory for the *lgm*. It is important to check the newly glaciated areas in the *lgm* experiment to ensure that grid cells under the grounded ice sheets (e.g. in the Hudson Bay area and over present-day Barents-Kara seas) are not specified as ocean cells.

2. The ice sheet mask should be implemented in the atmosphere/land surface model.

25    3. Changes in topography should be implemented by adding the anomaly in topography provided on the PMIP4 and PlioMIP web sites (http://pmip4.lsce.ipsl.fr and http://geology.er.usgs.gov/egpsc/prism/7_pliomip2.html) web sites to the topography used for the piControl simulation. This may mean re-computing parameters based on topography, such as those used in gravity wave drag parameterisations, because of the difference in surface roughness between ice

30    sheets and non-glaciated terrain.

4. Changes in ocean bathymetry should be implemented, if this is feasible for a given model, by using the more detailed bathymetry provided with the ice-sheet reconstructions. For the *midPlioceneEoi400* experiment the alternative is to leave bathymetry unchanged (i.e. the same as in the *PiControl*). The alternative for the *lgm* experiment is to lower mean sea level by the amount consistent with the ice-sheet

35    reconstruction used. If the ocean model includes a parameterization of the impact of tides on ocean circulation, it is recommended to re-compute the parameters as a function of the new bathymetry and land-sea mask.

5. River pathways and basins should be adjusted so that fresh water is conserved at the Earth's surface and rivers reach the ocean. This is particularly important given the large lowering of sea level in the *lgm*

[Figure]

[Figure]

experiment. River routing files will be provided for the *lgm* on the PMIP web site (http://pmip4.lsce.ipsl.fr), and these indicate how to change the course of rivers in regions covered by ice sheets. For the *midPlioceneEoi400* experiment, rivers pathways remain unchanged from modern except where there are new land grid cells when rivers should be routed to the nearest ocean grid box or most appropriate river outflow point.

**Figure 6: Changes in boundary conditions related to changes in ice sheets for the *midPliocene-eoi400* (top) and *lgm* (middle: ICE-6G_C and bottom: GLAC-1D) experiments. Coastlines for palaeo-period shown as brown contours. Ice sheet boundaries for each period shown as red contour. Bright shading: changes in altitude over regions covered by ice sheets during the considered palaeo-period. Faded shading: changes in altitude over ice-free regions.**

Some ice-sheet related changes must be implemented in the initial conditions:

- This atmospheric mass must be the same as today. For some models, this means that the initial surface pressure field has to be adjusted to the change in surface elevation.

- The mean ocean salinity has to be increased by +1 PSU everywhere at the beginning of the *lgm* simulation, to account for the lowering of sea level. Alkalinity also needs to be adjusted if an ocean biogeochemistry model is used.

**3.3 Vegetation and land use**

Palaeoenvironmental records show that natural vegetation patterns during each of the PMIP4-CMIP6 period

20   were different from today. However, in order to ensure comparability between past, present and future climate simulations, the PMIP4-CMIP6 palaeoclimate simulations should follow the same protocol as the DECK and historical simulations. If the DECK and historical simulations use dynamic vegetation, then the PMIP4-CMIP6 palaeoclimate simulations should also. If the DECK and historical simulations use prescribed modern vegetation, then modern vegetation should be prescribed in the PMIP4-CMIP6 palaeoclimate simulations. The only

25   exception to this is the *midPlioceneEoi400* experiment, where models which use prescribed modern vegetation in the DECK and historical simulations should use mid-Pliocene vegetation (Haywood et al., 2016) for their Pliocene simulation. Simulations to examine the impact of vegetation changes during other periods would be of interest, and could be evaluated using palaeodata. These could be made using prescribed vegetation changes, by running a model off line to compute vegetation patterns compatible with a past climate state, or by running

30   additional simulations with a non-standard version of the model with dynamic vegetation. Sensitivity experiments such as these will likely be run within PMIP4 but are not part of the PMIP4- CMIP6 experiments.

Land-use changes have to be implemented for the *past1000* simulation in the same manner as for the *historical* simulation (Hurtt et al., in prep.), using the land-use forcing provided by the Land Use Model Intercomparison

35   Project and the CMIP6 Land Use Harmonization dataset (https://cmip.ucar.edu/lumip; Hurtt et al., in prep.; Jungclaus et al., in prep.). This data set is derived from the HYDE3.2 (Klein Goldewijk et al., in prep.) estimates of the area of cropland, managed pasture, rangeland, urban, and irrigated land. Different crop types are treated separately and estimates of wood harvest are also provided.

[Figure]

[Figure]

**3.4 Natural aerosols**

**3.4.1 Mineral Dust**

Natural aerosols show large variations on glacial-interglacial time scales, with glacial climates having higher dust loadings than interglacial climates (Kohfeld and Harrison, 2001; Maher et al, 2010). Dust emissions from

5 northern Africa were significantly reduced during the MH (McGee et al., 2013). As is the case with vegetation, the treatment of dust in the *midHolocene* and *lgm* simulations should parallel the treatment in the *piControl*. However, some of the models in CMIP6 include representations of interactive dust. For those models, maps of soil erodibility, accounting for changes in the extension of possible dust sources, will be provided from recent simulations (Albani et al, 2014, 2015; Hopcroft et al, 2015) for the pre-industrial, mid-Holocene and the LGM

10 periods. Dust anomalies/ratios compared to the pre-industrial background should be used, for consistency with the DECK *piControl* simulation. As there have been instances of runaway climate-vegetation-dust feedback, leading to unrealistically cold LGM climates (Hopcroft and Valdes, 2015), it is advisable to test model behaviour before running the *lgm* simulation. To allow experiments with prescribed dust changes, a three-dimensional monthly climatology of dust atmospheric mass concentrations will be provided for the pre-industrial, MH, and

15 LGM based on two different modeling studies (Albani et al., 2014, 2015, 2016, Hopcroft et al., 2015). Additional dust-related fields (dust emission flux, dust load, dust aerosol optical thickness, short- and long-wave, surface and top of the atmosphere dust radiative forcing) will also be available from these simulations. Implementation should follow the same procedure as for the historical run (Albani et al, 2014, 2015). Since dust plays an important role in ocean biogeochemistry (e.g. Kohfeld et al, 2005), three dust maps will be provided.

20 Two of these are consistent with the climatologies of dust atmospheric mass concentrations; the other is primarily derived from observations (Lambert et al., 2015).
* * *
**Figure 7: Maps of dust deposition (g m-2 a-1) simulated with the Community Earth System Model for the a. PI (Albani et al., 2016), b. Mid-Holocene (Albani et al., 2015), and c. LGM (Albani et al., 2014). Maps of dust deposition**
25 **(g m-2 a-1) for the LGM d. simulated with the Hadley Centre Global Environment Model 2-Atmosphere (Hopcroft et al, 2015), and reconstructed from a global interpolation of paleodust data (Lambert et al., 2015).**
* * *
**3.4.2 Volcanoes and stratospheric aerosols**

The *past1000* experiment includes changes in volcanic aerosols, although these are not included in other PMIP4-CMIP6 experiments. The estimates of sulphur injections are derived from a recent compilation of synchronized

30 Antarctic and Arctic ice core records, which provides an improved history of the timing and magnitude of eruptions over the last 2500 years (Sigl et al. 2013). Ice core sulphate fluxes are translated into a time series of stratospheric sulphur injection via linear scaling (similar to Gao et al., 2008) and by matching the ice-core signals to historically confirmed eruptions. Unidentified eruptions are assigned as tropical when there are matching northern and southern hemisphere signals, signals only registered in the northern or southern

35 hemisphere are considered to be extratropical in origin. Modeling groups using interactive aerosol modules and sulphur injections in their historical simulations will follow the same method for the *past1000* experiment and use sulphur injection estimates directly. However, estimates of aerosol radiative properties as a function of latitude, height, and wavelength will be provided for other modelling groups using the Easy Volcanic Aerosol (EVA) module (Toohey et al., 2016), which is a parameterized three-box model of stratospheric transport that

40 uses simple scaling relationships to derive mid-visible aerosol optical depth (AOD) and aerosol effective radius

($r_{eff}$) from stratospheric sulphate mass. EVA uses model-specific information (grid, wave-length distribution) to produce annual volcanic aerosol forcing files for wavelength dependent aerosol extinction (EXT), single scattering albedo (SSA) and scattering asymmetry factor (ASY) as function of time, latitude, height and wave length. There are uncertainties associated with this approach, so additional sensitivity experiments to assess the impacts of these uncertainties on the *past1000* simulations will be made as part of the PMIP4 (see Jungclaus et al., in prep.).

**3.5 Spin-up and duration of experiments**

The data stored in the CMIP6 database should be representative of the equilibrium climates of the mid-Holocene, Last Glacial Maximum, Last Interglacial and mid-Pliocene Warm period, and of the transient evolution of climate between 850-1850 CE for the *past1000* simulations. Spin-up procedures will differ for different models and time periods, but the spin up should be long enough to avoid significant drift in the analysed data. Initial conditions can be taken from an existing simulation. A minimum of 100 years output is required for the equilibrium simulations but, given the increasing interest in analysing multi-decadal variability (e.g. Wittenberg, 2009), modelling groups are encouraged to provide outputs for a longer period of 500 years.

**3.6 Documentation**

Detailed documentation of the PMIP4-CMIP6 simulations is required. This should include:
- a description of the model and its components;
- information about the boundary conditions used, particularly when alternatives are allowed (Table 2);
- information on the implementation of boundary conditions and forcings. Figures showing the land-sea mask, land-ice mask, and topography as implemented in a given model are useful for the *lgm* and *midPliocene-eoi400* experiments, while figures showing insolation are particularly important for the *midHolocene* and *lig127k* experiments. Check lists for the implementation of simulations are provided in the PMIP4 papers providing detailed information for each experiment (*midHolocene*: Otto-Bliesner et al, 2016; *lgm;* Kageyama et al, 2016; past1000: Jungclaus et al, 2016; *lig127k*: Otto-Bliesner et al, 2016; *midPliocene-eoi400*: Haywood et al, 2016);
- information about the initial conditions and spin-up technique used. A measure of the changes in key variables (e.g. globally averaged 2m temperatures, sea-surface temperatures, bottom ocean temperatures, top-of-the-atmosphere radiative fluxes) should be provided in order to assess remaining drift.

Documentation should be provided via the ESDOC website and tools provided by CMIP6 (http://es-doc.org/) to facilitate communication with other CMIP6 MIPs. This documentation should also be provided on the PMIP4 website to facilitate linkages with non-CMIP6 simulations to be carried out in PMIP4. A PMIP4 special issue, shared between *Geoscientific Model Development* and *Climate of the Past*, will provide a further opportunity for modelling groups to document specific aspects of their simulations.

[Figure]

[Figure]

**4. Plan of Analyses**

The compatibility of past, historical and future climate simulations, through the use of seamless forcings and identical model versions, will allow benchmarking based on extensive syntheses of palaeoclimate data to be applied to models used for future projections. Planned analyses of the PMIP4-CMIP6 palaeoclimate simulations will make full use of the fact that modelling groups must also run the *piControl, historical* and *abrupt4xCO2* DECK experiments, by focusing on analyses that link past and future climates. The *piControl* and the *historical* simulations provide two alternative reference states for palaeoclimate simulations. Existing palaeoclimate reconstructions have used different modern reference states, and this has been shown to have an impact on the magnitude of reconstructed changes (e.g. Hessler et al., 2014). Comparisons of the simulated *piControl* and the *historical* climates will provide a way of quantifying this source of reconstruction uncertainty. Furthermore, links established with other CMIP6 MIPs (Section 4.3 and Table 3) will make it possible to capitalise on their analyses to improve understanding of specific aspects past climates and vice versa.

**4.2 Making use of PMIP4-CMIP6 multi time period**

Systematic benchmarking of each of the PMIP4-CMIP6 simulations will be a major aspect of the planned multi-period approach. This will require the development of new data syntheses, assessments of the regional-scale consistency of different sources of information, as well as the use of new tools that simulate the palaeoclimate sensors explicitly. Forward modelling of specific palaeoenvironmental records provides a way to quantify uncertainties in the climate reconstructions used for benchmarking. The ensemble of metrics developed in PMIP3-CMIP5 (e.g. Harrison et al. 2013) will be expanded to include more process-oriented metrics. Multi-period analyses will be particularly helpful for analyses of the hydrological cycle and the monsoons, including the how changes in land hydrology affect freshwater inputs to the ocean and water mass properties. Multi-period analyses will also help to address the role of vegetation feedbacks, particularly given the ambiguity as to whether these feedbacks are reproduced appropriately in simulations of the mid-Holocene.

There are many aspects of the climate system which are difficult to measure directly, and which are therefore difficult to evaluate using traditional methods. The "emergent constraint" (e.g. Sherwood et al., 2014) concept, which is based on identifying a relationship to a more easily measurable variable, has been successfully used by the carbon-cycle and modern climate communities and holds great potential for the analysis of palaeoclimate simulations. This could be particularly valuable to examine the realism of cloud feedbacks in the simulations or the contribution of seasonal climate changes to hydrological budgets.

Joint analysis of multiple paleoclimate simulations and climate reconstructions from different archives will be used to address the issue of climate sensitivity (*sensu stricto*) and earth-system sensitivity (PALEOSENS Project Members, 2012). The relationship between radiative forcing and global temperature is not straightforward, (Crucifix 2006, Yoshimori et al, 2011), partly because the nature of the forcing that drives the Earth to a cold climate differ from those that drive it into a warmer state. Nevertheless, estimates of climate sensitivity based on past climate states provide a starting point to establish the bounds of climate sensitivity to $CO_2$ doubling (Hargreaves 2012). The multi-period approach will bring new constraints to this analysis. Additional constraints

can be obtained by using perturbed-physics experiments, in which different members differ by the values of the parameters (Annan et al., 2005, Yoshimori et al, 2011). The 'perturbed forcing' approach (Bounceur et el., 2015, Araya-Melo 2015), using sensitivity experiments carried out in PMIP4, could provide a way to chart the sensitivity of the climate system in a multi-dimensional space of forcing conditions.

Multi-period analyses will also be useful to understand the relationship between mean climate state and modes of natural variability (e.g. Saint-Lu et al, 2015; Liu et al 2014). Future changes in modes of climate variability, such as ENSO, are poorly constrained (Christiansen et al., 2013) because model projections are insufficiently long to provide robust statistics for low frequency (multidecadal and longer) variations. Robust statistics of ENSO

10 changes have been derived through critical analysis of high-resolution palaeo-records (Emile-Geay et al., 2016). The equilibrium palaeoclimate experiments in PMIP4-CMIP6 provide an opportunity to sample simulations for long enough, at least 250 years, to obtain robust estimates of ENSO changes (Stevenson et al, 2010) and analyses of multiple long simulations with different forcings should provide a better understanding of changes in ENSO behaviour (Zheng et al. 2008, An et al. 2014) and to determine whether state-of-the-art climate models

15 underestimate low frequency noise (Laepple and Huybers, 2014). The PMIP Paleovariability Working Group will develop diagnostics for climate variability (Philips et al, 2014) to be applied to all the PMIP4-CMIP6 simulations. Analyses will focus on how models reproduce the relationship between changes in seasonality and interannual variability (Emile-Geay et al. 2016), the diversity of El-Niño events (Capotondi et al. 2015; Karamperidou et al. 2015, Luan et al 2015), and the stability of teleconnections within the climate system (e.g.

20 Gallant et al., 2013; Batehup et al., 2015).

**4.3 Interactions with other CMIP6 MIPs and the WCRP Grand Challenges**

Interactions between PMIP and other CMIP6 MIPs have mutual benefits: PMIP provides simulations of large climate changes that have occurred in the past and evaluation tools capitalizing on extensive data syntheses, while other MIPs will employ diagnostics and analyses which will be useful for analyzing the PMIP4

25 experiments. This is the case of AerChemMIP for the aerosol forcings, SIMIP (Notz et al, 2016) and OMIP (Griffies et al, 2016) for the sea-ice and ocean components, LS3MIP (van den Hurk, 2016) for the land surface, C4MIP (Jones et al, 2016) for the carbon cycle, ISMIP for ice sheets, and CFMIP for the cloud forcing and feedback analyses. VolMIP (Zanchettin et al, 2016) and LUMIP (Lawrence et al, 2016) analytical tools will be relevant for the analyses of the impacts of volcanic and land use forcings in the *past1000* simulation. The

30 *past1000* experiment also offers a long time series perturbed by natural forcings and observed land use changes for detection and attribution exercises and is therefore relevant for DAMIP (Gillett et al, 2016). We have ensured that all the outputs necessary for the application of common diagnostics across PMIP and other CMIP6 MIPs will be available (see section 4.4).

35 PMIP has already developed strong links with several other CMIP6 MIPs (Table 3). CFMIP includes an idealized experiment mimicking the *lgm* simulation: *AMIPminus4K* is an atmosphere-only experiment in which the sea-surface temperatures are uniformly lowered by 4K is a mirror of the *AMIP4K* experiment in which sea-surface temperatures are increased by 4K. These experiments allow investigations of cloud feedbacks and associated circulation changes in a colder versus a warmer world and this will assist in disentangling the

processes at work in the *lgm* climate. Some MIPs have designed experiments based on PMIP data, including VolMIP for the study of the impact of large past volcanic eruptions and ISMIP6 for the impact of the last interglacial climate on the Greenland ice sheet. Links with CFMIP and ISMIP6 mean that PMIP will also contribute to the WCRP Grand Challenges "Clouds, Circulation and Climate Sensitivity" and "Cryosphere and

5    Sea Level" respectively. PMIP will also provide input to the WCRP Grand Challenge on "Regional Climate Information", through a focus on evaluating the mechanisms of regional climate change in the past.
* * *
Table 3: interactions of PMIP with other CMIP6 MIPs
* * *
10    **4.4 Implications: required variables for the PMIP4-CMIP6 database**

The list of variables required to analyse the PMIP4-CMIP6 palaeoclimate experiments (https://wiki.lsce.ipsl.fr/pmip3/doku.php/pmip3:wg:db:cmip6request) reflects plans for multi-time period analyses and for interactions with other CMIP6 MIPs. We have included pertinent variables from the data requests of other MIPs, including the CFMIP specific diagnostics on cloud forcing, land surface, snow, ocean,

15    sea ice, aerosol, carbon cycle and ice sheet variables from LS3MIP, OMIP, SIMIP, AerChemMIP, C4MIP, and ISMIP6 respectively. Some of these variables are also required to diagnose how climate signals are recorded by palaeoclimatic sensors via models of e.g. tree growth (Li et al., 2014), vegetation dynamics (Prentice et al., 2011) or marine micro-flora/fauna (e.g. planktonic foraminifera: Lombard et al, 2011, Kageyama et al, 2013). The only set of variables defined specifically for PMIP are those describing oxygen isotopes in the climate

20    system. Isotopes are widely used for palaeoclimatic reconstruction and are explicitly simulated in several models.

We have asked that average annual cycles of key variables are included in the PMIP4-CMIP6 data request for equilibrium simulations, as these proved exceptionally useful for analyses in PMIP3-CMIP5. Daily values of

25    some variables are required for analyzing simulations with large changes in astronomical parameters (*midHolocene* and *lig127k*), as these changes result in modifications of the duration of each month of the year (Braconnot and Joussaume 1997). Modifications to month length are not usually taken into account in the model output post-treatment procedures. Daily values are also useful for running regional models. It is important to test the use of regional models for climate model projections at the regional scale. These models are also used to

30    produce fine-scale palaeoclimate scenarios for use by the impact community, for example to study past climate impacts on biodiversity via ecological niche modelling.

**5. Conclusions**

PMIP4-CMIP6 simulations provide a framework to compare current and future anthropogenic climate change with past natural variations of the Earth's climate. PMIP4-CMIP6 is a unique opportunity to simulate past

35    climates with exactly the same models as used for simulations of the future. This approach is only valid if the model versions and implementation of boundary conditions are consistent for all periods, and if these boundary conditions are seamless for overlapping periods.

[Figure]

[Figure]

PMIP4-CMIP6 simulations are important in terms of model evaluation for climate states significantly different from the present and historical climates. We have chosen climatic periods well documented by paleoclimate and paleoenvironmental records, with climate and environmental changes relevant for the study and projections of future climate changes: the mid-Holocene, the Last Glacial Maximum, which are the periods over which PMIP has developed its largest experiments since its beginning, together with the last millennium before the industrial era (850-1850), the last interglacial and the mid-Pliocene Warm Periods.

The PMIP community anticipates major benefits from analysis techniques developed by the other CMIP6 MIPs, in particular in terms of learning about the processes of past climate changes in response to forcings (e.g. greenhouse gases, astronomical parameters, ice sheet and sea level changes) as well as feedbacks (e.g. clouds, ocean, sea-ice). Collaborations have already been developed with e.g. CFMIP, ISMIP6 and VolMIP, but the hope is to build additional collaborations with other CMIP6 MIPs. PMIP4-CMIP6 has the potential to be mutually beneficial for the paleoclimate and present/future climate scientists to learn about natural large climate changes and the mechanisms at work in the climate system for climates states as different from today as future climate is projected to be.

**Data availability**

[revised manuscript text omitted]

Kaplan, J.O, Krumhardt, K.M., Ellis, E.C., Ruddiman, W.F., Lemmen, C., and Klein Goldewijk, K., 2011. Holocene carbon emissions as a result of anthropogenic land cover change. The Holocene 21: 775-791.

Klein-Goldwijk, K., et al., 2013: Long-term historical rice estimates; new Holocene land use patterns from HYDE3.2. In preparation.

Kohfeld, K. E., Harrison, S. P.: How well can we simulate past climates? Evaluating the models using global palaeoenvironmental datasets, Quaternary Science Reviews, 19, 321-346, 2000.

[Figure]

[Figure]

[revised manuscript text omitted]
 et al., 2016: ??Historical greenhouse gases for CMIP6??. In prep. For CMIP6 Special Issue, Geosci. Model Dev. Discuss.

Miller, K. E., et al: High tide of the warm Pliocene: Implications of global sea level for Antarctic deglaciation, Geology, G32869, doi:10.1130/G32869.1, 2012.

Naish, T.R., Wilson, G.S.: Constraints on the amplitude of Mid-Pliocene (3.6–2.4Ma) eustatic sea-level fluctuations from the New Zealand shallow-marine sediment record. Philosophical Transactions of the Royal Society A: Mathematical, Physical and Engineering Sciences, 367(1886), 169-187, 2009.

Notz, D., Jahn, A., Holland, M., Hunke, E., Massonnet, F., Stroeve, J., Tremblay, B., and Vancoppenolle, M.: Sea Ice Model Intercomparison Project (SIMIP): Understanding sea ice through climate-model simulations, Geosci. Model Dev. Discuss., doi:10.5194/gmd-2016-67, in review, 2016.

Oakley, J. E. and O'Hagan, A.: Probabilistic sensitivity analysis of complex models: a Bayesian approach, Journal of the Royal Statistical Society: Series B (Statistical Methodology), 66, 751-769, doi:10.1111/j.1467-9868.2004.05304.x, 2004.

Otto-Bliesner, B., E. Brady, J. Fasullo, A. Jahn, L. Landrum, S. Stevenson, N. Rosenbloom, A. Mai, and G. Strand: Climate Variability and Change since 850 C.E.: An Ensemble Approach with the Community Earth System Model (CESM). Bull. Amer. Meteor. Soc. doi:10.1175/BAMS-D-14-00233.1, in press, 2016.

Otto-Bliesner, B. E. et al: PMIP4 Last Interglacial and Mid-Holocene experiments, to be submitted May 2016.

[revised manuscript text omitted]

| Name of MIP | Themes of interactions |
|---|---|
| CF-MIP | dedicated common idealized sensitivity experiment to be run in aquaplanet set up, *AMIPminus4K*, to be co-analysed in CF-MIP and PMIP. |
| ISMIP6 | Assessment of the climate and cryosphere interactions and the sea level changes associated with large ice sheets. In particular, the *lig127k* simulation will be used to force ice sheet models in ISMIP6. Additional experiments co-designed by the PMIP and ISMIP groups are foreseen outside the CMIP6 exercise: transient interglacial experiments, with climate model output forcing an ice sheet model, and coupled climate-ice sheet experiments. |
| OMIP | Mutual assessment of the role of the ocean in low-frequency variability, e.g. multi-decadal changes in ocean heat content or heat transport. Provide initial conditions for the ocean including long-term forcing history. |
| SIMIP | Assessment of role of sea-ice in climate changes |
| AerChemMIP | Assessment of role of aerosols in climate changes, very helpful since this is a new aspect in PMIP experiments for the midHolocene, last interglacial and LGM |
| LS3MIP | Assessment of role of land surface processes in climate changes. |
| C4MIP | Assessment of carbon-cycle evolution and feedbacks between sub-components of the Earth System. Evaluation of paleo reconstructions of carbon storage. |
| LUMIP | Analysis of climate changes associated with Land Use changes (*past1000* experiment) |
| VolMIP | Analysis of specific volcanic events very useful for critical analysis of *past1000* simulations. VolMIP would systematically assess uncertainties in the climate response to volcanic forcing, whereas *past1000* simulations describe the climate response to volcanic forcing in long transient simulations where related uncertainties are due to chosen input data for volcanic forcing: mutual assessment of forced response. |
| DAMIP | past1000 simulations provide long-term reference background including natural climate variability for detection and attribution. |

5 **Table 3: interactions of PMIP with other CMIP6 MIPs**

[Figure]

[Figure]

5    FIGURE CAPTIONS

Figure 1: Context of the PMIP4 experiments (from left to right: MPWP, Mid-Pliocene Warm Period; LIG, last interglacial; LGM, last glacial maximum; MH, mid-Holocene; LM, last millennium; H, CMIP6 historical simulation): (a)-(d) insolation anomalies (differences from 1950 CE), for July at 65°N, calculated using the
10   programs of Laskar et al. (2004, panel (a)) and Berger (1978, panels (b)-(d)); (e) $\delta^{18}O$ (magenta, Lisiecki and Raymo, 2005, scale at left), and sea level (blue line, Rohling et al., 2014; blue shading, a density plot of eleven Mid-Pliocene sea level estimates (Dowsett and Cronin 1990; Wardlaw and Quinn, 1991; Krantz, 1991; Raymo et al., 2009; Dwyer and Chandler, 2009; Naish and Wilson, 2009; Masson-Delmotte et al., 2013; Rohling et al., 2014; Dowsett et al., 2016) scale at right); (f) and (g) $\delta^{18}O$ (magenta, Lisiecki and Raymo, 2005, $\delta^{18}O$ scale at
15   left), and sea level (blue dots, with light-blue 2.5, 25, 75 and 97.5 percentile bootstrap confidence intervals, Spratt and Lisiecki, 2015; blue rectangle, LIG high-stand range, Dutton et al., 2015; dark blue lines, Lambeck et al., 2014, sea-level scale at right on panel (g)), (h) sea level (Kopp, et al., 2016, scale at right); (i) $CO_2$ for the interval 3.0-3.3 Ma shown as a density plot of eight Mid-Pliocene estimates (Raymo et al., 1996; Stap et al., 2016; Pagani et al., 2010; Seki et al., 2010; Tripati et al., 2009; Bartoli et al., 2011; Seki et al., 2010; Kurschner
20   et al., 1996); (j) and (k) $CO_2$ measurements (Bereiter et al., 2015, scale at left); (l) $CO_2$ measurements (Schmidt et al, 2011, scale at right); (m) and (n) $CH_4$ measurements (Loulergue et al., 2008, scale at left); (o) $CH_4$ measurements (Schmidt et al, 2011, scale at right); (p) volcanic radiative forcing (Schmidt et al., 2012, scale at right); (q) total solar irradiance (Schmidt et al., 2012, scale at right).

25   Figure 2. Data-model comparisons in PMIP2 and CMIP5/PMIP3: (a) Land-ocean contrast in past, present and projected future climates. The black dots are the simulated long-term mean differences (*experiment − piControl*) in the relative warming/cooling over global land and global ocean. The red crosses show simulated changes where the model output has been sampled only at the locations for which there are temperature reconstructions for the *lgm*, *midHolocene* and *historical* (post-1850 CE) CMIP5 simulations. Area averages of palaeoclimate
30   data are shown by bold blue crosses, with reconstruction uncertainties indicated by the finer lines. The regression line (magenta) shows that land-ocean contrasts are maintained across different climate states and are also consistent with palaeoclimatic data. (b) Boxplots of reconstructions based on fossil-pollen data (gray, Bartlein et al. 2011) and simulations (at the locations of the data) for the difference in mean annual precipitation (MAP) for the mid-Holocene (relative to present) in northern Africa (20°W-30°E; 5-30°N). The comparison
35   shows that although all models simulated wetter-that-present conditions in northern Africa for the mid-Holocene, they systematically underestimated the magnitude of the precipitation difference.

Figure 3: Color lines: temperatures anomalies (w.r.t. the 1500-1850 CE average) simulated by PMIP3-CMIP5 models for the last millennium and historical periods, averaged for the northern (l.h.s) and southern (r.h.s)
40   hemisphere. Grey shading: uncertainty envelope of available reconstructions. All series are filtered using a 31-point moving average filter. Adapted from Fernandez-Donado, 2015.

Figure 4: Figure 5.6 from Chapter 5 of the IPCC AR5 WGI report (Masson-Delmotte et al., 2013, page 408). Changes in surface temperature for the Last Interglacial (LIG) as reconstructed from data and simulated by an
45   ensemble of climate model experiments in response to orbital and well-mixed greenhouse gas (WMGHG) forcings. (a) Proxy data syntheses of annual surface temperature anomalies as published by Turney and Jones (2010) and McKay et al. (2011). McKay et al., (2011) calculated an annual anomaly for each record as the average sea surface temperature (SST) of the 5-kyr period centred on the warmest temperature between 135 ka and 118 ka and then subtracting the average SST of the late Holocene (last 5 kyr). Turney and Jones (2010)
50   calculated the annual temperature anomalies relative to 1961–1990 by averaging the LIG temperature estimates

5    across the isotopic plateau in the marine and ice records and the period of maximum warmth in the terrestrial records (assuming globally synchronous terrestrial warmth). (b) Multi-model average of annual surface air temperature anomalies simulated for the LIG computed with respect to preindustrial. The results for the LIG are obtained from 16 simulations for 128 to 125 ka conducted by 13 modelling groups (Lunt et al., 2013). (c) Seasonal SST anomalies. Multi-model zonal averages are shown as solid line with shaded bands indicating 2

10   standard deviations. Plotted values are the respective seasonal multi-mean global average. Symbols are individual proxy records of seasonal SST anomalies from McKay et al. (2011). (d) Seasonal terrestrial surface temperature anomalies (SAT). As in (c) but with symbols representing terrestrial proxy records as compiled from published literature (Table 5.A.5). Observed seasonal terrestrial anomalies larger than 10°C or less than –6°C are not shown. In (c) and (d) JJA denotes June – July – August and DJF December – January – February,

15   respectively.

Figure 5: Figure 1 in Box 5.1 from Chapter 5 of the IPCC AR5 WGI report (Masson-Delmotte et al., 2013, page 397). Comparison of data and multi-model mean (MMM) simulations, for four periods of time, showing (a) sea surface temperature (SST) anomalies, (b) zonally averaged SST anomalies, (c) zonally averaged global (green)

20   and land (grey) surface air temperature (SAT) anomalies and (d) land SAT anomalies. The time periods are 2081–2100 for the Representative Concentration Pathway (RCP) 8.5 (top row), Last Glacial Maximum (LGM, second row), mid-Pliocene Warm Period (MPWP, third row) and Early Eocene Climatic Optimum (EECO, bottom row). Model temperature anomalies are calculated relative to the pre-industrial value of each model in the ensemble prior to calculating the MMM anomaly (a, d; colour shading). Zonal MMM gradients (b, c) are

25   plotted with a shaded band indicating 2 standard deviations. Site specific temperature anomalies estimated from proxy data are calculated relative to present site temperatures and are plotted (a, d) using the same colour scale as the model data, and a circle-size scaled to estimates of confidence. Proxy data compilations for the LGM are from Multiproxy Approach for the Reconstruction of the Glacial Ocean surface (MARGO) Project Members (2009) and Bartlein et al. (2011), for the MPWP are from Dowsett et al. (2012), Salzmann et al. (2008) and

30   Haywood et al. (2013) and for the EECO are from Hollis et al. (2012) and Lunt et al. (2012). Model ensemble simulations for 2081–2100 are from the CMIP5 ensemble using RCP 8.5, for the LGM are seven Paleoclimate Modelling Intercomparison Project Phase III (PMIP3) and Coupled Model Intercomparison Project Phase 5 (CMIP5) models, for the Pliocene are from Haywood et al., (2013), and for the EECO are after Lunt et al. (2012). [ Note: permission has been sought to use the third line of this figure only (i.e. the MPWP results). Until

35   this permission is received, we follow the IPCC rules for using the figures from the fifth assessment report. ]

Figure 6: Changes in boundary conditions related to changes in ice sheets for the midPliocene-eoi400 (top) and lgm (middle: ICE-6G_C and bottom: GLAC-1D) experiments. Coastlines for palaeo-period shown as brown contours. Ice sheet boundaries for each period shown as red contour. Bright shading: changes in altitude over

40   regions covered by ice sheets during the considered palaeo-period. Faded shading: changes in altitude over ice-free regions.

Figure 7: Maps of dust deposition (g m-2 a-1) simulated with the Community Earth System Model for the a. PI (Albani et al., 2016), b. Mid-Holocene (Albani et al., 2015), and c. LGM (Albani et al., 2014). Maps of dust

45   deposition (g m-2 a-1) for the LGM d. simulated with the Hadley Centre Global Environment Model 2-Atmosphere (Hopcroft et al, 2015), and reconstructed from a global interpolation of paleodust data (Lambert et al., 2015).

[Figure]

[Figure]

[Figure]

Figure 1

[Figure]

[Figure]

Figure 2

[Figure]

[Figure]

Figure 3

[Figure]

[Figure]

Figure 4

[Figure]

[Figure]

[Figure]

[Figure]

Figure 6

[Figure]

[Figure]

[Figure]

Figure 7

---

## Short Comment (SC1) · 2 Jun 2016

This paper nicely lays out the plan for PMIP4, which is part of CMIP6. It describes the history and rationale for this project, then reviews the experiments which cover five periods in recent Earth history and expand on those of previous PMIPs. The protocols for these experiments are described, a few potential pitfalls are noted for modellers to beware of, and useful links to other MIPs are noted. I think the project sounds exciting and hope that plenty of modelling centres are able to participate.

This article would benefit from some very minor revisions to improve clarity, but otherwise I am happy with it and don't see any major omissions. The minor revisions to suggest are:

[Figure]

6, 18-29: Please mention (as I gather later though am still not 100% sure) that all of these experiments are "time-slice" experiments, i.e., the model is run to statistical equilibrium with time-invariant forcing specified based on the quoted point in time to give a representative mean state, except for the last millennium which is transient and includes time-varying forcings such as volcanic eruptions.

8, 19: The word "observed" should be reserved for the instrumental period with real observations — can we instead say "proxy-estimated" (or similar)?

8, 29: By "mean values" do you mean global means? Time averages? Changes in global mean relative to modern?

10,40 By "trends" do you mean differences (relative to modern)?

11, 36: please fix error message

11,34: suggest "current" rather than "modern" (which can mean many things…from a palaeo perspective preindustrial could be viewed as modern)

17, 36-37: I don't think the polarity of the forcing is the real problem, but rather, the fact that some radiative forcing agents produce larger responses per unit global-average power input than do others, and/or provoke "rapid adjustments" to the forcing that are unrelated to global-mean warming or cooling. This is due to the spatial pattern of the forcing. Given the fact that a major goal here is to test model responses to forcings, and given that past forcings are different from the dominant ones we worry about for the future (greenhouse gases), this topic may deserve a bit more discussion.

Table 2: some of the cells in the table are blank, and I am not sure what this means. Suggest every box should say something (even if it is "see text")

Figure 1. I found this figure confusing; maybe you are trying to cram too much into one panel. The caption refers to panel labels (a, b, …) but there aren't any. It is hard to figure out what each curve as and which axis refers to what (especially when there is one on each side, or where it switches from right side to left going from one column to

another). I would prefer the axes to be individually labelled with the quantity and units, or give the quantity in the title or above the curve and the units on the axis so we know we are looking at the right one. It is not at all obvious that they grey banded stripe is meant to represent the possible range of CO2 in the left column; nothing wrong with showing it this way but please make clearer what everything is!

Figure 2a. I presume there are black dots hiding underneath the red ones for mid-Holocene and Historical? To help avoid confusion perhaps the caption could mention this.

Figure 2b. Please give units! Also, please explain in the caption what "OAV" and "OAC" refer to.

Figure 3. Please be consistent in labelling the panels (they are given (a,b) in the figure but you say lhs and rhs in the caption). Is this for land only, or land+ocean? What is STSI and ssTSI? Does 31 points mean 31 years? (give the time width of the smoothing window rather than the number of points)

Figure 4. It doesn't appear to me that the proxy data are able to tell us anything about the seasonal cycle, given that the differences are small compared to the scatter — so is it worth including the two right panels in a review paper on PMIP? Especially since the figure is reproduced from another source so anyone who really wants to see the seasonal results can find them. I see later you are already requesting permission to use a portion of another IPCC figure, I'd suggest making a similar request here.

Figure 7. Please spell out "preindustrial" rather than PI since PI is not one of the study time periods and you haven't used this acronym much.

———————————————

---

## Referee Comment (RC1) · Anonymous Referee #1 · 12 Jun 2016

There is little doubt that PMIP made significant contributions in assessing the role of different forcings on past climate changes. The strength of a MIP comes of course from the clear definition of boundary conditions, forcings etc. Hence, it is absolutely useful to describe these as clearly as possible in the peer-reviewed literature. Having said this, I am struggling with the purpose of this ms. Large parts read like an (un-convincing) attempt to justify PMIP4. Instead of repeating what was done in PMIP 3, the ms. would be much more convincing if the authors could outline which insights into climate processes were gained that would have been impossible without PMIP (using a few examples). In my view, there is too much description of modeled changes and matches/mismatches with proxy data but too little information on real insights into

climate processes - especially on processes that are of relevance outside the paleoclimate community.

My understanding is that the ms. should serve as an entry point to a series of more specialized descriptions of the experimental setup for the individual time slices. While such a publication strategy seems very useful, it also means that repetition can and should be avoided. In my view this overarching ms. should have a clear focus on what is now sect. 3 as well as on common issues among the experiments. Here, I am surprised that some important aspects are only mentioned in passing:

- River routing: the recommendations are quite vague for the LIG and the Pliocene (recalling, for example, that the modern river-system draining into the Arctic developed to a large part after the LGM).

- Which plans exist to assess the results from LGM experiments using different ice-sheet configurations?

- Spin up: how is an insignificant trend defined in the framework of PMIP?

- I was surprised to read nothing about the calendar problem [Joussaume and Braconnot, 1997], that featured high in earlier cycles of PMIP.

While I am generally very much in favor of the PMIP community to publish their experimental strategy, the current ms. needs considerable re-writing to become a useful contribution. I would suggest to drastically reduce the length and to focus it on new aspects:

- Sections 1 and 2 contain almost no new information and should be largely replaced by a concise summary of dynamical insights gained from earlier PMIP phases (see above)

- Figs. 2-5 were published elsewhere and should be removed

- Focus should be on what is now sections 3 and 4.

ᴄᴄ2

[Figure]

Finally, someone out the large group of authors should read the ms. from beginning to end to ensure that the wording/style is consistent (incl. the ref. list; "et al" vs. et al."; paleo vs. palaeo) and that geological ages are correct and consistent among the ms. (mPWP 3.2 Ma vs. 3.3-3 Ma; incorrect start of the Quaternary at 2.5 Ma).

---

## Referee Comment (RC3) · Anonymous Referee #3 · 2 Jul 2016

This manuscript details the contribution of the paleoclimate modeling community to the new Coupled Modelling Intercomparison Project. For this phase the PMIP community is planning to expand its contribution significantly, including 5 different periods or experiments. In order that as many modelling groups as possible participate in this effort a clear modelling setup. This manuscript gives an overview of the rationale and broadly outlines the experimental setup. Hence the importance of this paper is clear

General Comments

I find the manuscript a bit long without going into the specific details of each experiment. This is ok as it is mentioned that there will be special papers for each one of them. I would suggest shortening section 1, and section 2 could benefit for a clearer discussion

on current modelling gaps.

I have also personal comment. Having worked with the past1000 simulations, it would be very useful if all the modelling groups planning to run this experiment, would run it up to present. (850-2005 CE instead of stopping in 1850).

Figures: I a not familiar with the journal's policy in this respect, but it seems odd to me to use so many previously published figures. Also the quality and style of the figures is too diverse. For example, please use just one projections for all maps.

There is a lot of inconsistent naming of the experiments throughout the text. Please use either "midHolocene" or "MH". "LM" or "past1000" and so on.

Specific comments

Page3, Line 13: Missing comma after However. Or are you using "however = to whatever extent" ? Page3, Lines 14-15: used twice "well outside" Page 5, lin34: include years for "the last interglacial". Page 6, line 1: . . .of participating MODELLING groups. Page 6, line5: in this section maybe include the CMIP6 questions? Page6, line 7: is "How does the Earth System respond to forcing" question 1? How many questions are there? Please introduce a bit better. Page 6, line35: define DECK Page 7, line 20: change "interesting" for "valuable" Page 7, lines 41-42: please be consistent between title and text about MH and LGM. Page 10, line 24: question: is there a separate paper with specifics? Page10, line 34: needs a reference. Page 11, line 36: there is a typo. Page 13, line 27: delete "web sites". Its repeated. Page 17: line 22: delete "the" Figure 1: needs more information of the different panels. Very hard to understand at this point

---

## Short Comment (SC2) · 21 Jul 2016

Dear PMIP authors,

The CMIP Panel is undertaking a review of the CMIP6 GMD special issue papers to ensure a level of consistency in answering the key questions that were outlined in our request to submit a paper to all co-chairs of CMIP6-Endorsed MIPs. These questions are outline in the overview paper (Eyring et al, GMD, 2016) and the relevant section is summarised below:

'Each of the 21 CMIP6-Endorsed MIPs is described in a separate invited contribution to this Special Issue. These contributions will detail the goal of the MIP and the major sci-

[Figure]

entific gaps the MIP is addressing, and will specify what is new compared to CMIP5 and previous CMIP phases. The contributions will include a description of the experimental design and scientific justification of each of the experiments for Tier 1 (and possibly beyond), and will link the experiments and analysis to the DECK and CMIP6 historical simulations. They will additionally include an analysis plan to fully justify the resources used to produce the various requested variables, and if the analysis plan is to compare model results to observations, the contribution will highlight possible model diagnostics and performance metrics specifying whether the comparison entails any particular requirement for the simulations or outputs (e.g. the use of observational simulators). In addition, possible observations and reanalysis products for model evaluation are discussed and the MIPs are encouraged to help facilitate their use by contributing them to the obs4MIPs/ana4MIPs archives at the ESGF (see Section 3.3). In some MIPs additional forcings beyond those used in the DECK and CMIP6 historical simulations are required, and these are described in the respective contribution as well.'

We very much welcome the PMIP contribution and the detailing of the experimental design, analysis plan and diagnostic output that you currently cover in sections 3 and 4. We also welcome the strong links that PMIP has clearly forged with other CMIP6 MIPs and look forward to the joint analysis that you describe.

Additionally, we would like to see some more detail on some of the issues raised above, notably;

a. More discussion on the specific goals of PMIP4 in CMIP6 and what science gaps it is attempting to fill. You describe the 3 CMIP6 science questions and PMIP links to them in Section 1.3 and the links to the WCRP GCs in section 4.3, but it would be good to see some discussion on what PMIP4 is hoping to achieve that is new since PMIP3.

b. The description of the experimental design for each experiment is comprehensive and very useful. There are however, a worrying large number of papers 'to be submitted'. Is it clear that once this paper is published modelling groups will be able to rely on

it to provide a comprehensive start point for setting up their experiments?

c. A lot of focus in the analysis plan is given to the multi-time period analysis, but not all modelling centres will be contributing to all (or indeed in some cases more than 1) of the entry card/tier 1 experiments. Could more be said about analysis of the specific experiments and what new we will learn from these experiments since PMIP3-CMIP5 (higher resolution, better data, more ES components)?

d. You make the point that the comparison of these time periods to palaeodata is one of the key drivers but say very little about the observational data sources or whether these products will be made available to the community to facilitate comparison. In section 4.4 you describe the new metrics and forward modelling you request the models output. It would be good to document how these will be evaluated.

We hope you agree that some level of consistency across the MIP papers in this special issue is valuable and that the above suggestions can be accommodated in your paper.

Other comments:

e. There is a lack of consistency in the naming of the experiments e.g. notably the use of LM and past1000 somewhat interchangeably. Please clarify

f. In section 3.2, the implementation of the ice-sheets needs to be a bit clearer. For example do all points 2-5 refer to both midplioceneEoi400 and lgm?

With many thanks for your ongoing efforts in the CMIP6 process.

The CMIP Panel
* * *

---

## Editor Comment (EC1) · J. C. Hargreaves (Editor) · 21 Jul 2016

I think most things have been covered by the reviewers and the CMIP panel's comment, so please enjoy responding to their comments.

In common with one of the reviewers, I'm not enchanted by the history lesson. I understand that you want to draw people into considering paleoclimate, but I think the resulting length of the paper is more likely to turn people away.

However, the real problem for publication of this paper in GMD is the possibly incomplete protocols.

From Section 3: "The modified forcings and boundary conditions for each PMIP4-

CMIP6 palaeoclimate simulation are summarised in Table 2. The complete details of the experimental protocols are given in a series of companion papers: Otto-Bliesner et al for the midHolocene and lig127ka experiments, Kageyama et al for the lgm, Jung-claus et al for the past1000 and Haywood et al (2016) for the midPliocene-eoi400 experiment. These papers also explain how the boundary conditions for each period have been built and constitute key references for the experimental protocol for each of the PMIP4-CMIP6 simulations."

The problem, as highlighted by the CMIP panel, is that most of these papers are not published. I can't control what is going to be in those other papers - all I can do is make sure that this paper meets the peer review requirements. Therefore, for all experiments please include in this paper, "the complete details of the experimental protocols" for the PMIP4-CMIP6 experiments. You can leave the details on "how the boundary conditions for each period have been built" to the still to be submitted papers. You can also leave all alternate experiments that are within PMIP4 but outside CMIP6 to those other papers. Please make sure that Table 2 is edited so that it does not appear that essential details required for setting up the experiments are included in these unpublished papers. Basically, a modeller should be able to set up the PMIP4-CMIP6 runs using the information contained in this paper.

The alternative is that we put this paper hold until the other papers have passed through peer review. This could be workable as I understand that you intend to submit the other papers also to GMD. Even in that case, I would like to see the complete protocols for PMIP4-CMIP6 detailed here (citing the other papers as required), as trying to extract the CMIP6 protocols from the much more elaborate single-interval papers is likely to be a trying process.
* * *

---

## Editor Comment (EC2) · J. C. Hargreaves (Editor) · 6 Oct 2016

I'm also editing the DeepMIP paper, and they happen to have a rather prominently positioned section about data for comparison with the model output, which caused me to pay attention to it, and then criticise it as inadequate! I think that more or less the same criticism can be levelled at this PMIP4 manuscript. Section 4.2 touches on ways in which the models may be compared to data, but this is less important than outlining the datasets that are available and including specifics about any new datasets that are planned to emerge within the timeframe of the project. Please add this information for all the PMIP4 periods in the revised manuscript. It doesn't have to be a huge amount of material - the point is a practical one - to lead users of PMIP model output in the

right direction so they can discover the relevant datasets. I expect you would include citations in the revised section 4.2, but you might also wish to include some additional information (DOIs/weblinks etc) in the Data Availability section.

---

## Editor Comment (EC3) · J. C. Hargreaves (Editor) · 27 Oct 2016

In response to my previous comment which highlighted the problem of citing unsubmitted works, the authors elected to put this manuscript on hold until those manuscripts were submitted to GMD. Those manuscripts are now almost complete, and the authors have enabled me to see two out of the three of them for approval before submitting to GMD.

I find there is a real problem, which is that these two draft manuscripts both contain substantial discussion of the background and rationale behind the \*CMIP6\* PMIP4 protocols. These discussions are too entwined with the non-CMIP6 experiments within PMIP4 to be removed from those draft papers, and including the content within this

overview manuscript under consideration would be entirely impractical.

In terms of the peer review at GMD I find it unacceptable that we should in this overview manuscript be approving the protocols for the CMIP6 runs without first reading the rationale behind them!

All this suggests that the complete protocols should be peer reviewed within those other papers. Since those papers describe the rationale behind the CMIP6 protocols they must also be included in the CMIP6 special issue.

As for the present paper under consideration, a purely descriptive paper about PMIP does not fit into the peer review criteria and I have great difficulty accepting it as a standalone paper. However, if it were presented as the introductory part of a multi-part paper, I think it will be a useful contribution to the whole. This has led me to come up with a solution that solves all these problems, and I think it also produces a nice final product. I suggest that the papers be submitted as 4 parts of a multi-part paper. The titles should be made consistent with each other and include the Part number in the title. They do not have to be exactly this, but something like this would work...

PMIP4-CMIP6, the contribution of the Paleoclimate Modelling Intercomparison Project to CMIP6, Part 1: Introduction and(/or?) Overview

PMIP4-CMIP6, the contribution of the Paleoclimate Modelling Intercomparison Project to CMIP6, Part 2: Eemian and midHolocene

PMIP4-CMIP6, the contribution of the Paleoclimate Modelling Intercomparison Project to CMIP6, Part 3: The Last Millennium

PMIP4-CMIP6, the contribution of the Paleoclimate Modelling Intercomparison Project to CMIP6, Part 3: The Last Glacial Maximum

From the point of view of this paper, the authors are welcome to submit a revised version, but if outline protocols for the experiments are included then the final publication will need to wait until all the other three papers are accepted.

I appreciate the difficulties the authors have had with trying to write a single coherent paper as a contribution to the GMD CMIP6 special issue, but I think it was an intractable problem, because the resulting paper could not pass the GMD peer review criteria of including both the rationale and the protocols. In that context, I'd like to contrast this MIP with one of the others I edited. CFMIP have a huge number of experiments fully described within their paper, including tier 1 and tier 2 experiments. I think it is an elegant paper. However, CFMIP experiments are all highly idealised, and thus there is no great debate to be had in defining the protocols and they can be very simply described. PMIP, on the other hand is trying to model real and disparate intervals in the earth's climate history. This is an order of magnitude more complex, involving several different communities of scientists, and it so it is not appropriate to squeeze it all into a single paper. The multi-part paper will enable these communities to each take the responsibility for the defence of their own experiment protocols.

---

## Author Comment (AC2) · 14 Nov 2016

Reponse to reviews of "PMIP4-CMIP6: the contribution of the Paleoclimate Modelling Intercomparison Project to CMIP6" by Kageyama et al.

[ for clarity, we reproduce *the comments by the reviewers, editor and CMIP panel in blue/italic* and provide our answers in black ]

We thank the reviewers for their comments which helped focussing and clarifying the manuscript. We have attempted to reply to all their comments. In particular, we have:

- reduced sections 1 and 2 and removed figures 3 to 5 which were published elsewhere

- expanded sections 3 and 4, with a more complete protocol and a more complete analyses plan, including an overview of the  analyses plans for individual experiments.

We have also attempted to harmonise the text and to remove inconsistencies.

**Reponse to Anonymous Referee #1**

*There is little doubt that PMIP made significant contributions in assessing the role of different forcings on past climate changes. The strength of a MIP comes of course from the clear definition of boundary conditions, forcings etc. Hence, it is absolutely useful to describe these as clearly as possible in the peer-reviewed literature. Having said this, I am struggling with the purpose of this ms. Large parts read like an (unconvincing) attempt to justify PMIP4. Instead of repeating what was done in PMIP 3, the ms. would be much more convincing if the authors could outline which insights into climate processes were gained that would have been impossible without PMIP (using a few examples). In my view, there is too much description of modeled changes and matches/mismatches with proxy data but too little information on real insights into climate processes - especially on processes that are of relevance outside the paleoclimate community.*

We have shortened Sections 1 and 2 in agreement with this comment, removed Figures 3 to 5 which were describing previous results in terms of model-data comparisons and added examples of how PMIP can help and illustrate the processes at work in climate model responses to forcings. The available paleoclimatic reconstructions, which were also illustrated on these figures, are now the topic of a special section of the analyses plan (Section 4.1). We have attempted to complete the protocol wherever it lacked information, so that all necessary information to run the PMIP4 experiments for CMIP6 can be found in the present manuscript.

*My understanding is that the ms. should serve as an entry point to a series of more specialized descriptions of the experimental setup for the individual time slices. While such a publication strategy seems very useful, it also means that repetition can and should be avoided. In my view this overarching ms. should have a clear focus on what is now sect. 3 as well as on common issues among the experiments.*

This particular topic has been discussed between the CMIP Panel and the Editor. We have followed the CMIP Panel recommendation and added details on the protocols so that their description is complete. The complete justification of these protocols, though, would take too much space in the present manuscript and is developed in the companion papers on each period. These papers also detail the sensitivity experiments based on the PMIP4-CMIP6 experiment. They are being submitted

to GMD. The present manuscript will be held until acceptance of the companion paper and modified if necessary to guarantee consistence between all papers.

*Here, I am surprised that some important aspects are only mentioned in passing:*

*- River routing: the recommendations are quite vague for the LIG and the Pliocene (recalling, for example, that the modern river-system draining into the Arctic developed to a large part after the LGM).*

We have now devoted a section to river routing. This was previously in the section about implementing the ice sheets and it is true that there was no specification for periods other than the LGM. Our understanding, though, is that we are currently lacking data sets to take these changes in river routing into account for the LIG and mPWP experiments. We have therefore made the conservative choice of not requiring changes in river routing in those experiments. This could of course be the topic of additional sensitivity experiments, as we know this could be of importance for the oceanic circulation (e.g. Alkama et al, 2008, for the Last Glacial Maximum). This is now clearly stated in the manuscript (Section 3.3).

*- Which plans exist to assess the results from LGM experiments using different ice sheet configurations?*

It is the first time that we consider several possible ice sheets in our LGM experiments, and this is in acknowledgement that several approaches lead to significantly different reconstructions. It is therefore very important to assess these results and this is now better highlighted in our manuscript (Section 4.2). A first analysis will be to evaluate those simulations in comparison to paleo-climatic reconstructions and assess whether this comparison yields different results for the different ice sheet reconstructions. A second step will be to understand the differences between the climates simulated with the different reconstructions. This is why we encourage groups which can afford it to run several simulations with these different reconstructions. The "LGM ice sheets" working group has been set up to investigate these questions.

*- Spin up: how is an insignificant trend defined in the framework of PMIP?*

The criteria for defining a spun-up experiments are now quantified in section 3.6, with the same criteria as for PMIP2: "We recommend that the spin-up should be run until the trend in global mean sea-surface temperature is <0.05K per century and the Atlantic Meridional Overturning Circulation (AMOC) is stable; a parallel requirement for carbon-cycle models and/or models with dynamic vegetation is that the 100-year average global carbon uptake or release by the biosphere is <0.01 Pg C a$^{-1}$"

*- I was surprised to read nothing about the calendar problem [Joussaume and Braconnot,1997], that featured high in earlier cycles of PMIP.*

This is absolutely right and the calendar "problem" should still be dealt with, of course. We acknowledge that it would be very difficult for most modelling groups to adapt their online averaging procedures to account for the different definitions of the months for *midHolocene* and *lig127k*. This is why we require daily data for a few key variables so that the differences due to the changing calendars can be computed offline. This is now explained in section 4.5 and illustrated (Figure 5).

*While I am generally very much in favor of the PMIP community to publish their experimental strategy, the current ms. needs considerable re-writing to become a useful contribution. I would suggest to drastically reduce the length and to focus it on new aspects:*

*- Sections 1 and 2 contain almost no new information and should be largely replaced by a concise summary of dynamical insights gained from earlier PMIP phases (see above)*

We have considerably shortened these sections and added a few examples of "dynamical insights".

*- Figs. 2-5 were published elsewhere and should be removed*

We have kept Fig. 2 which was created for this manuscript but removed all others as required.

*- Focus should be on what is now sections 3 and 4.*

These sections have been extended with a complete protocol and more detailed analyses plan.

*Finally, someone out the large group of authors should read the ms. from beginning to end to ensure that the wording/style is consistent (incl. the ref. list; "et al" vs. et al."; paleo vs. palaeo) and that geological ages are correct and consistent among the ms. (mPWP 3.2 Ma vs. 3.3-3 Ma; incorrect start of the Quaternary at 2.5 Ma).*

Thank you for spotting these. We have done our best to have fully consistent text in this new version. The confusion about the dates for the mid Pliocene Warm Period might arise from the fact that while PRISM, the project which specifically deals with climate reconstructions for this period, has considered the full mid Pliocene Warm Period (from ~3.3 to 3 million years ago), the second phase of PlioMIP specifically focuses on a warm interglacial within the mid Pliocene Warm Period, and this interval, termed KM5c, has been dated to 3.205 Ma ago. We have added a sentence to clarify this, in section 1.3: "The mPWP experiment focuses on a specific interglacial period, dated at ~3.2 Ma before present, during the mid-Pliocene interval (3.3 to 3 Ma before present)."

**Response to Steve Sherwood**

*This paper nicely lays out the plan for PMIP4, which is part of CMIP6. It describes the history and rationale for this project, then reviews the experiments which cover five periods in recent Earth history and expand on those of previous PMIPs. The protocols for these experiments are described, a few potential pitfalls are noted for modellers to beware of, and useful links to other MIPs are noted. I think the project sounds exciting and hope that plenty of modelling centres are able to participate. This article would benefit from some very minor revisions to improve clarity, but otherwise I am happy with it and don't see any major omissions.*

We thank the reviewer for his enthusiasm. Despite this support, we have largely revised the manuscript according to the other reviewers' comments and the editor's comments.

*The minor revisions to suggest are:*

*6, 18-29: Please mention (as I gather later though am still not 100% sure) that all of these experiments are "time-slice" experiments, i.e., the model is run to statistical equilibrium with time-invariant forcing specified based on the quoted point in time to give a representative mean state, except for the last millennium which is transient and includes time-varying forcings such as volcanic eruptions.*

This was already mentioned in section 3.1 but we have added the information in section 1.3 too, following this comment.

*8, 19: The word "observed" should be reserved for the instrumental period with real observations. Can we instead say "proxy-estimated" (or similar)?*

We distinguish between primary observations, such as pollen counts, geomorphic features, or isotopic measurements, and the climate inferences or reconstructions made using these observations. We have therefore now distinguished between palaeoenvironmental observations and climate and/or vegetation reconstructions consistently throughout the text.

*8, 29: By "mean values" do you mean global means? Time averages? Changes in global mean relative to modern?*

This sentence has disappeared from the text following our decision to shorten Section 2.

*10,40 By "trends" do you mean differences (relative to modern)?*

We actually meant both differences relative to modern and trends, for transient simulations of the last interglacial. This has now been clarified.

*11, 36: please fix error message*

The error message has been fixed.

*11,34: suggest "current" rather than "modern" (which can mean many things… from a palaeo perspective preindustrial could be viewed as modern)*

ok, done.

*17, 36-37: I don't think the polarity of the forcing is the real problem, but rather, the fact that some radiative forcing agents produce larger responses per unit global-average power input than do others, and/or provoke "rapid adjustments" to the forcing that are unrelated to global-mean warming or cooling. This is due to the spatial pattern of the forcing. Given the fact that a major goal here is to test model responses to forcings, and given that past forcings are different from the dominant ones we worry about for the future (greenhouse gases), this topic may deserve a bit more discussion.*

It is true that the spatial patterns of the different forcings (and their responses) can be very different and they will be probably even more different when the dust forcing is included, because it introduces a radiative forcing which is quite regional. We have rephrased the sentence which could be misinterpreted due to the allusion to different forcings being responsible for cooling and warming. We have also introduced the single forcing experiments which are coordinated in PMIP4 and will help disentangling the response to individual forcings and their contribution to the total response.

*Table 2: some of the cells in the table are blank, and I am not sure what this means. Suggest every box should say something (even if it is "see text")*

All boxes of table 2 are now filled.

*Figure 1. I found this figure confusing; maybe you are trying to cram too much into one panel. The caption refers to panel labels (a, b, …) but there aren't any. It is hard to figure out what each curve as and which axis refers to what (especially when there is one on each side, or where it switches from right side to left going from one column to another). I would prefer the axes to be individually labelled with the quantity and units, or give the quantity in the title or above the curve and the units on the axis so we know we are looking at the right one. It is not at all obvious that they grey banded stripe is meant to represent the possible range of CO2 in the left column; nothing wrong with showing it this way but please make clearer what everything is!*

We apologize for this confusion. The labels have been added. Colored axes are used so that it is clear which axis a given line corresponds to.

*Figure 2a. I presume there are black dots hiding underneath the red ones for mid- Holocene and Historical? To help avoid confusion perhaps the caption could mention this.*

Yes, this is right and indicated in the caption.

*Figure 2b. Please give units! Also, please explain in the caption what "OAV" and "OAC" refer to.*

The unit (mm/year) has been added to the caption.

*Figure 3. Please be consistent in labelling the panels (they are given (a,b) in the figure but you say lhs and rhs in the caption). Is this for land only, or land+ocean? What is STSI and ssTSI? Does 31 points mean 31 years? (give the time width of the smoothing window rather than the number of points)*

Figure 3 has been removed following the other comments on this manuscript.

*Figure 4. It doesn't appear to me that the proxy data are able to tell us anything about the seasonal cycle, given that the differences are small compared to the scatter --- so is it worth including the two*

*right panels in a review paper on PMIP? Especially since the figure is reproduced from another source so anyone who really wants to see the seasonal results can find them. I see later you are already requesting permission to use a portion of another IPCC figure, I'd suggest making a similar request here.*

Figure 4 has been removed following the other comments on this manuscript.

*Figure 7. Please spell out "preindustrial" rather than PI since PI is not one of the study time periods and you haven't used this acronym much.*

This information has been placed in the caption as it was not fitting in the title of the plot.

Response to Anonymous Referee #3

*This manuscript details the contribution of the paleoclimate modeling community to the new Coupled Modelling Intercomparison Project. For this phase the PMIP community is planning to expand its contribution significantly, including 5 different periods or experiments. In order that as many modelling groups as possible participate in this effort a clear modelling setup. This manuscript gives an overview of the rationale and broadly outlines the experimental setup. Hence the importance of this paper is clear*

*General Comments*

*I find the manuscript a bit long without going into the specific details of each experiment. This is ok as it is mentioned that there will be special papers for each one of them. I would suggest shortening section 1, and section 2 could benefit for a clearer discussion on current modelling gaps.*

We have indeed considerably shortened Sections 1 and 2 and removed figures 3 to 5. Section 2 was originally written to summarize previous work on each period, including aspects which models could not account for. We have chosen to keep this section short and have not added information but have also attempted to clarify what is new in the PMIP4-CMIP6 experiments compared to the previous phases, e.g. including the dust forcing, which is new for the *midHolocene* and *lgm* experiments, including the analysis of the sensitivity of the *lgm* results to the imposed ice sheets, using an improved volcanic forcing for the *past1000* experiment. In our opinion, this addresses "modelling gaps" in a positive manner.

*I have also personal comment. Having worked with the past1000 simulations, it would be very useful if all the modelling groups planning to run this experiment, would run it up to present. (850-2005 CE instead of stopping in 1850).*

This is an important point which has been added to section 2.2 and explained in the specific manuscript on the past1000 experiment (Jungclaus et al, 2016).

*Figures: I a not familiar with the journal's policy in this respect, but it seems odd to me to use so many previously published figures. Also the quality and style of the figures is too diverse. For example, please use just one projections for all maps.*

We have removed Figures 3 to 5. Figure 6 (now Figure 3) has been redrawn with projections similar to the other maps displayed in the paper.

*There is a lot of inconsistent naming of the experiments throughout the text. Please use either "midHolocene" or "MH". "LM" or "past1000" and so on.*

We agree this could be confusing. We use different names for the periods and the experiments. This is now clearly explained in Section 1.3. We have been careful in updating the manuscript accordingly.

*Specific comments*

*Page3, Line 13: Missing comma after However. Or are you using "however = to whatever extent" ?*

A comma has been added.

*Page3, Lines 14-15: used twice "well outside"*

This repetition was made on purpose so that the two facts could be linked.

*Page 5, lin34: include years for "the last interglacial".*

done.

*Page 6, line 1: : : :of participating MODELLING groups.*

done

*Page 6, line5: in this section maybe include the CMIP6 questions?*

We did include the CMIP6 questions in the first paragraph, but do not feel it is necessary to lengthen the title of the section since we also discuss the link with the CMIP6 key questions in Section 4 (plan of analyses).

*Page6, line 7: is "How does the Earth System respond to forcing" question 1? How many questions are there? Please introduce a bit better.*

This is the exact formulation of the CMIP6 first key question, this is why we give it between quotes. The reference to Eyring et al (2016) has been added to indicate that this formulation does not depend on us. Examples of analyses of PMIP4 results to answer these questions are further given in the analysis plan.

*Page 6, line35: define DECK*

done, with a reference to Eyring et al, 2016.

*Page 7, line 20: change "interesting" for "valuable"*

done

*Page 7, lines 41-42: please be consistent between title and text about MH and LGM.*

As explained above, we use "MH" and "LGM" for the periods, and *midHolocene* and *lgm* for the experiments themselves. We hope that it is clearer now.

*Page 10, line 24: question: is there a separate paper with specifics?*

Yes, this is indicated in Section 3.

*Page10, line 34: needs a reference.*

A reference to Capron et al (2014) has been added

*Page 11, line 36: there is a typo.*

The faulty reference has been removed. The following sentence has been reformulated.

*Page 13, line 27: delete "web sites". Its repeated.*

The sentence has been fixed (and has no occurrence of "web sites" now).

*Page 17: line 22: delete "the"*

done

*Figure 1: needs more information of the different panels. Very hard to understand at this point*

We have modified the figure – cf. response to the second reviewer.

Response to the CMIP Panel

_Dear PMIP authors,_

_The CMIP Panel is undertaking a review of the CMIP6 GMD special issue papers to ensure a level of consistency in answering the key questions that were outlined in our request to submit a paper to all co-chairs of CMIP6-Endorsed MIPs. These questions are outline in the overview paper (Eyring et al, GMD, 2016) and the relevant section is summarised below:_

_'Each of the 21 CMIP6-Endorsed MIPs is described in a separate invited contribution to this Special Issue. These contributions will detail the goal of the MIP and the major sci entific gaps the MIP is addressing, and will specify what is new compared to CMIP5 and previous CMIP phases. The contributions will include a description of the experimental design and scientific justification of each of the experiments for Tier 1 (and possibly beyond), and will link the experiments and analysis to the DECK and CMIP6 historical simulations. They will additionally include an analysis plan to fully justify the resources used to produce the various requested variables, and if the analysis plan is to compare model results to observations, the contribution will highlight possible model diagnostics and performance metrics specifying whether the comparison entails any particular requirement for the simulations or outputs (e.g. the use of observational simulators). In addition, possible observations and reanalysis products for model evaluation are discussed and the MIPs are encouraged to help facilitate their use by contributing them to the obs4MIPs/ana4MIPs archives at the ESGF (see Section 3.3). In some MIPs additional forcings beyond those used in the DECK and CMIP6 historical simulations are required, and these are described in the respective contribution as well.'_

_We very much welcome the PMIP contribution and the detailing of the experimental design, analysis plan and diagnostic output that you currently cover in sections 3 and 4. We also welcome the strong links that PMIP has clearly forged with other CMIP6 MIPs and look forward to the joint analysis that you describe._

We thank the CMIP Panel for these positive comments

_Additionally, we would like to see some more detail on some of the issues raised above, notably;_

_a. More discussion on the specific goals of PMIP4 in CMIP6 and what science gaps it is attempting to fill. You describe the 3 CMIP6 science questions and PMIP links to them in Section 1.3 and the links to the WCRP GCs in section 4.3, but it would be good to see some discussion on what PMIP4 is hoping to achieve that is new since PMIP3._

We have largely re-written section 4 with this goal in mind.

_b. The description of the experimental design for each experiment is comprehensive and very useful. There are however, a worrying large number of papers 'to be submitted'._

_Is it clear that once this paper is published modelling groups will be able to rely on it to provide a comprehensive start point for setting up their experiments?_

This is right, and we have clarified the situation in the revised manuscript. All necessary information to set up the experiments is now given in this manuscript. The accompanying manuscripts (now submitted except for the LGM paper which has now been sent to co-authors) are giving additional

justifications and technical explanations about this set-up and the design of complementary (sensitivity) experiments.

*c. A lot of focus in the analysis plan is given to the multi-time period analysis, but not all modelling centres will be contributing to all (or indeed in some cases more than 1) of the entry card/tier 1 experiments. Could more be said about analysis of the specific experiments and what new we will learn from these experiments since PMIP3-CMIP5 (higher resolution, better data, more ES components)?*

We rely on Table 1 to give a summary of the analyses which will be carried specific experiments and explain it in Section 4. We have chosen not to lengthen the manuscript with extended analyses plans for each periods because these are given in the companion papers, in which they are better articulated with the associated sensitivity experiments. On the other hand, we have given more detail on the reconstructions available to evaluate the models for the five PMIP4-CMIP6 periods, hence giving more details on the possible benchmarking analyses, in line with the fact that CMIP6 required data to be associated with each MIP.

*d. You make the point that the comparison of these time periods to palaeodata is one of the key drivers but say very little about the observational data sources or whether these products will be made available to the community to facilitate comparison. In section 4.4 you describe the new metrics and forward modelling you request the models output. It would be good to document how these will be evaluated.*

We have dedicated a new section to the available paleodata, pointing to available data sets (Section 4.1). This is by no means exhaustive but demonstrates the existence of reconstructions for each PMIP4 period. This is a very active area of research and there will be new syntheses, which we have indicated too.

*We hope you agree that some level of consistency across the MIP papers in this special issue is valuable and that the above suggestions can be accommodated in your paper.*

*Other comments:*

*e. There is a lack of consistency in the naming of the experiments e.g. notably the use of LM and past1000 somewhat interchangeably. Please clarify*

We use different names for the periods (LM: last millennium, MH: mid Holocene, etc) and the associated experiments (past1000, midHolocene, etc). This is now clarified as early as possible in the manuscript, in section 1.3.

*f. In section 3.2, the implementation of the ice-sheets needs to be a bit clearer. For example do all points 2-5 refer to both midplioceneEoi400 and lgm?*

This has been clarified:

Steps 2 and 3 are compulsory for both experiments,

*With many thanks for your ongoing efforts in the CMIP6 process.*

*The CMIP Panel*

Response to the Editor's comments

*[21 July 2016]*

*I think most things have been covered by the reviewers and the CMIP panel's comment, so please enjoy responding to their comments. In common with one of the reviewers, I'm not enchanted by the history lesson. I understand that you want to draw people into considering paleoclimate, but I think the resulting length of the paper is more likely to turn people away.*

We have shortened the history lesson and re-focussed the manuscript on the protocol and analysis plan.

*However, the real problem for publication of this paper in GMD is the possibly incomplete protocols.*

*From Section 3: "The modified forcings and boundary conditions for each PMIP4-CMIP6 palaeoclimate simulation are summarised in Table 2. The complete details of the experimental protocols are given in a series of companion papers: Otto-Bliesner et al for the midHolocene and lig127ka experiments, Kageyama et al for the lgm, Jungclaus et al for the past1000 and Haywood et al (2016) for the midPliocene-eoi400 experiment. These papers also explain how the boundary conditions for each period have been built and constitute key references for the experimental protocol for each of the PMIP4-CMIP6 simulations."*

*The problem, as highlighted by the CMIP panel, is that most of these papers are not published. I can't control what is going to be in those other papers - all I can do is make sure that th is paper meets the peer review requirements. Therefore, for all experiments please include in this paper, "the complete details of the experimental protocols" for the PMIP4-CMIP6 experiments. You can leave the details on "how the boundary conditions for each period have been built" to the still to be submitted papers. You can also leave all alternate experiments that are within PMIP4 but outside CMIP6 to those other papers. Please make sure that Table 2 is edited so that it does not appear that essential details required for setting up the experiments are included in these unpublished papers. Basically, a modeller should be able to set up the PMIP4-CMIP6 runs using the information contained in this paper.*

*The alternative is that we put this paper hold until the other papers have passed through peer review. This could be workable as I understand that you intend to submit the other papers also to GMD. Even in that case, I would like to see the complete protocols for PMIP4-CMIP6 detailed here (citing the other papers as required), as trying to extract the CMIP6 protocols from the much more elaborate single-interval papers is likely to be a trying process.*

The accompanying papers have been or will soon be submitted. We agree that the present manuscript is held until the companion papers are accepted, and to update the protocol accordingly, if necessary, to guarantee consistency between the manuscripts.

*[6 October 2016]*

*I'm also editing the DeepMIP paper, and they happen to have a rather prominently positioned section about data for comparison with the model output, which caused me to pay attention to it, and then criticise it as inadequate! I think that more or less the same criticism can be levelled at this PMIP4 manuscript. Section 4.2 touches on ways in which the models may be compared to data, but this is*

*less important than outlining the datasets that are available and including specifics about any new datasets that are planned to emerge within the timeframe of the project. Please add this information for all the PMIP4 periods in the revised manuscript. It doesn't have to be a huge amount of material - the point is a practical one - to lead users of PMIP model output in the right direction so they can discover the relevant datasets. I expect you would include citations in the revised section 4.2, but you might also wish to include some additional information (DOIs/weblinks etc) in the Data Availability section.*

We have added section 4.1 and one table to give an overview on the available data sets for each period.

*[ 27 October 2016 ]*

*In response to my previous comment which highlighted the problem of citing un-submitted works, the authors elected to put this manuscript on hold until those manuscripts were submitted to GMD. Those manuscripts are now almost complete, and the authors have enabled me to see two out of the three of them for approval before submitting to GMD.*

*I find there is a real problem, which is that these two draft manuscripts both contain substantial discussion of the background and rationale behind the \*CMIP6\* PMIP4 protocols. These discussions are too entwined with the non-CMIP6 experiments within PMIP4 to be removed from those draft papers, and including the content within this overview manuscript under consideration would be entirely impractical.*

*In terms of the peer review at GMD I find it unacceptable that we should in this overview manuscript be approving the protocols for the CMIP6 runs without first reading the rationale behind them!*

*All this suggests that the complete protocols should be peer reviewed within those other papers. Since those papers describe the rationale behind the CMIP6 protocol they must also be included in the CMIP6 special issue.*

*As for the present paper under consideration, a purely descriptive paper about PMIP does not fit into the peer review criteria and I have great difficulty accepting it as a standalone paper. However, if it were presented as the introductory part of a multi-part paper, I think it will be a useful contribution to the whole. This has led me to come up with a solution that solves all these problems, and I think it also produces a nice final product. I suggest that the papers be submitted as 4 parts of a multi-part paper. The titles should be made consistent with each other and include the Part number in the title. They do not have to be exactly this, but something like this would work...*

*PMIP4-CMIP6, the contribution of the Paleoclimate Modelling Intercomparison Project to CMIP6, Part 1: Introduction and(/or?) Overview*

*PMIP4-CMIP6, the contribution of the Paleoclimate Modelling Intercomparison Project to CMIP6, Part 2: Eemian and midHolocene*

*PMIP4-CMIP6, the contribution of the Paleoclimate Modelling Intercomparison Project to CMIP6, Part 3: The Last Millennium*

*PMIP4-CMIP6, the contribution of the Paleoclimate Modelling Intercomparison Project to CMIP6, Part 3: The Last Glacial Maximum*

*From the point of view of this paper, the authors are welcome to submit a revised version, but if outline protocols for the experiments are included then the final publication will need to wait until all the other three papers are accepted.*

*I appreciate the difficulties the authors have had with trying to write a single coherent paper as a contribution to the GMD CMIP6 special issue, but I think it was an intractable problem, because the resulting paper could not pass the GMD peer review criteria of including both the rationale and the protocols. In that context, I'd like to contrast this MIP with one of the others I edited. CFMIP have a huge number of experiments fully described within their paper, including tier 1 and tier 2 experiments. I think it is an elegant paper. However, CFMIP experiments are all highly idealised, and thus there is no great debate to be had in defining the protocols and they can be very simply described. PMIP, on the other hand is trying to model real and disparate intervals in the earth's climate history. This is an order of magnitude more complex, involving several different communities of scientists, and it so it is not appropriate to squeeze it all into a single paper. The multi-part paper will enable these communities to each take the responsibility for the defence of their own experiment protocols.*

As stated before, we have accepted to publish this manuscript only after the other three, on each PMIP experiments, are accepted. Nonetheless, we have prepared this revision and entitled it "Part 1" to make explicit the changes we have already implemented following the comments from the reviewers and the editor and the fact that this manuscript is linked to three other manuscripts + the manuscript by Haywood et al, 2016, already published in CP. For the moment, we have kept the complete protocols in this manuscript as recommended by the CMIP Panel.

---

## Author Response (AR2)

Response to the Editor (initial comment in black, our response in blue)

There has been some deliberation over this paper among the three editors handling the PMIP experiment description paper (one of whom has a self-declared conflict of interest, which was taken into account). Fundamentally none of us consider the paper satisfactory at present, but James Annan and I (who do not have a conflict of interest) have come up with a potential solution, which should allow this PMIP4 "bookmark" to appear in the tome of the CMIP6 special issue.

We think that the paper should be reformulated as a review or summary paper. One framing could be to think in terms of being of interest and use to people currently of outside of the PMIP mainstream, who would benefit from an introduction to PMIP and a rough guide to which experiments they might be able to do. They probably wouldn't be so willing to wade through 4 different detailed experimental protocol papers to see if they would work out which are relevant to them and what the important differences are. Such people exist! We have met them! I imagine that the resulting summary would also be useful to new PMIP researchers as well as those who so far have only been involved in one of the PMIP sub-MIPs, but are interested in getting a wider appreciation of the whole project.

In order to fulfil these requirements the paper needs a major overhaul. The manuscript is too long and does not need to cover so much detail, I don't think it needs so much background, and many of the figures seem unnecessary, and they are anyway rather incomplete (eg 3 and 4 are two sets of partial forcings for a couple of experiments, 2 and 5 are bits of analysis that people have done for past PMIPs, but are by no means comprehensive). The tables are better, although they will need updating with the information from the now accepted/published sub-MIP papers. The text also needs to be made consistent with the other three papers, which we went to some lengths to coordinate, in terms of both the language and the actual requirements of the different MIPs.

We thank the Editor(s) for finding the solution for this manuscript to be treated as a "review" of the four PMIP papers describing the experimental protocols of the 5 PMIP4-CMIP6 simulations. We now refer to the companion papers, which are all accepted, on the protocols whenever possible.

We have shortened the manuscript, in particular sections 1 and 2, but also section 3 (in particular the ice sheet part which was indeed very technical). We have removed the figures on previous PMIP experiments but kept summary figures for the ice sheets and dust, the former because it helps comparing the LGM and Pliocene forcings, the second because we want to emphasize this new forcing in the PMIP experiments. We have also kept Figure 4 on the "calendar effect" because this has important consequences on the required output and is an important message for outsiders from the PMIP community.

We hope the manuscript now reads better for people from outside PMIP and is fulfilling all requirements to be accepted in GMD.

Yours sincerely,

Masa Kageyama (on behalf of the authors)

The PMIP4 contribution to CMIP6 - Part 1: Overview and over-arching analyses plan\*

Masa Kageyama1, Pascale Braconnot1, Sandy P. Harrison2, Alan M. Haywood3, Johann H. Jungclaus4, Bette L. Otto-Bliesner5, Jean-Yves Peterschmitt1, Ayako Abe-Ouchi6,7, Samuel Albani8, Patrick J. Bartlein9, Chris Brierley10, Michel Crucifix11, Aisling Dolan3, Laura Fernandez-Donado12, Hubertus

5 Bartlein9, Chris Brierley10, Michel Crucifix11, Aisling Dolan3, Laura Fernandez-Donado12, Hubertus Fischer13, Peter O. Hopcroft14, Ruza F. Ivanovic3, Fabrice Lambert15, Daniel J. Lunt14, Natalie M. Mahowald16, W. Richard Peltier17, Steven J. Phipps18, Didier M. Roche1,19, Gavin A. Schmidt20, Lev Tarasov21, Paul J. Valdes14, Qiong Zhang22, Tianjun Zhou23

1Laboratoire des Sciences du Climat et de l'Environnement, LSCE/IPSL, CEA-CNRS-UVSQ, Université Paris-Saclay, F-91191 Gif-sur-Yvette, France 2Centre for Past Climate Change and School of Archaeology, Geography and Environmental Science (SAGES) University of Reading, Whiteknights, RG6 6AH, Reading, United Kingdom 3School of Earth and Environment, University of Leeds, Woodhouse Lane, Leeds, LS2 9JT, United Kingdom

4Max Planck Institute for Meteorology, Bundesstrasse 53, 20146 Hamburg, Germany 5National Center for Atmospheric Research, 1850 Table Mesa Drive, Boulder, Colorado 80305, United States of America 6America Descent Institute University of Tabua 5, 1, 5, Kashiwanaka, Kashiwa aki, Chiba 277

6Atmosphere Ocean Research Institute, University of Tokyo, 5-1-5, Kashiwanoha, Kashiwa-shi, Chiba 277-8564, Japan

- 20 7Japan Agency for Marine-Earth Science and Technology, 3173-25 Showamachi, Kanazawa, Yokohama, Kanagawa, 236-0001, Japan
  - 8Institute for Geophysics and Meteorology, University of Cologne, Cologne, Germany 9Department of Geography, University of Oregon, Eugene, OR 97403-1251, United States of America
- 10University College London, Department of Geography, WC1E 6BT, United Kingdom

13Climate and Environmental Physics, Physics Institute & Oeschger Centre for Climate Change Research, University of Bern, Sidlerstrasse 5, CH-3012 Bern, Switzerland

14School of Geographical Sciences, University of Bristol, Bristol, United Kingdom
 15Catholic University of Chile, Department of Physical Geography, Santiago, Chile
 16Department of Earth and Atmospheric Sciences, Bradfield 1112, Cornell University, Ithaca, NY 14850, United States of America
 17Department of Physics, University of Toronto, 60 St. George Street, Toronto, Ontario M5S 1A7, Canada

35 18Institute for Marine and Antarctic Studies, University of Tasmania, Private Bag 129, Hobart, TAS 7001, Australia

19Earth and Climate Cluster, Faculty of Earth and Life Sciences, Vrije Universiteit Amsterdam, Amsterdam, the Netherlands

20NASA Goddard Institute for Space Studies and Center for Climate Systems Research, Columbia University
 2880 Broadway, New York, NY 10025, United States of America

21Department of Physics and Physical Oceanography, Memorial University of Newfoundland and Labrador, St. John's, NL, A1B 3X7, Canada

- 22Department of Physical Geography, Stockholm University, Stockholm, Sweden
- 23LASG, Institute of Atmospheric Physics, Chinese Academy of Sciences, P.O. Box 9804, Beijing 100029,

45 China

50

**Correspondence to: Masa Kageyama (Masa.Kageyama@lsce.ipsl.fr)**

\* This paper is the first of a series of 4 GMD papers on the PMIP4-CMIP6 experiments. Part 2 (Otto-Bliesner et al. 20162017) gives the details about the two PMIP4-CMIP6 interglacial experiments, Part 3 (Jungclaus et al., 20162017) about the last millennium experiment, and Part 4 (Kageyama et al., 2017) about the Last Glacial Maximum experiment. The mid Pliocene Warm Period experiment is part of the Pliocene Model Intercomparison Project (PlioMIP) - Phase 2, detailed in Haywood et al. (2016).

**Abstract.**

The goal of the Paleoclimate Modelling Intercomparison Project (PMIP) is to understand the response of the climate system to different climate forcings and feedbacks. Through comparison with observations of the 5 environmental impact of these climate changes, or with climate reconstructions based on physical, chemical or biological records, PMIP also addresses the issue of how well state-of-the-art numerical models simulate climate change. Paleoclimate states can be radically different from those of the recent past documented by the instrumental record, and thus provide an out-of-sample test of the models used for future climate projections and a way to assess whether they have the correct sensitivity to forcings and feedbacks. Five different periods have

- 10 been designed to contribute to the objectives of the sixth phase of the Coupled Model Intercomparison Project (CMIP6): the millennium prior to the industrial epoch (CMIP6 name: past1000), the mid-Holocene, 6,000 years ago (*midHolocene*); the Last Glacial Maximum, 21,000 years ago (*lgm*); the Last Interglacial, 127,000 years ago (lig127k) and mPWP, the mid-Pliocene Warm Period, 3.2 million years ago (midPliocene-eoi400). These climatic periods are well documented by paleoclimatic and paleoenvironmental records, with climate and
- 15 environmental changes relevant for the study and projections of future climate changes. This manuscript describes the motivation for the choice of these periods and the design of the numerical experiments\_and database requests, with a focus on their novel features compared to the experiments performed in previous phases of PMIP and CMIP. Analyses of the individual periods, across all the periods and comparisons with other CMIP6 simulations, will allow examination of relationships between forcings of different nature and amplitude 20 and climate responses, and comparison of the processes involved in these responses. The evolution of interannual variability in the past is also expected to provide some clues on the linkages between mean climate and climate variability. This manuscript also describes the information needed to document each experiment, the experimental protocols, and the model outputs required for analysis and benchmarkingIt also outlines the analysis plan that takes advantage of the individual periods, the comparisons of the results across periods and 25 across CMIP6 in collaboration with other MIPs,

[revised manuscript text omitted]

|----------------------------------------------------------------------------------------------------------------------------------------------------------------------------------|
|                                                                                                                                                                                  |

[revised manuscript text omitted]

5

15

Webb, M. J., Andrews, T., Bodas-Salcedo, A., Bony, S., Bretherton, C. S., Chadwick, R., Chepfer, H., Douville, H., Good, P., Kay, J. E., Klein, S. A., Marchand, R., Medeiros, B., Siebesma, A. P., Skinner, C. B., Stevens, B., Tselioudis, G., Tsushima, Y., and Watanabe, M.: The Cloud Feedback Model Intercomparison Project (CFMIP) contribution to CMIP6, Geosci. Model Dev. Discuss., 10, 359-384, https://doi+.org/10.5194/gmd-2016 70, in review, 2016/10-359-2017, 2017.

- Williamson, D., Goldstein, M., Allison, L., Blaker, A., Challenor, P., Jackson, L., and Yamazaki, K: History matching for exploring and reducing climate model parameter space using observations and a large perturbed physics ensemble, Climate Dynamics, 41, 1703–1729, doi:10.1007/s00382-013-1896-4, 2013.
- Winsor, K., Carlson, A.E., Klinkhammer, G.P., Stoner, J.S. and Hatfield, R.G.: Evolution of the northeast Labrador Sea during the last interglaciation. Geochemistry Geophysics Geosystems, 13, 2012.
  - Wittenberg, A. T.: Are historical records sufficient to constrain ENSO simulations? Geophys. Res. Lett., 36, L12702, doi:10.1029/2009GL038710, 2009.
- Yeo, K. L., N. A. Krivova, S. K. Solanki, and Glassmeier, K.H.: Reconstruction of total and spectral solar irradiance from 1974 to 2013 based on KPVT, SoHOMDI, and SDOHMI observations. Astron. Astrophys., 570, A85, DOI: 10.10510/004-6361/201423628, 2014.
- Yoshimori, M., Hargreaves, J. C., Annan, J. D., Yokohata, T. and Abe-Ouchi, A.: Dependency of feedbacks on forcing and climate state in physics parameter ensembles, Journal of Climate, 24, 6440-6455, 2011.
- Zanchettin, D., Khodri, M., Timmreck, C., Toohey, M., Schmidt, A., Gerber, E. P., Hegerl, G., Robock, A., Pausata, F. S. R., Ball, W. T., Bauer, S. E., Bekki, S., Dhomse, S. S., LeGrande, A. N., Mann, G. W., Marshall, L., Mills, M., Marchand, M., Niemeier, U., Poulain, V., Rozanov, E., Rubino, A., Stenke, A., Tsigaridis, K., and Tummon, F.: The Model Intercomparison Project on the climatic response to Volcanic forcing (VolMIP): experimental design and forcing input data for CMIP6, Geosci. Model Dev., 9, 2701-2719, doi:10.5194/gmd-9-2701-2016, 2016.
- Zhang, R., Yan, Q., Zhang, Z. S., Jiang, D., Otto-Bliesner, B. L., Haywood, A. M., Hill, D. J., Dolan, A. M.,
  Stepanek, C., Lohmann, G., Contoux, C., Bragg, F., Chan, W.-L., Chandler, M. A., Jost, A., Kamae, Y., Abe-Ouchi, A., Ramstein, G., Rosenbloom, N. A., Sohl, L., and Ueda, H.: Mid-Pliocene East Asian monsoon climate simulated in the PlioMIP, Clim. Past, 9, 2085-2099, doi:10.5194/cp-9-2085-2013, 2013.

30 Zheng, W., P. Braconnot, E. Guilyardi, U. Merkel, and Y. Yu: ENSO at 6ka and 21ka from ocean-atmosphere coupled model simulations, Climate Dynamics, 30, 745-762, doi:DOI 10.1007/s00382-007-0320-3, 2008 Mis en forme : pb\_toc\_link

5

10

Zheng, W., Braconnot, P.: Characterization of model spread in PMIP2 Mid-Holocene simulations of the African Monsoon, Journal of Climate, 26, 1192-1210, 2013.

|    | DI | TO  |
|----|----|-----|
| IA | BL | 'ES |

| Period                                                           | CMIP6 Priority                                                                                                                                                                                                                                                                                                                                                                |                                      |
|------------------------------------------------------------------|-------------------------------------------------------------------------------------------------------------------------------------------------------------------------------------------------------------------------------------------------------------------------------------------------------------------------------------------------------------------------------|--------------------------------------|
| Last millennium
( past1000 )
850-1849 CE     |  <li>a) Evaluate the ability of models to capture reconstructed variability on multi-decadal and longer time-scales.</li> <li>b) Determine what fraction of the variability is attributable to "external" forcing and what fraction reflects purely internal variability.</li> <li>c) Provide a longer-term perspective for detection and attribution studies.</li>  | Tier 1*                              |
| Mid-Holocene
( midHolocene )
6 kyr ago       | <li>a) Compare the model response to known orbital forcing changes
and changes in greenhouse gas concentrations to paleodata,
describing major temperature and hydrological changes.</li><li>b) Relationships between changes in mean state and variability</li>                                                                                             | Tier 1*
PMIP4-CMIP6
entry card |
| Last
MaximumGlacial(lgm)21 kyr ago                            | <li>a) Compare the model response to ice-age boundary conditions with paleodata.</li><li>b) Attempt to provide empirical constraints on global climate sensitivity.</li>                                                                                                                                                                                             | Tier 1*
PMIP4-CMIP6
entry card |
| Last Interglacial
( lig127k )
127 kyr ago           | <li>a) Evaluate climate model for warm period in northern hemisphere and high sea-level stand.</li><li>b) Impacts of this climate on sea ice and ice sheets.</li>                                                                                                                                                                                                    | Tier 1*                              |
| Mid-Pliocene
Warm Period
(midPlioceneEoi400)
3.2 Ma ago |  <li>a) Earth System response to a long term to CO2 forcing analogous to that of the modern.</li> <li>b) Significance of CO2-induced polar amplification for the stability of the ice sheets, sea-ice and sea-level.</li>                                                                                                                      | Tier 1*                              |

 Table 1: Characteristics, purpose and CMIP6 priority of the five PMIP4-CMIP6 experiments. \* All experiments can be run independently. It is not mandatory to perform all Tier 1 experiments to take part in PMIP4-CMIP6, but it is mandatory to run at least one of the PMIP4-CMIP6 entry cards.

| Period                  | GHG                                                    | Astronomical  | Ice-sheets | Tropospheric     | Land surface**                     | Volcanoes     | Solar    | Reference to be            |
|-------------------------|--------------------------------------------------------|---------------|------------|------------------|------------------------------------|---------------|----------|----------------------------|
|                         |                                                        | parameters    |            | aerosols *       |                                    |               | activity | cited                      |
| PMIP4-CMIP6 entry cards |                                                        |               |            |                  |                                    |               |          |                            |
| Mid-Holocene            | CO 2 : 264.4 ppm                            | 6 kyr BP      | as in PI   | modified         | Interactive vegetation             | as in PI      | as in PI | Otto-Bliesner et al,       |
| (midHolocene)           | CH 4 : 597 ppb
N 2 O: 262 ppb |               |            | (if possible)    | OR Interactive carbon cycle        |               |          | <del>20162017</del> |
| 6 ky ago                | CFC:0                                                  |               |            |                  | OR fixed to present day            |               |          |                            |
|                         | O 3 : pre-industrial                        |               |            |                  | (depending on model
complexity) |               |          |                            |
| Last Glacial            | CO 2 : 190 ppm                              | 21 kyr BP     | modified   | modified         | Interactive vegetation             | as in PI      | as in PI | Kageyama et al, in         |
| Maximum                 | CH 4 : 375 ppm                              |               | (larger)   | (if possible)    | OR Interactive carbon              |               |          | prep,                      |
| (lgm)                   | N 2 O: 200 ppb                              |               |            |                  | cycle                              |               |          | <del>2016</del> 2017       |
| 011                     |                                                        |               |            |                  | (depending on model                |               |          |                            |
| 21 ky ago               | $O_3$ : pre-industrial                                 |               |            |                  | (depending on model                |               |          |                            |
| Tior 1 PMIP4_CMIP       | S experiments                                          |               | I          |                  | complexity)                        |               | I        |                            |
| Last millonnium         | Time verving                                           | time verying  | as in DI   | os in D I | time verying                       | timo vorving  | timo     | Jungelous at al. to        |
| Last millennum          | (Moinshouson of                                        | (Borgor 1078  | as III F I |                  | (land use)                         | radiativa     | vorving  | $2016\ 2017$               |
| (past1000)              | (Wiemsnausen et                                        | Schmidt at al |            |                  | (land use)                         | forcing due   | varynig  | 2010 2017                  |
| 850-1849 CE             | $\frac{d1.}{d100}$                                     | 2011          |            |                  |                                    | to            |          |                            |
| 050-1047 CL             | 13500 2017 )                                    | 2011)         |            |                  |                                    | stratospheric |          |                            |
|                         |                                                        |               |            |                  |                                    | aerosols      |          |                            |
| Last Interglacial       | CO 2 : 275 ppm                              | 127 ky BP     | as in PI   | modified         | Interactive vegetation             | as in PI      | as in PI | Otto-Bliesner et al.       |
| Lust Inter gruenur      | CH 4 : 685 ppb                              | 127 119 21    |            | (if possible)    | OR Interactive carbon              |               |          | <del>2016</del> 2017       |
| (lig127k)               | N 2 O: 255 ppb                              |               |            |                  | cvcle                              |               |          |                            |
| 127 ky ago              | CFC 0                                                  |               |            |                  | OR fixed to present day            |               |          |                            |
|                         | O 3 : pre-industrial                        |               |            |                  | (depending on model                |               |          |                            |
|                         | 51                                                     |               |            |                  | complexity)                        |               |          |                            |
| Mid-Pliocene            | CO 2 : 400 ppm                              | as in PI      | modified   | as in PI         | Interactive vegetation             | as in PI      | as in PI | Haywood et al,             |
| Warm Period             |                                                        |               | (smaller)  |                  | OR Modified to mid-                |               |          | 2016                       |
|                         |                                                        |               |            |                  | Pliocene                           |               |          |                            |
| (midPlioceneEoi400)     |                                                        |               |            |                  | OR fixed to present day            |               |          |                            |
| 3.2 My ago              |                                                        |               |            |                  | (depending on model                |               |          |                            |
| 5.0                     |                                                        |               |            |                  | complexity)                        |               |          |                            |

5 Table 2: summary of change in boundary conditions with respect to *-piControl* (abbreviated as "PI") for each PMIP4-CMIP6 experiment \* Only for models without fully interactive dust (see section 3.3). \*\* interactive carbon cycle, with computation of some characteristics of the vegetation such as the leaf area index (LAI), but without full vegetation dynamics.

| Atmospheric variables         | top of atmosphere energy budget (global and annual average)                |  |  |
|-------------------------------|----------------------------------------------------------------------------|--|--|
|                               | surface energy budget (global and annual average)                          |  |  |
|                               | northern surface air temperature (annual average over northern hemisphere) |  |  |
|                               | southern surface air temperature (annual average over southern hemisphere) |  |  |
| Oceanic variables             | Sea surface temperatures (global and annual average)                       |  |  |
|                               | deep ocean temperatures (global and annual average over depths below       |  |  |
|                               | 2500m)                                                              |  |  |
|                               | deep ocean salinity (global and annual average over depths below 2500m)    |  |  |
|                               | Atlantic Meridional Overturning Circulation (maximum overturning           |  |  |
|                               | between 0 and 80°N and below 500 m depth)                                  |  |  |
| Sea ice variables             | northern sea-ice (annual average over northern hemisphere)                 |  |  |
|                               | southern sea-ice (annual average over southern hemisphere)                 |  |  |
| Carbon cycle variables        | Global carbon budget                                                       |  |  |
| Table 3: Variables to be save | d for the documentation of the spin-up phase of the models.                |  |  |

| Reference        | Variables    | Time period     | Comments                 | Data available from                      |
|------------------|--------------|-----------------|--------------------------|------------------------------------------|
| Mann et al.      | MAT          | 500-2006 CE     | Gridded data set (5°)    | http://science.sciencemag.org/conte      |
| (2009)           |              |                 |                          | nt/suppl/2009/11/25/326.5957.1256        |
|                  |              |                 |                          | .DC1                                     |
| PAGES 2k         | MAT          | past 2000 years | Individual sites; Arctic | https://www.ncdc.noaa.gov/cdo/f?         |
| Consortium       |              |                 | data updated 2014        | p=519:1:::::P1_STUDY_ID:12621            |
| (2013)           |              |                 |                          |                                          |
| Bartlein et al.  | MAT, MAP,    | 6000±500 yr BP; | Gridded data set (2°)    | https://www.ncdc.noaa.gov/paleo/s        |
| (2011)           | α, ΜΤCO,     | 21000±1000 yr   |                          | tudy/9897                                |
|                  | MTWA         | BP              |                          |                                          |
| MARGO            | Mean annual, | 21000±2000 yr   | Gridded data set (5°)    | http://www.ncdc.noaa.gov/paleo/study/    |
| Project          | winter,      | BP              |                          | 12034                                    |
| Members          | summer SST   |                 |                          | http://doi.pangaea.de/10.1594/PANGA      |
| (2009)           |              |                 |                          | EA.733406                                |
| Turney and       | MAT, SST     | Maximum         | Individual sites (100    | http://onlinelibrary.wiley.com/store/10. |
| Jones (2010)     |              | warmth during   | terrestrial; 162 marine) | 1002/jqs.1423/asset/supinfo/JQS_1423     |
|                  |              | LIG             |                          | _sm_suppInfo.pdf?v=1&s=1726938c4         |
|                  |              |                 |                          | 4b8762e15aaf17514fc076c855b8ed1          |
| Capron et al.    | MAT,         | 114-116ka, 119- | 47 high latitude sites   | doi.pangaea.de/10.1594/PANGAEA.84        |
| (2014); Capron   | summer SST   | 121ka, 124-     |                          | 1672                                     |
| et al. (subm.)   |              | 126ka, 126-128  |                          |                                          |
|                  |              | ka, 129-131ka   |                          |                                          |
| Dowsett et al,   | SST          | 3.264-3.025 Ma  | Further information      | http://www.nature.com/nclimate/journa    |
| (2012)           |              |                 | available in Dowsett et  | l/v2/n5/full/nclimate1455.html#supple    |
|                  |              |                 | al. (2016)               | mentary-
[revised manuscript text omitted]

20

---

## Author Response (AR3)

**The PMIP4 contribution to CMIP6 – Part 1: Overview and over-arching analysis plan – response to the reviewer and editor (3rd revision)**

**Dear Editor, dear reviewers of the successive versions of this manuscript,**

This is the fourth version of the manuscript describing the PMIP4 contribution to CMIP6. It is probably useful that at this stage, we summarise the different stages it went through. The first version was written according to the CMIP Panel's recommendations, namely, as stated in Eyring et al. (2016) and restated in the comment from the CMIP Panel on our manuscript:

*"Each of the 21 CMIP6-Endorsed MIPs is described in a separate invited contribution to this Special Issue. These contributions will detail the goal of the MIP and the major scientific gaps the MIP is addressing, and will specify what is new compared to CMIP5 and previous CMIP phases. The contributions will include a description of the experimental design and scientific justification of each of the experiments for Tier 1 (and possibly beyond), and will link the experiments and analysis to the DECK and CMIP6 historical simulations. They will additionally include an analysis plan to fully justify the resources used to produce the various requested variables, and if the analysis plan is to compare model results to observations, the contribution will highlight possible model diagnostics and performance metrics specifying whether the comparison entails any particular requirement for the simulations or outputs (e.g. the use of observational simulators). In addition, possible observations and reanalysis products for model evaluation are discussed and the MIPs are encouraged to help facilitate their use by contributing them to the obs4MIPs/ana4MIPs archives at the ESGF (see Section 3.3). In some MIPs additional forcings beyond those used in the DECK and CMIP6 historical simulations are required, and these are described in the respective contribution as well."*

We therefore tried to fulfill these requirements, and at the same time made sure that all the PMIP community, modellers and experts on the forcings and boundary conditions, agreed on the rationale and protocol described in this manuscript. The manuscript was therefore the result of long discussions and different types of compromises, with the main objective of the simulations were driven by the science that the community wants to investigate and that it also fits with scientific and technical constraints from CMIP6. This was also done while coordinating the content of the companion papers that focus more on the way the boundary conditions were determined for each period and how the PMIP4-CMIP6 periods fit in a more complete set of PMIP4 simulations. The resulting manuscript was 20 pages long for the main text, included 14 pages of references, 3 tables and 7 figures.

In the second version, following the reviewers' comments, we actually lengthened the manuscript to have more complete protocols and more complete information concerning the analysis plan. On the other hand we tried to shorten the introduction and "history" parts which were deemed boring. This more complete manuscript had 24 pages of main text, 16 pages of references, 4 tables and 6 figures.

This second version was not sent to review. Instead, the Editor wrote:

*"There has been some deliberation over this paper among the three editors handling the PMIP experiment description paper (one of whom has a self-declared conflict of interest, which was taken into account).*

*Fundamentally none of us consider the paper satisfactory at present, but James Annan and I (who do not have a conflict of interest) have come up with a potential solution, which should allow this PMIP4 "bookmark" to appear in the tome of the CMIP6 special issue.*

*We think that the paper should be reformulated as a review or summary paper. One framing could be to think in terms of being of interest and use to people currently of outside of the PMIP mainstream, who would benefit from an introduction to PMIP and a rough guide to which experiments they might be able to do. They probably wouldn't be so willing to wade through 4 different detailed experimental protocol papers to see if they would work out which are relevant to them and what the important differences are. Such people exist! We have met them! I imagine that the resulting summary would also be useful to new PMIP researchers as well as those who so far have only been involved in one of the PMIP sub-MIPs, but are interested in getting a wider appreciation of the whole project.*

*In order to fulfil these requirements the paper needs a major overhaul. The manuscript is too long and does not need to cover so much detail, I don't think it needs so much background, and many of the figures seem unnecessary, and they are anyway rather incomplete (eg 3 and 4 are two sets of partial forcings for a couple of experiments, 2 and 5 are bits of analysis that people have done for past PMIPs, but are by no means comprehensive). The tables are better, although they will need updating with the information from the now accepted/published sub-MIP papers. The text also needs to be made consistent with the other three papers, which we went to some lengths to coordinate, in terms of both the language and the actual requirements of the different MIPs."*

This was quite at odds with the first requirements from the CMIP6 panel. Nonetheless, we prepared a third version of the manuscript, by shortening the second version, especially sections 1 and 2 (again) and section 3 on the protocol. Section 4, on the analysis, was left after the protocol section. We chose to keep quite a complete protocol because it was a necessary requirement from the CMIP6 panel, and also because the other papers refer to it. This third version has 21 pages of text, 14 pages of references, 5 tables and 5 figures. It was meant to give an overview of the PMIP4 experiments for CMIP6, an overview of the protocol, in particular of new features of the protocol. This overview allowed a comparison of the forcings of the 5 PMIP4-CMIP6 periods and an overview of possible analyses, focusing on multi-period analyses, since single period protocols and analyses are now presented in full detail in the companion PMIP4 GMD papers. As such, we thought it would fulfill the CMIP6 panel requirements and the wishes of the Editor.

At this stage, the manuscript was sent to a fourth, new reviewer, who advised to reject the paper on the motive that it was found "neither particularly useful nor exciting to read". The Editor then took her decision (of asking major revision, this is very kind of her) based solely on this review and her own opinion, which obviously agreed with the review. She now asks for "*an enlightening overview of PMIP activities*". So the requirement has changed again. It should be noted that the CMIP6 panel never asked for "an enlightening review" from the start of the process. The requirements for this manuscript have now changed four times, we have made efforts to adapt each time, to fulfill the changing requirements, but we are now wondering if the process will ever stop.

Thus, we are now submitting the fourth version of the manuscript. The main text is 15 pages long (13 if the title page and abstract are not taken into account). There is 1 page of appendix, 14 pages of references (should we cut these too? this is not very nice for researchers who provided the work thanks to which PMIP can be conducted), 5 tables, 4 figures in the main text, 1 in the appendix.

In this version, we have followed the suggestion from the reviewer and the Editor to place the scientific motivation first. The paper is now structured as follows (changing the order of sections 3 and 4):

- 1) introduction (with some examples of challenging questions rising from the previous phases of PMIP),

- 2) description of the PMIP4-CMIP6 periods,

- 3) examples of scientific questions (termed "analyses" to fit the CMIP6 suggestion) which will benefit from the full PMIP4-CMIP6 and other CMIP6 experiments,

- 4) a short section on model configuration and experimental set up, which largely refers to the companion papers for single experiments but better documents the rationale of the requested documentation and output. For us, this section is mandatory because the companion papers, already published, refer to this one as this one gives common requirements for all PMIP4-CMIP6 experiments.

- 5) a short conclusion.

We hope that this version now reads better and fulfils both the CMIP6 panel and GMD requirements. At this stage, we may add that this manuscript was never meant to be a thrilling text. It was meant to be a protocol paper for GMD, following the CMIP6 panel recommendation. The solution found by the Editor so that we do not end up with very long paper including all justifications, experimental set up and analysis plans for all the periods (i.e. splitting the manuscript in this overview and companion papers) is much appreciated, but we feel her expectations might be too high with respect to what was initially expected from the CMIP6 panel.

**Response to reviewer 4**

We reproduce the reviewer's comments in *italic* and provide our reply in blue.

*This manuscript aims at providing an overview of the paleo climate modelling inter comparison (PMIP) contributions within CMIP6, as the detailed descriptions of the experiments are distributed over four separate manuscripts. The protocol is also sketched and some ideas for future analysis is provided in the contribution.*

*I struggled to find a good reason for publishing this paper, as I find it neither particularly useful nor exciting to read. As an introduction to the other four papers, it could have placed the scientific questions first and spend more energy on explaining how the new experiments are exciting and can be analysed in ways that were not possible before, e.g. multi-state constraints, but this part (section 4) appears more to be an afterthought and not particularly concrete. This could have been done in a shorter format.*

*Instead the reader is dragged through the experiment protocols, which are, at best redundant. In the worst case, however, the redundancy of protocol is confusing and can lead to errors in implementation. I therefore recommend stripping the manuscript of experimental protocol information, i.e. practically all of section 3 and parts of section 4.*

As explained above, the manuscript submitted to the fourth reviewer resulted from a rather long history of requirements from both the CMIP6 panel and the Editor, which were sometimes contradictory. We apologize for the text not being "particularly useful nor exciting to read" and have reorganized the text keeping the reviewer's suggestion in mind, i.e. putting the scientific questions first and keeping the information on the protocol to a minimum. To keep the analysis plan short, we focus on the multi-period analyses and benchmarking.

We have kept a section on the protocol, though, to warrant consistency in the set up for the different periods, which are now fully described in the "companion" papers. We have reduced this section to the bare minimum. However we still want this protocol section to be in, to ensure consistency of all the PMIP4-CMIP6 experiments and because the other papers refer to this paper for common aspects and documentation of the simulations.

*A perhaps more salient point concerns the repeated statements that the paleo experiments offer "out-of-sample" tests of models and that these will facilitate "to assess whether they have the correct sensitivity to forcing":*

*- The idea that paleo climate experiments are out-of-sample tests is strictly speaking only true the first time a particular experiment is conducted. Once the results are known to people that prepare the next round of experiments then it is no longer out of sample, rather there is a risk that protocol, boundary conditions, or even the models are modified such as to perform better, in some sense. That this is happening is admitted to on page 7, lines 8-10. Since this is now the fourth PMIP there can really be no justification of claiming the experiments are out of sample.*

*- There can be no such thing as a model with "correct sensitivity to forcing" as all models are wrong to begin with. Instead one can use model runs in different climates to constrain sensitivity using so-called emergent constraints, and/or one can see where models collectively or partly is inconsistent with proxies. I would suggest scanning the document for the word "correct" and rewrite these sentences.*

These ideas have been rephrased and now reads (first paragraph of the introduction):

"In making future projections, models are operating well outside the conditions under which they have been developed and validated. Changes in the recent past provide only limited evidence about how climate responds to changes in external factors and internal feedbacks of the magnitude expected in the future. Paleoclimate states radically different from those of the recent past provide a way to test model performance outside the range of recent climatic variations and to study the role of forcings and feedbacks in establishing these climates. Although palaeoclimate simulations strive for verisimilitude in terms of forcings and the treatment of feedbacks, none of the models used for future projection have been developed or calibrated to reproduce past climates."

*As I doubt the editor would follow my recommendation to postpone publication of this paper, I would recommend at least addressing the above before publication. I would, however, suggest that the authors try to learn from this, as they plan their future publications, that the papers should each have a clear purpose.*

Again, we apologize for this manuscript not being what the Editor and reviewer thought it should be, but as stated above, the purpose of the manuscript and the requirements from the Editor and reviewers have changed quite a lot. We hope that this version will be easier and less boring to read.

**Response to the Editor**

The reviewer recommends rejection. If this manuscript were merely a review of the PMIP experiments as presented in the GMD/CP special issue, then I would reject, and encourage the authors to simply write a non-peer-reviewed preface to the special issue. However, this manuscript is also part of the CMIP6 special issue, and has value in that context, as an overview paper that summarises PMIP activities to non-PMIP modellers (and to PMIP modellers that are involved in only part of the project).

My first impression of this revision was that the manuscript is still much too long to serve its purpose as an enlightening overview of PMIP activities. However, other scientists are usually more patient than I am with overly long papers, so I decided to send the manuscript back out for review. I chose the reviewer quite carefully to be someone who I think is precisely the kind of scientist to whom the manuscript should be of interest (a paleo-curious climate modeller). It is therefore thoroughly disappointing that they find the manuscript "neither particularly useful nor exciting to read". The reviewer's general suggestions sound very attractive, "As an introduction to the other four papers, it could have placed the scientific questions first and spend more energy on explaining how the new experiments are exciting and can be analysed in ways that were not possible before, e.g. multi-state constraints, but this part (section 4) appears more to be an afterthought and not particularly concrete. This could have been done in a shorter format. ... In the worst case, however, the redundancy of protocol is confusing and can lead to errors in implementation. I therefore

recommend stripping the manuscript of experimental protocol information, i.e. practically all of section 3 and parts of section 4."

A reduction in length of 50% would not be inappropriate in my (possibly still a bit extreme!) opinion.

I suspect that the other comments from the reviewer (about the out of sample-ness of the experiments and correctness of forcing) really refer to a slight imprecision in language rather than any fundamental disagreement, and so it is worth amending the text to make sure that such misunderstandings are minimised.

We hope we have now addressed all concerns with this manuscript. We reduced it by 9 pages compared to the longest version, and the main text now holds in 12 1/2 pages, which is very short compared to the other papers in the CMIP6 special issue. We did not plan to write "an enlightening overview of PMIP activities" from the start, this will come in due time, when the PMIP4 results are out and numerous enough to perform enlightening analyses.

[revised manuscript text omitted]

* * *
Margin annotations (tracked changes):


5 the second key CMIP6 question "What are the origins and consequences of systematic model biases?", PMIP simulations and data-model comparisons will show whether the biases in the present-day simulations are found in other climate states. Also, analyses of PMIP simulations will show whether present-day biases have an impact on the magnitude of simulated climate changes. Finally, PMIP is also relevant to the third CMIP6 question "How can we assess future climate changes given climate variability, predictability and uncertainties in scenarios?"

10 through examination of these questions for documented past climate states and via the use of the last millennium simulations as a reference state for natural variability.

The detailed justification of the experimental protocols and analysis plans for each period are given in a series of companion papers: Otto-Bliesner et al. (2017) for the *midHolocene* and *lig127ka* experiments, Kageyama et al.

15 (2017) for the *lgm*, Jungclaus et al. (2017) for the *past1000* and Haywood et al. (2016) for the *midPliocene-eoi400* experiment. These papers also explain how the boundary conditions for each period should be implemented and include the description of sensitivity studies using the PMIP4-CMIP6 simulation as a reference. Here we provide an overview of the PMIP4-CMIP6 simulations and highlight the scientific questions that will benefit from the CMIP6 environment. In section 2, we give a summary on the PMIP4-CMIP6 periods

20 and the associated forcings and boundary conditions. The analysis plan is outlined in Section 3. Critical points in the experimental set-up are briefly described in section 4. A short conclusion is given in section 5.

**2. The PMIP4 experiments for CMIP6 and associated paleoclimatic and paleoenvironmental data**

The choice of the climatic periods for CMIP6 is based on past PMIP experience and is justified by the need to address new scientific questions, while also tracing back model evolution and ability to represent these climate

25 states since the first phase of PMIP (Table 1). The forcings and boundary conditions for each PMIP4-CMIP6 paleoclimate simulation are summarised in Table 2. All the experiments can be run independently and have value for comparison to the CMIP6 DECK (Diagnostic, Evaluation and Characterization of Klima) and historical experiments (Eyring et al., 2016). They are therefore all considered as Tier 1 within CMIP6. It is not mandatory for groups wishing to take part in PMIP4-CMIP6 to run all five PMIP4-CMIP6 experiments. It is however

30 mandatory to run at least one of the two entry cards, i.e. the *midHolocene* or the *lgm*.

**2.1 PMIP4-CMIP6 entry cards: the mid-Holocene (*midHolocene*) and last glacial maximum (*lgm*)**

The MH and LGM periods are strongly contrasting climate states. The MH provides an opportunity to examine the response to orbitally-induced changes in the seasonal and latitudinal distribution of insolation (Figure 1). It is a period of strongly enhanced northern hemisphere summer monsoons, extra-tropical continental aridity and much warmer summers. The LGM provides an opportunity to examine the impact of changes in ice sheets and continental extent (which increases due to the drop in sea level, Figure 2) and of the decrease in atmospheric greenhouse gases on climate. The LGM is particularly relevant because the forcing and temperature response from the LGM to present was as large as that projected from present to the end of the 21st century (Braconnot et al., 2012).

Evaluation of the PMIP3-CMIP5 MH and LGM experiments has demonstrated that climate models simulate changes in large-scale features of climate that are governed by the energy and water balance reasonably well (Harrison et al., 2014, 2015, Li et al., 2013), including changes in land-sea contrast and high-latitude amplification of temperature changes (Izumi et al., 2013; Izumi et al., 2015). These results confirm that the simulated relationships between large-scale patterns of temperature and precipitation change in future projections are credible. However, the PMIP3-CMIP5 simulations of MH and LGM climates show only moderate skill in predicting reconstructed patterns of climate change overall (Hargreaves et al., 2013; Hargreaves and Annan, 2014; Harrison et al., 2014; Harrison et al., 2015). This arises because of persistent problems in simulating regional climates. For example, state-of-the-art models cannot fully properly reproduce the northward penetration of the African monsoon in response to the MH orbital forcing (Perez-Sanz et al., 2014; Pausata et al., 2016), which was already noted in PMIP1 (Joussaume et al, 1999). While this likely reflects inadequate representation of feedbacks, model biases could also contribute to this mismatch (e.g. Zheng and Braconnot, 2013). Systematic benchmarking of the PMIP3-CMIP5 MH and LGM also show that better performance in paleoclimate simulations is not consistently related to better performance under modern conditions, stressing that the ability to simulate modern climate regimes and processes does not guarantee that a model will be good at simulating climate changes (Harrison et al., 2015).

For PMIP4-CMIP6, we have modified the experimental design of the *midHolocene* and *lgm* experiments with the aim of obtaining more realistic representations of these climates (Table 2, Otto-Bliesner et al., 2017 for *midHolocene* and Kageyama et al., 2017 for *lgm*). One of these modifications is the inclusion of changes in atmospheric dust loading (Figure 3), which can have a large effect on regional climate changes. For

**Déplacé vers le haut [4]:** All the experiments can be run independently and have value for comparison to the CMIP6 DECK (Diagnostic, Evaluation and Characterization of Klima) and historical experiments (Eyring et al., 2016). They are therefore all considered as Tier 1 within CMIP6

**Déplacé vers le haut [5]:** It is not mandatory for groups wishing to take part in PMIP4-CMIP6 to run all five PMIP4-CMIP6 experiments. It is however mandatory to run at least one of the two entry cards, i.e. the *midHolocene* or the *lgm*.¶

**Déplacé (insertion) [6]**

**Déplacé vers le haut [6]:** 2012). ¶

**Déplacé (insertion) [7]**

**Déplacé vers le haut [7]:** reconstructed patterns of climate

**Déplacé (insertion) [8]**

**Déplacé vers le haut [8]:** While this likely reflects inadequate

**Déplacé (insertion) [9]**

**Déplacé vers le haut [9]:** ¶

*midHolocene*, realistic values of the concentration of atmospheric $CO_2$ and other trace gases will be also used (Otto-Bliesner et al. 2017). This makes this experiment more realistic than in PMIP3 where it was designed as a simple test to changes in insolation forcing. The PMIP3-CMIP5 *lgm* experiments considered a single ice sheet reconstruction (Abe-Ouchi et al., 2015). However, there is uncertainty about the geometry of the ice sheets at the Last Glacial Maximum. The protocol for the PMIP4-CMIP6 *lgm* simulations accounts for this uncertainty and includes a choice between the old PMIP3 ice sheet (Abe-Ouchi et al., 2015) or one of two new reconstructions: ICE-6G_C (Argus et al., 2014; Peltier et al., 2015) and GLAC-1D (Tarasov et al., 2012; Briggs et al., 2014, Ivanovic et al., 2016). Altogether, the *lgm* experiments will allow testing the impact of these different ice sheet reconstructions (Figure 2) and of the dust forcing, which was not included in the PMIP3 set-up.

**2.2 The last millennium (*past1000*)**

The millennium prior to the industrial era, 850-1849 CE, provides a well-documented (e.g. PAGES2k-PMIP3 group, 2015) period of multi-decadal to multi-centennial changes in climate, with contrasting periods such as the Medieval Climate Anomaly and the Little Ice Age. This interval was characterised by variations in solar, volcanic and orbital forcings (Figure 1), which acted under climatic background conditions similar to today. This interval provides a context for analysing earlier anthropogenic impacts (e.g. land-use changes) and the current warming due to increased atmospheric greenhouse gas concentrations. It also helps constrain the uncertainty in the future climate response to a sustained anthropogenic forcing.

[revised manuscript text omitted]

**3.1. Comparisons with paleoclimate and paleoenvironemental reconstructions, benchmarking and beyond**

Model-data comparisons for each period will be one of the first tasks conducted after completion of these simulations. One new feature common to all periods is that we will make full use of the fact that modelling groups must also run the *piControl* and *historical* experiments. Indeed, existing paleoclimate reconstructions have used different modern reference states, and this has been shown to have an impact on the magnitude of reconstructed changes (e.g. Hessler et al., 2014). Running both the *piControl* and *historical* simulations was not systematic in previous phases of PMIP, which prevented investigation of the impact of these reference states for model-data comparisons. The new set of simulations will provide a way of quantifying this source of reconstruction uncertainty, as will comparisons with present-day observations and reanalysis data sets (Obs4MIPS, Ferraro et al, 2015).

Comparisons of PMIP4-CMIP6 simulations with available paleoclimatic reconstructions is a unique contribution to CMIP6 in terms of evaluating model biases for climates different from the historical one. An ensemble of metrics has already been developed for the PMIP3-CMIP5 *midHolocene* and *lgm* simulations (e.g. Harrison et al. 2014). These, applied to the PMIP4-CMIP6 *midHolocene* and *lgm* "entry card" simulations, will provide a rigorous assessment of model improvements compared to previous phases of PMIP. Furthermore, for the first time, thanks to the design of the PMIP4-CMIP6 experiments, we will be able to take the impact of uncertainties in the forcings into account in the benchmarking. The benchmarking metrics will also be expanded to other periods and data sets so that systematic biases for different periods and for the present-day can be compared. Benchmarking the ensemble of the PMIP4-CMIP6 simulations, for all the periods, will therefore allow quantifying the climate-state dependence of the model biases, a topic which is highly relevant for a better assessment of potential biases in the projected climates in CMIP6.

Another promising activity will consist in analysing the potential relationships between model biases in different regions and/or in different variables (such as temperature vs. hydrological cycle) across the PMIP ensemble, as

well as for the recent climate. One further objective for the PMIP4-CMIP6 benchmarking will be to develop more process-oriented metrics making use of the fact that paleoclimatic data document different aspects of climate change. There are many aspects of the climate system which are difficult to measure directly, and which are therefore difficult to evaluate using traditional methods. The "emergent constraint" (e.g. Sherwood et al., 2014) concept, which is based on identifying a relationship to a more easily measurable variable, has been successfully used by the carbon-cycle and modern climate communities and holds great potential for the analysis of paleoclimate simulations. This could be particularly valuable to examine the realism of e.g. cloud feedbacks in the simulations or the contribution of seasonal climate changes to hydrological budgets. Using multiple time periods to examine "emergent" constraints will ensure that they are robust across climate states.

**3.2. Analysing the response of the climate system to multiple forcings**

Multi-period analyses provide a way of determining whether systematic model biases affect the overall response and the strength of feedbacks independent of climate state. One challenge will be to develop new approaches to analyse the PMIP4-CMIP6 ensemble so as to separate the impacts of model resolution, content, or complexity on the simulated climate. Similarly the uncertainties in boundary conditions will be addressed for periods for which alternative forcing is proposed.

Quantifying the role of forcings and feedbacks in creating climates different from today has been a focus of PMIP for many years. Many CMIP6 models will include new processes, such as dust, or improved representations of major radiative feedback processes, such as clouds. This will allow a broader analysis of feedbacks than was possible in PMIP3-CMIP5. We will evaluate the impact of these new processes and improved realisations of key forcings on climates at global, large-scale (e.g. polar amplification, land-sea contrast) as well as regional scales, together with the mechanisms explaining these impacts. A particular emphasis will be put on the modulation of the climate response to a given forcing by the background climate state and how it affects changes in cloud feedbacks, snow and ice sheets (such as in e.g. Yoshimori et al., 2011), vegetation and ocean deep water formation. Identification of similarities between past climates and future climate projections such as the one found for land-sea contrast or polar amplification (Izumi et al., 2013; Masson-Delmotte et al., 2006; Izumi et al., 2015) or for snow and cloud feedbacks for particular seasons (Braconnot and Kageyama, 2015) will be used to provide better understanding of the relationship between patterns and time scales of external forcings and patterns and timing of the climate responses.

**Déplacé vers le bas [45]:** These attempts also ignored uncertainties in forcings and boundary conditions. PMIP4-CMIP6 is expected to result in a much larger ensemble of *lgm* experiments. The issue of climate sensitivity (*sensu stricto*) and earth-system sensitivity (PALEOSENS Project Members, 2012) will also be examined through joint analysis of multiple paleoclimate simulations and climate reconstructions from different archives.

Attempts

**Déplacé vers le bas [44]:** Hargreaves et al., 2012; Harrison et al., 2014; Hopcroft and Valdes, 2015b).

**Déplacé (insertion) [40]**

**Déplacé (insertion) [41]**

**Déplacé (insertion) [38]**

**Déplacé (insertion) [39]**

[revised manuscript text omitted]

The community using PMIP simulations is very broad, from climate modellers and palaeoclimatologists to biologists studying changes in biodiversity and archaeologists studying potential impacts of past climate changes on human populations. Because of this, we do not aim to give a comprehensive plan of PMIP analyses, but instead here we focus on topics closely related to the CMIP6 key questions. Each PMIP4-CMIP6 period has been selected for specific reasons (Table 1). Here, we list several analyses which are important for single periods as well as for the full PMIP4-CMIP6 ensemble, starting first by presenting examples of paleoclimate reconstructions available for comparison to the PMIP4-CMIP6 simulations. .

**4.1 Paleoclimatic and paleoenvironmental reconstructions, model-data comparisons.**

on human populations. Because of this, we do not aim to give a comprehensive plan of PMIP analyses, but instead here we focus on topics closely related to the CMIP6 key questions. Each PMIP4-CMIP6 period has been selected for specific reasons (Table 1). Here, we list several analyses which are important for single periods as well as for the full PMIP4-CMIP6 ensemble, starting first by presenting examples of paleoclimate reconstructions available for comparison to the PMIP4-CMIP6 simulations. .

**4.1 Paleoclimatic and paleoenvironmental reconstructions, model-data comparisons.**

| Page 11 : [39] Supprimé | masa | 29/12/2017 22:30:00 |
|---|---|---|

this

| Page 11 : [39] Supprimé | masa | 29/12/2017 22:30:00 |
|---|---|---|

this

| Page 11 : [39] Supprimé | masa | 29/12/2017 22:30:00 |
|---|---|---|

this

| Page 11 : [39] Supprimé | masa | 29/12/2017 22:30:00 |
|---|---|---|

this

| Page 11 : [39] Supprimé | masa | 29/12/2017 22:30:00 |
|---|---|---|

this

| Page 11 : [39] Supprimé | masa | 29/12/2017 22:30:00 |
|---|---|---|

this

| Page 12 : [40] Supprimé | masa | 29/12/2017 22:30:00 |
|---|---|---|

Compared to the PMIP3-CMIP5 models, many CMIP6 models will include new processes, such as dust, or improved representations of major radiative feedback processes, such as clouds. Improvements to the design of the *past1000*, *midHolocene* and *lgm* experiments are also proposed (section 2). We will evaluate the impact of these changes on the PMIP4-CMIP6 climates at global, large-scale (e.g.

[revised manuscript text omitted]

Anglais (Royaume-Uni)

| Page 33 : [70] Mis en forme | masa | 29/12/2017 22:30:00 |
| --- | --- | --- |

Anglais (Royaume-Uni)

| Page 33 : [71] Supprimé | masa | 29/12/2017 22:30:00 |
| --- | --- | --- |

Rubino, M., Velders, G. J. M., Vollmer, M. K., Wang, R. H. J., and Weiss, R.: Historical greenhouse gas concentrations for climate modelling (CMIP6), Geosci. Model Dev., 10, 2057-2116, https://doi.org/10.5194/gmd-10-2057-2017, 2017.

| Page 33 : [72] Mis en forme | masa | 29/12/2017 22:30:00 |
| --- | --- | --- |

pb_toc_link, Police :10 pt

| Page 33 : [73] Supprimé | masa | 29/12/2017 22:30:00 |
| --- | --- | --- |

Legrande

| Page 33 : [73] Supprimé | masa | 29/12/2017 22:30:00 |
| --- | --- | --- |

Legrande

| Page 33 : [73] Supprimé | masa | 29/12/2017 22:30:00 |
| --- | --- | --- |

Legrande

---

## Author Response (AR4)

Dear Editor

The manuscript has been entirely proof-read and its English corrected by several native speakers. We have attempted to clarify the text following your indications, especially at the beginning of Section 3.1.

We have also modified table 2 to warrant consistency between the different GMD PMIP4-CMIP6 papers. By stating the greenhouse gases and insolation values should be those of e.g. 6 kyr BP, we invite the reader to refer to the specific GMD companion paper for the exact values.

Finally, we have added a section on Author Contributions.

We hope that you will find this version suitable for publication.

Best wishes

Masa Kageyama, on behalf of the authors.

[revised manuscript text omitted]

---

## Author Response (AR5)

Response to Editor – revision 5

Dear Editor,

Thank you for your comments (which are reproduced in black). We reply to each of them in *blue and italics*.

One substantive comment.

p12 "No additional interactive component should be included in the model unless it is already included in the DECK version. Such changes would affect the global energetics (Braconnot and Kageyama, 2015) and therefore prevent rigorous analyses integrating across multiple time periods or between MIPs (section 3).

Because of this, even though environmental records show that natural vegetation patterns during each of the PMIP4-CMIP6 periods were different from today, the PMIP4-CMIP6 palaeoclimate simulations should use the same model configuration to the DECK and historical simulations. If the DECK and historical simulations use dynamic vegetation, then the PMIP4-CMIP6 palaeoclimate simulations should do so too. If the DECK and
historical simulations use prescribed vegetation, then the same vegetation should be prescribed in the PMIP4- CMIP6 palaeoclimate simulations."

Please rephrase this to be consistent with what is written in the other PMIP4 papers. To start with I suggest removing, "Such changes would affect the global energetics (Braconnot and Kageyama, 2015) and therefore prevent rigorous analyses integrating across multiple time periods or between MIPs (section 3). Because of this."

A main focus of this manuscript is the promotion of PMIP as a series of experiments, the output from which may be compared with real data. If you knowingly implement your protocols to have very incorrect vegetation then these arguments are rendered much less convincing. It is not at all clear here why dynamic vegetation should be permitted but different fixed-vegetation not permitted, especially when many other boundary conditions are changed for the experiments. I went back to the LGM paper, and there it is simply stated that present day vegetation should be used without a reason being given. However, in that paper, modellers are also encouraged to run sensitivity experiments with altered vegetation, as well as the default protocol. This makes sense to me in the following way: the response to vegetation is so variable between models that we want all those who alter their paleo fixed-vegetation to first do the run with present day fixed-vegetation - otherwise we have no hope of understanding the response of the models. Not sure if that is your intended meaning, but rather than attempt to add further explanation, which is not anyway included in the LGM paper, I suggest that here you again encourage modellers to perform the vegetation sensitivity tests. (That these sensitivity tests may be not strictly CMIP6 runs is unimportant - this manuscript has to make sense scientifically!)

*This comment covers different points of the experimental set-up.*

*One first point is that we want to ensure the same model is used for PMIP4-CMIP6 experiments, the DECK experiments, and if possible the other CMIP6 experiments. For instance, if a group chooses to*

*use interactive vegetation or aerosols for the* lgm *experiment but not for the* piControl *experiment, we will actually end studying two different models. There is nothing that will warrant that these models have the same climate sensitivity, and the same responses to external forcings and even the same* piControl *characteristics. We want to avoid this situation. Groups might want to represent interactive vegetation or dust to get more realistic experiments, but they have to also perform the DECK experiments. This was the main message of the first paragraph of section 4.1. We do not want to discourage modelling groups from running experiments which are the most realistic as possible, but we want to ensure consistency of the model version used for the DECK and PMIP4-CMIP6 experiments.*

*We have therefore modified the end of this paragraph, hoping that the message is now clearer:*

*"Such changes would prevent rigorous analyses of the responses to forcings across multiple time periods or between MIPs (section 3) because the differences between the experiments could then arise from both the models' characteristics and the response to changes in external forcings. Adding an interactive component usually affects the piControl simulation as well as simulations of past climates (Braconnot et al., 2007) so it is very important that experiments for PMIP4-CMIP6 and the DECK are run with exactly the same model version."*

*This point is common to all PMIP4-CMIP6 simulations, it is very important, we have therefore written it in the overview. It is not new compared to the companion papers. Actually it is stated in the first paragraph of section 3.1 of the LGM paper.*

*The second point is about using more realistic vegetation. It is clearly stated in the companion paper on interglacials that the data coverage is not sufficient to produce global vegetation maps for these periods, which would be used in order to prescribe realistic vegetation. It is the same situation for the lgm experiment, even though it is not written in the corresponding paper as clearly as in the interglacial one. If a model does not include dynamic vegetation, then a first simulation has to be run with present day vegetation. The resulting climate can then be used to compute an LGM vegetation map. This vegetation can then be used for a second lgm simulation, which will therefore be a sensitivity experiment to the vegetation. This is what the Editor suggests and apart from directly using a dynamic vegetation model within the Earth System Model, there is no other way of taking the LGM vegetation changes into account. We have therefore added one sentence to encourage the groups to run sensitivity experiments to vegetation, as suggested by the Editor. This is at the end of the second paragraph of Section 4.1.*

Apart from that, it will helpful to fix the following.

p3 l22 "feedbackfeedbacks" → corrected!

p4 l23 "between climate-ice sheet system"
Missing "the"

l29-31
Can't understand this sentence:

" Finally, PMIP is also relevant to the third CMIP6 question "How can we assess future climate changes given climate variability, predictability and uncertainties in scenarios?" through examination of these questions for documented past climate states and via the use of the last millennium simulations as a reference state for natural variability."

*We modified this sentence to make it clearer:" Finally, PMIP is also relevant to the third CMIP6 question "How can we assess future climate changes given climate variability, predictability and uncertainties in scenarios?" because the simulation of the last millennium climate includes more processes (e.g. volcanic and solar forcings) to describe natural climate variability than the piControl experiment."*

p6 l22 "Mediaeval Climate Anomaly"
Google Trends for the United Kingdom show no hits at all for this phrase, and far, far fewer hits for "Mediaeval" vs. "Medieval". I think, therefore, it makes no sense to use your alternative spelling.
(The split for paleoclimate and palaeoclimate is much more even!)
(Don't even think of trying "musaeum".)

*The Oxford disctionary gives both writings (I actually checked). I put it back to the previous spelling.*

p7 l6-8 "The PMIP4-CMIP6 past1000 protocol will use a new, more comprehensive reconstruction of volcanic forcing (Sigl et al., 2015) and ensures a more continuous transition from the pre-industrial past to the future."

"ensures" -> "ensure"
or
"will use" -> "uses"

…depending on which tense you want to use. First option sounds a bit better than the second, to me.

*Well, I chose the second. Now that the protocols are published, there is no reason to employ the future tense.*

l8-9 "The final choices resulted from strong interactions with the groups producing the different forcing fields for the historical simulations (Jungclaus et al, 2017)."

"choices" here is ambiguous. Are you talking about choices you made (perhaps "decisions"?) to produce the forcings for a single protocol, or are you alluding a the range of possible forcings that people running the experiments may choose from?

*"Decisions" is indeed more appropriate.*

p8 12-14 "However, as is the case for the Last Interglacial, the PlioMIP simulations were not always derived from the same models as for the CMIP5 future projections."

perhaps,
"..the PlioMIP simulations were not always performed using the same models that were used for the

CMIP5 experiments."

Apart from awkward syntax, surely PlioMIP models were also different from those used in CMIP5-PMIP3 (e.g. the PMIP3 LGM).

*Yes, this is true. The sentence was modified according to the Editor's suggestion.*

l16-18
"The PMIP4-CMIP6 midPliocene-eoi400 experiment (Haywood et al., 2016) is designed to elucidate the long- term response of the climate system to a near modern concentration of atmospheric $CO_2$ (long term climate sensitivity or Earth system sensitivity)."

I have seen "modern" used in the paleoclimate literature to have a rather vague meaning, sometimes closer to pre-industrial. Here I think you specifically mean close to 400ppm, so it might be clearer to say this…? After all, people may read this paper in 20 years time when CO2 is 500ppm... !

*Yes. I have modified the sentence to: "The PMIP4-CMIP6 midPliocene-eoi400 experiment (Haywood et al., 2016) is designed to elucidate the long-term response of the climate system to a concentration of atmospheric $CO_2$ close to the present one: 400 ppm"*

p9 l22 "also run historical experiment, in addition to the piControl one. "

Missing "the"

*OK!*